# Transformers Trained via Gradient Descent Can Provably Learn a Class of Teacher Models

**Chenyang Zhang**[†], **Qingyue Zhao**[‡], **Quanquan Gu**[‡], **Yuan Cao**[†]
[†] The University of Hong Kong    [‡] University of California, Los Angeles
`chyzhang@connect.hku.hk,zhaoqy24@ucla.edu`
`qgu@cs.ucla.edu,yuancao@hku.hk`

## Abstract

Transformers have achieved great success across a wide range of applications, yet the theoretical foundations underlying their success remain largely unexplored. To demystify the strong capacities of transformers applied to versatile scenarios and tasks, we theoretically investigate utilizing transformers as students to learn from a class of teacher models. Specifically, the teacher models covered in our analysis encompass convolution layers with average pooling, graph convolution layers, and various classic statistical learning models, including a variant of sparse token selection models (Sanford et al., 2023; Wang et al., 2024) and group-sparse linear predictors (Zhang et al., 2025c). When learning from this class of teacher models, we prove that one-layer transformers with simplified "position-only" attention can successfully recover all parameter blocks of the teacher models, thus achieving the optimal population loss. Building upon the efficient mimicry of trained transformers towards teacher models, we further demonstrate that they can generalize well to a broad class of out-of-distribution data under mild assumptions. The key in our analysis is to identify a fundamental bilinear structure shared by various learning tasks, which enables us to establish unified learning guarantees for these tasks when treating them as teachers for transformers.

## 1 Introduction

Transformers have rapidly become a cornerstone in the field of modern machine learning, demonstrating exceptional performance and versatility across diverse applications, including natural language processing (Vaswani et al., 2017; Radford et al., 2019; OpenAI, 2023; Devlin, 2018; Achiam et al., 2023; Vig & Belinkov, 2019; Touvron et al., 2023; Ouyang et al., 2022), computer vision (Dosovitskiy et al., 2020; Rao et al., 2021; Liu et al., 2021; Yuan et al., 2021; Zhang et al., 2025b;a), and reinforcement learning (Jumper et al., 2021; Chen et al., 2021; Janner et al., 2021; Reed et al., 2022). Acting as a critical component of transformers, self-attention layers assign varying weights to features based on their relevance and embedded positional context. This design principle intuitively endows transformers with a remarkable ability to efficiently process both structural and positional information, as empirically validated in numerous applications mentioned above. However, despite their profound impact, the theoretical foundations of transformers, especially the mechanisms of how self-attention layers work, remain largely unexplored due to their intricate architecture.

Some recent theoretical studies aimed to understand transformers by analyzing their capability in solving specific tasks (Zhang et al., 2024b; Frei & Vardi, 2025; Jelassi et al., 2022; Wang et al., 2024; Zhang et al., 2025c). Specifically, Zhang et al. (2024b) considered in-context linear regression, and demonstrated that for Gaussian data, a one-layer transformer with linear attention can perform linear regression based on the context, and then apply the obtained linear model to make predictions on query data. Later, Frei & Vardi (2025) further extended the setting to in-context linear classification, and studied the in-context benign overfitting phenomena when learning from Gaussian mixture data. Jelassi et al. (2022) investigated a specific data model based on the 'patch association' assumption, where an image is divided into disjoint partitions, and patches within the same partition share similar characteristics. They theoretically demonstrate that a one-layer vision transformer (ViT) can extract

the spatial structure among patches when trained on this data model. Wang et al. (2024) studied a problem termed 'sparse token selections', where the objective is to find the average of several tokens from specific positions, and they proved that a one-layer transformer can successfully solve this task on Gaussian data when the positional information of the target positions is embedded into the query token. Zhang et al. (2025c) considered a group sparse linear model, where the input's label is determined by features from only one of several input feature groups (the 'label-relevant group'), and prove that for Gaussian data, a trained one-layer transformer can achieve correct classification by identifying features from this group and learning the ground truth linear classifier. Although these works have offered valuable insights into the underlying mechanisms of transformers, their focus on very specific learning tasks limits the generality of their theoretical findings, prompting us to seek a unified theoretical framework accounting for a broader range of examples.

Despite the distinctions among the model simplifications and technical assumptions, we observe that for some learning tasks discussed above, including a variant of the sparse token selection (Sanford et al., 2023; Wang et al., 2024), the group sparse linear predictors (Zhang et al., 2025c), and patch association (Jelassi et al., 2022), their true responses are essentially given by bilinear functions. In addition, the linear attention studied in Zhang et al. (2024a); Frei & Vardi (2025) inherently constitutes a bilinear structure with respect to its parameter matrices. Motivated by this observation, we define a general class of "teacher models" that employ a bilinear structure, and investigate the setting where one-layer transformers are trained as "student" models under the supervision from these teacher models. Our framework not only encompasses the learning tasks from prior works but also covers popular, previously unexplored models such as convolution layers with average pooling and graph convolution layers on regular graphs. The purpose of our analysis is to establish unified theoretical guarantees for one-layer transformer models trained with gradient descent in learning this class of teacher models.

The major contributions of this work are as follows.

- We theoretically demonstrate that one-layer transformers trained via gradient descent can effectively recover a general class of teacher models. To support this claim, we establish a tight convergence guarantee for the population loss, with matching upper and lower bounds at the rate of $\Theta\left(\frac{1}{T}\right)$, where $T$ is the iteration number of gradient descent. We also establish out-of-distribution generalization bounds for the obtained transformer model and demonstrate that it is competitive with the teacher model over a wide rage of learning tasks. This illustrates the effectiveness and robustness of transformer models in learning from diverse teacher models.

- Our theory covers a wide range of learning tasks, including some settings closely related to those studied in (Wang et al., 2024; Zhang et al., 2025c). Specifically, Wang et al. (2024) study a type of "sparse token selection" task where the goal is to select a number of target input tokens specified by a query column, and then output their average. Assuming that the positions of the target tokens are randomly generated for each data point, the authors establish an $\mathcal{O}\left(\frac{\log(T)}{T}\right)$ convergence rate. In comparison, our setting covers a slightly different task where the target positions are fixed but are not explicitly fed to model, and our theoretical results demonstrate a tight $\Theta\left(\frac{1}{T}\right)$ convergence rate with matching upper and lower bounds. Compared with Zhang et al. (2025c) which focuses on group sparse linear classification, our work provides complementary results and demonstrates that transformers can also perform efficient group sparse linear regression.

- Experiments on both synthetic and real-world data are conducted to verify our theory through the examples of learning a convolution layer with average pooling, learning a graph convolution layer with regular graphs, learning sparse token selection, and group sparse linear regression. In all experiments, we can observe clear loss convergence and parameter convergence that match our theory. The experiments setup does not exactly match our theory assumptions, indicating that our theory conclusions can also hold in more practical training setups and real-data learning tasks.

## 2 PROBLEM SETUP

In this section, we introduce the definition of the teacher models we study in this paper, and give various examples covered in our definition.

We consider a teacher model with an input matrix $\mathbf{X} \in \mathbb{R}^{d \times D}$ of the following form:

$$f^*(\mathbf{X}) = \sigma(\mathbf{V}^*\mathbf{X}\mathbf{S}^*), \tag{2.1}$$

where $\mathbf{V}^* \in \mathbb{R}^{M \times d}$ is the ground truth value matrix of the teacher model, and $\mathbf{S}^* \in \mathbb{R}^{D \times D}$ is the ground truth softmax scores. Each column of $\mathbf{S}^*$ has $K$ non-zero entries equivalent to $\frac{1}{K}$. In addition, $\sigma(\cdot)$ denotes either an identity map, ReLU, or Leaky ReLU activation function.

The teacher models defined in (2.1) can cover a general class of functions (models). Notably, when $K = 1$ and all the non-zero entries of $\mathbf{S}^*$ appear on its diagonal, $\mathbf{S}^*$ equals the identity matrix $\mathbf{I}_D$. In this scenario, the teacher model (2.1) reduces to $f^*(\mathbf{X}) = \sigma(\mathbf{V}^*\mathbf{X})$, and can be seen as a single-layer neural network. Besides this naive example where $\mathbf{S}^* = \mathbf{I}_D$, the teacher model (2.1) also includes some other common architectures and models. We discuss these examples in the following.

**Example 2.1** (Single convolutional layer with average pooling). We consider a convolution layer consisting of convolution operation, average pooling, and then the activation function. The convolution operation is essentially performed by taking inner products between each convolution kernel with each patch of the input. We consider a convolution layer with $M$ (vectorized) kernels $\mathbf{v}_1^*, \ldots, \mathbf{v}_M^*$, and consider an input consisting of $D$ (vectorized) patches $\mathbf{x}_1, \ldots, \mathbf{x}_D$. In average pooling, we take averages according to a partition of the $D$ patches. Let $\mathcal{G} = \{g_1, g_2, \ldots, g_J\}$ be a disjoint partition of $[D]$, forming $J$ pooling groups with $|g_j| = K$, $j \in [J]$. Then the final output of this convolution layer corresponding to the $j$-th pooling group and the $m$-th kernel is given as

$$\sigma\left(\frac{1}{K}\sum_{i \in g_j}\langle\mathbf{v}_m^*, \mathbf{x}_i\rangle\right) = \sigma(\mathbf{v}_m^{*\top}\mathbf{X}\mathbf{1}_{g_j}/K), \; m \in [M], \; j \in [J],$$

where $\sigma$ is the activation function, $\mathbf{X} = [\mathbf{x}_1, \mathbf{x}_2, \ldots, \mathbf{x}_D] \in \mathbb{R}^{d \times D}$, and $\mathbf{1}_{g_j} \in \mathbb{R}^D$ is a vector whose entries are 1 for indices in $g_j$, and 0 otherwise. Then, we can summarize all outputs into a matrix:

$$F_{\mathrm{CNN}}(\mathbf{X}) = \sigma(\mathbf{V}^*\mathbf{X}[\mathbf{1}_{g_1}, \ldots, \mathbf{1}_{g_J}]/K) \in \mathbb{R}^{M \times J},$$

where $\mathbf{V}^* = [\mathbf{v}_1^*, \ldots, \mathbf{v}_M^*]^\top \in \mathbb{R}^{M \times d}$. Here, the $j$-th column of $F_{\mathrm{CNN}}(\mathbf{X})$ corresponds to the output of $j$-th pooling group $g_j$, and $m$-th row of $F_{\mathrm{CNN}}(\mathbf{X})$ corresponds to the output of $m$-th kernel $\mathbf{v}_m^*$.

To formulate the convolution layer above as a teacher for transformers, we further specify the correspondence between each input patch and the output. The teacher model can then be given as $f^*(\mathbf{X}) = \sigma(\mathbf{V}^*\mathbf{X}\mathbf{S}^*)$, where the $i$-th column of $\mathbf{S}^*$ is $\mathbf{1}_{g_j}/K$, with $g_j$ being the group containing $i$.

**Example 2.2** (Single graph convolution layer on a regular graph). Let $\mathbf{A} \in \mathbb{R}^{D \times D}$ be an adjacency matrix of a degree-$(K-1)$ regular graph with $D$ nodes, and $\mathbf{X} = [\mathbf{x}_1, \mathbf{x}_2, \ldots, \mathbf{x}_D] \in \mathbb{R}^{d \times D}$ be the feature matrix of this graph, with each column $\mathbf{x}_i$ (for all $i$ in $[D]$) representing the $d$-dimensional feature vector of the $i$-th node. A typical single graph convolution layer (Kipf & Welling, 2017), with weight matrix $\mathbf{V}^* \in \mathbb{R}^{M \times d}$ is defined as

$$F_{\mathrm{GCN}}(\mathbf{X}) = \sigma(\mathbf{V}^*\mathbf{X}\widetilde{\mathbf{D}}^{-1/2}\widetilde{\mathbf{A}}\widetilde{\mathbf{D}}^{-1/2}), \tag{2.2}$$

where $\widetilde{\mathbf{A}} = \mathbf{A} + \mathbf{I}_D$ is the adjacency matrix with self-connections added, and $\widetilde{\mathbf{D}}$ is the diagonal degree matrix of $\widetilde{\mathbf{A}}$. For a degree-$(K-1)$ regular graph, each node has $K-1$ neighbors, and hence each column of $\widetilde{\mathbf{A}}$ contains $K$ ones and $D - K$ zeroes, and $\widetilde{\mathbf{D}} = K \cdot \mathbf{I}_D$. Therefore, the GCN defined in (2.2) is equivalent to a $f^*(\mathbf{X}) = \sigma(\mathbf{V}^*\mathbf{X}\mathbf{S}^*)$ with $\mathbf{V}^*$ and $\mathbf{S}^* = \widetilde{\mathbf{D}}^{-1/2}\widetilde{\mathbf{A}}\widetilde{\mathbf{D}}^{-1/2} = \widetilde{\mathbf{A}}/K$.

**Example 2.3** (Sparse token selection model (Sanford et al., 2023; Wang et al., 2024)). Let $\mathbf{X} = [\mathbf{x}_1, \mathbf{x}_2, \ldots, \mathbf{x}_D] \in \mathbb{R}^{d \times D}$ be a sequence of $d$-dimensional tokens. Given a $K$-element index set $g \subseteq [D]$, the goal of sparse token selection is to (i) select the tokens $\mathbf{x}_i$, $i \in g$, and (ii) take an average over the selected tokens. Hence, we can define

$$F_{\mathrm{STS}}(\mathbf{X}) = \frac{1}{K}\sum_{i \in g}\mathbf{x}_i.$$

Then it is clear that $f^*(\mathbf{X}) = \sigma(\mathbf{V}^*\mathbf{X}\mathbf{S}^*)$ with $\mathbf{V}^* = \mathbf{I}_D$, $\mathbf{S}^* = \frac{1}{K}\mathbf{1}_g \cdot \mathbf{1}_D^\top \in \mathbb{R}^{D \times D}$, and $\sigma(\cdot)$ being identity map is equivalent to $F_{\mathrm{STS}}(\mathbf{X})$, except that $f^*(\mathbf{X})$ duplicates the output $D$ times to match the output dimensions of a self-attention layer.

**Remark 2.4.** The "sparse token selection" task defined in Example 2.3 is slightly different from that studied in Wang et al. (2024). In our setting, the index set $g$ is specified as part of the learning objective and therefore remains fixed across all inputs. In contrast, Wang et al. (2024) considers a setting in which $g$ is provided as part of the input, allowing target positions to vary between different inputs. We remark that despite the difference, our learning task and that studied in Wang et al. (2024) essentially lead to very similar learning dynamics. We provide a detailed discussion in Appendix C.

**Example 2.5** (Group sparse linear predictors (Zhang et al., 2025c))**.** Let $\mathbf{X} = [\mathbf{x}_1, \mathbf{x}_2, \ldots, \mathbf{x}_D] \in \mathbb{R}^{d \times D}$ be a sequence of $d$-dimensional feature groups. For a given ground truth vector $\mathbf{v}^* \in \mathbb{R}^d$, and a label-relevant group index $i^*$, the group sparse linear predictor will first search for the variable group $\mathbf{x}_i$ corresponding to the label-relevant index $i^*$, and then calculate its inner product with the ground truth vector $\mathbf{v}^*$. Hence, we define

$$F_{\text{GSLP}} = \langle \mathbf{v}^*, \mathbf{x}_{i^*} \rangle.$$

Consider a teacher model $f^*(\mathbf{X}) = \sigma(\mathbf{V}^*\mathbf{X}\mathbf{S}^*)$ with $\mathbf{V}^* = \mathbf{v}^*$ by reducing $M$ to 1, $\mathbf{S}^* = \mathbf{e}_{i^*} \cdot \mathbf{1}_D^\top$, and $\sigma(\cdot)$ being identity map. Then similar to Example 2.3, $f^*(\mathbf{X})$ duplicates the output of $F_{\text{GSLP}}(\mathbf{X})$ for $D$ times, and is essentially equivalent to $F_{\text{GSLP}}(\mathbf{X})$.

**One-layer transformer.** A one-layer transformer model Vaswani et al. (2017); Dosovitskiy et al. (2020) can be defined as

$$\text{TF}(\mathbf{Z}; \mathbf{W}_V; \mathbf{W}_Q; \mathbf{W}_K) = \sigma\left( \mathbf{W}_V \mathbf{Z} \mathcal{S}\left( \frac{\mathbf{Z}^\top \mathbf{W}_K^\top \mathbf{W}_Q \mathbf{Z}}{\sqrt{D}} \right) \right). \tag{2.3}$$

In this formulation, $\mathbf{Z}$ represents the input matrix of the transformers, obtained by concatenating the original feature matrix $\mathbf{X}$ with its positional encoding matrix $\mathbf{P}$. Specifically, for each column $\mathbf{x}_i$ (for all $i \in [D]$) of the original feature matrix $\mathbf{X}$, we concatenate it with the position encoding vector $\mathbf{p}_i$, which contains the positional information of this specific index, to generate a column of $\mathbf{Z}$ as $\mathbf{z}_i = [\mathbf{x}_i^\top, \mathbf{p}_i^\top]^\top$. The complete positional encoding matrix is denoted as $\mathbf{P} = [\mathbf{p}_1, \mathbf{p}_2, \ldots, \mathbf{p}_D]$, and we employ an orthogonal design for $\mathbf{P}$, meaning that $\mathbf{P}$ is an $D \times D$ orthogonal matrix. For analytical convenience, the practice of concatenating feature and positional encoding matrices has been widely adopted in recent theoretical studies (Nichani et al., 2024; Bai et al., 2024; Wang et al., 2024; Zhang et al., 2025c). Furthermore, $\mathcal{S}(\cdot) : \mathbb{R}^{D \times D} \mapsto \mathbb{R}^{D \times D}$ denotes the softmax operator, which implements the softmax function column-wisely, and $\mathbf{W}_V$, $\mathbf{W}_Q$, $\mathbf{W}_K$ represent the value matrix, query matrix, and key matrix in a typical self-attention structure, respectively. Instead of studying the typical structure (2.3), we consider a moderately simplified "position-only" softmax self-attention in this paper, which is defined as

$$\text{TF}(\mathbf{Z}; \mathbf{W}_V; \mathbf{W}_{KQ}) = \sigma\left( \mathbf{W}_V \mathbf{X} \mathcal{S}\left( \frac{\mathbf{P}^\top \mathbf{W}_{KQ} \mathbf{P}}{\sqrt{D}} \right) \right) = \sigma(\mathbf{W}_V \mathbf{X} \mathbf{S}) \in \mathbb{R}^{M \times D}. \tag{2.4}$$

In comparison with the typical single-head self-attention architecture (2.3), our model (2.4) is simplified from the following two aspects: (i). We re-parameterize the original key matrix $\mathbf{W}_K$ and query matrix $\mathbf{W}_Q$ into one trainable key-query matrix $\mathbf{W}_{KQ}$, which has been adopted in almost theoretical studies regarding the optimization of transformers (Tian et al., 2023; Zhang et al., 2024b; Wang et al., 2024; Huang et al., 2024; Frei & Vardi, 2025; Zhang et al., 2025c; He et al., 2025a). (ii). We employ an architecture such that only the positional encoding matrix $\mathbf{P}$ is involved when calculating the softmax attention score, and the value matrix $\mathbf{W}_V$ only interacts with the feature matrix $\mathbf{X}$. To illustrate a rationale for this design, consider the following one-layer transformers:

$$\widetilde{\text{TF}}(\mathbf{Z}; \widetilde{\mathbf{W}}_V; \widetilde{\mathbf{W}}_{KQ}) = \sigma\left( \widetilde{\mathbf{W}}_V \mathbf{Z} \mathcal{S}\left( \frac{\mathbf{Z}^\top \widetilde{\mathbf{W}}_{KQ} \mathbf{Z}}{\sqrt{D}} \right) \right), \tag{2.5}$$

where the entire input matrix $\mathbf{Z}$ is involved in both the calculation of attention score and interactions with the value matrix. Empirical observations (illustrated in Figure 1) reveal that when the transformer model $\widetilde{\text{TF}}$ in (2.5) is used to learn a teacher model $f^*$ in (2.1), substantial training predominantly occurs in the left block of $\widetilde{\mathbf{W}}_V$ and the 'bottom-right' block of $\widetilde{\mathbf{W}}_{KQ}$. These actively trained blocks map to $\mathbf{W}_V$ and $\mathbf{W}_{KQ}$ respectively in our model (2.4), while other parameter blocks of $\widetilde{\text{TF}}$ exhibit negligible changes from their initial values. Consequently, our model (2.4) can be considered essentially equivalent to the transformer model $\widetilde{\text{TF}}$ if these rarely updated blocks within $\widetilde{\mathbf{W}}_V$ and $\widetilde{\mathbf{W}}_{KQ}$ are fixed to zero. This strategy of fixing certain transformer parameters during training is widely adopted in the theoretical studies on the optimization of transformers (Wu et al., 2023; Tarzanagh et al., 2023a; Huang et al., 2024; Sakamoto & Sato, 2024; Frei & Vardi, 2025; He et al., 2025a), and analogous "position-only" attention structures are also adopted in Jelassi et al. (2022); Wang et al. (2024).

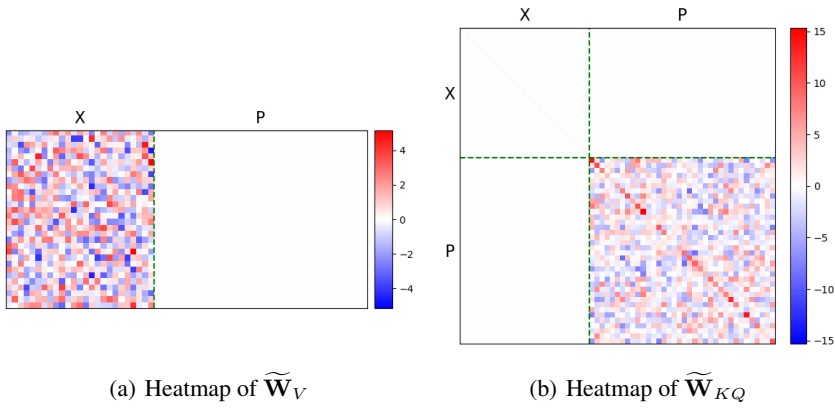

(a) Heatmap of $\widetilde{\mathbf{W}}_V$          (b) Heatmap of $\widetilde{\mathbf{W}}_{KQ}$

Figure 1: Visualization of parameter matrices for the transformer $\widetilde{\text{TF}}$ in (2.5), obtained after training to learn the teacher model $f^*$ and achieving loss convergence. The formal illustration of the loss function and training algorithm is provided in the next section.

## 3   MAIN RESULTS

In this section, we demonstrate our theoretical conclusions of utilizing a one-layer transformer (2.4) to learn a given teacher model $f^*$ in (2.1). For a teacher model $f^*$ parameterized with the ground truth value matrix $\mathbf{V}^*$ and ground truth softmax scores $\mathbf{S}^*$, the observed label $\mathbf{Y}$ for an input matrix $\mathbf{X}$ is assumed to be generated as:

$$\mathbf{Y} = f^*(\mathbf{X}) + \mathcal{E} = \sigma(\mathbf{V}^*\mathbf{X}\mathbf{S}^*) + \mathcal{E} \in \mathbb{R}^{M \times D}, \tag{3.1}$$

where $\mathcal{E} \in \mathbb{R}^{M \times D}$ is a noise matrix independent of $\mathbf{X}$ and following a zero-mean distribution. To train a one-layer transformer (2.4), we consider the population mean squared error as the objective loss function. Specifically, given an input-label pair $(\mathbf{X}, \mathbf{Y})$, the loss function is defined as

$$\mathcal{L}(\mathbf{W}_V; \mathbf{W}_{KQ}) = \frac{1}{2}\mathbb{E}_{\mathbf{X},\mathbf{Y}}\big[\|\mathbf{Y} - \text{TF}(\mathbf{Z}; \mathbf{W}_V; \mathbf{W}_{KQ})\|_F^2\big]. \tag{3.2}$$

Here, each column of $\mathbf{X}$ is assumed to independently follow the standard Gaussian distribution during the training stage of (2.4), i.e. $\mathbf{x}_i \overset{\text{i.i.d}}{\sim} \mathcal{N}(0, \mathbf{I}_d)$ for all $i \in [D]$. Due to the variance introduced by the noise component $\mathcal{E}$, even the loss of the ground truth model $f^*$ has an irreducible term, and we denote this term as the optimal loss, i.e.

$$\mathcal{L}_{\mathbf{opt}} = \frac{1}{2}\mathbb{E}_{\mathbf{X},\mathbf{Y}}\big[\|\mathbf{Y} - f^*(\mathbf{X})\|_F^2\big] = \frac{1}{2}\mathbb{E}\big[\|\mathcal{E}\|_F^2\big].$$

To evaluate the performance of one-layer transformer with different $\mathbf{W}_V$ and $\mathbf{W}_{KQ}$, we consider the excess loss defined as: $\mathcal{L}(\mathbf{W}_V; \mathbf{W}_{KQ}) - \mathcal{L}_{\mathbf{opt}}$. While the choice population loss implicitly suggests an infinite training data set—a scenario not feasible in practice—it significantly simplifies the technical challenges of conducting a rigorous optimization analysis for transformer models. This approach enables us to focus on the global optimization trajectories, and has been adopted in most of the recent theoretical studies regarding the optimization of transformer models (Zhang et al., 2024b; Huang et al., 2024; Wang et al., 2024; Jelassi et al., 2022; Frei & Vardi, 2025; Zhang et al., 2025c).

For the training objective loss (3.2), we utilize the gradient descent to derive the optimal solutions for the value matrix $\mathbf{W}_V$, and key-query matrix $\mathbf{W}_{KQ}$. The iterative rule for $\mathbf{W}_V$ and $\mathbf{W}_{KQ}$ during the learning process can be expressed as

$$\mathbf{W}_V^{(t+1)} = \mathbf{W}_V^{(t)} - \eta\nabla_{\mathbf{W}_V}\mathcal{L}(\mathbf{W}_V^{(t)}; \mathbf{W}_{KQ}^{(t)}); \tag{3.3}$$

$$\mathbf{W}_{KQ}^{(t+1)} = \mathbf{W}_{KQ}^{(t)} - \eta\nabla_{\mathbf{W}_{KQ}}\mathcal{L}(\mathbf{W}_V^{(t)}; \mathbf{W}_{KQ}^{(t)}), \tag{3.4}$$

where $\eta$ is the learning rate, and the initializations are set as $\mathbf{W}_V^{(0)}, \mathbf{W}_{KQ}^{(0)} = \mathbf{0}$. Based on these preliminaries, the following theorem characterizes the convergence of gradient descent (3.3) and (3.4).

**Theorem 3.1.** Suppose that $D \geq \Omega\big(\mathrm{poly}(M, K)\big)$, $\eta \leq \mathcal{O}(M^{-1}D^{-5/2})$. Under these conditions, there exists $T^* = \Theta\Big(\frac{KD^2}{\eta\|\mathbf{V}^*\|_F^2}\Big)$, such that for all $T \geq T^*$, the following results hold.

1. The attention scores achieved by the one-layer transformer (2.4), match the ground truth softmax scores of the teacher model: $\mathbf{S}^{(T)}$ at the $T$-th iteration satisfies that

$$\big\|\mathbf{S}^{(T)} - \mathbf{S}^*\big\|_F = \Theta\left(\frac{D^{\frac{5}{2}}}{\|\mathbf{V}^*\|_F \sqrt{\eta T}}\right).$$

2. The value matrix $\mathbf{W}_V$ of the one-layer transformer (2.4) aligns with the ground truth value matrix of the teacher model:

$$\big\|\mathbf{W}_V^{(T)} - \mathbf{V}^*\big\|_F = \Theta\left(D^2\sqrt{\frac{K}{\eta T}}\right).$$

3. The excess loss is minimized with matching lower and upper bounds:

$$\frac{\underline{c}KD^4}{\eta T} \leq \mathcal{L}\Big(\mathbf{W}_V^{(T)}; \mathbf{W}_{KQ}^{(T)}\Big) - \mathcal{L}_{\mathbf{opt}} \leq \frac{\bar{c}KD^4}{\eta T},$$

where $\underline{c}$ and $\bar{c}$ are two positive constants satisfying $\underline{c} \leq \bar{c}$.

The proof of Theorem 3.1 is given in Appendix D. Theorem 3.1 demonstrates that a one-layer transformer can learn the teacher model $f^*$ formulated in (2.1) from two aspects. The first and second results show that the one-layer transformer's value matrix $\mathbf{W}_V^{(T)}$ and attention scores $\mathbf{S}^{(T)}$ converge (in the Frobenius norm) to the teacher model's ground truth value matrix $\mathbf{V}^*$ and softmax scores $\mathbf{S}^*$, respectively. This reveals that a one-layer transformer trained via gradient descent can correctly recover the teacher model by accurately learning all its core components. The third result in Theorem 3.1 shows that the training loss will eventually converge to the optimal loss at a rate of $\Theta\big(\frac{KD^4}{\eta T}\big)$. The third result characterizes the convergence of the training loss. It shows that the excess loss decreases at the rate $\Theta\big(\frac{KD^4}{\eta T}\big)$, with matching upper and lower bounds. We note that the factor $D^4$ indicates that the convergence takes a large number of iterations when the sequence length $D$ is large. However, the matching lower bound in Theorem 3.1 confirms that this rate is already optimal and cannot be improved under our current setting. In fact, this polynomial dependence on $D$ originates from two intrinsic aspects of the learning task: (i) Since the loss is the squared Frobenius distance between two $M \times D$ matrices, it necessarily aggregates errors over all $D$ columns, and thus scales proportionally with the sequence length; (ii) The $1/\sqrt{D}$ factor appears in the gradients of $\mathbf{W}_{KQ}$ and requires $\mathbf{W}_{KQ}$ to scale larger to achieve sufficient convergence, thereby introducing additional factors of $D$ into the convergence rate.

As illustrated in Examples 2.3 and 2.5, our teacher model $f^*$ encompasses settings that are closely related to the learning tasks studied in Wang et al. (2024) and Zhang et al. (2025c). For the "sparse token selection" problem, Theorem 3.1 establishes a learning guarantee for the setting in which the target index set is fixed by the learning objective and not provided as part of the input. This offers a complementary perspective to the settings in Wang et al. (2024), where the target index set is given as a part of input, and may vary across different data points. Under our setting, Theorem 3.1 yields a tight $\Theta\big(\frac{1}{T}\big)$ convergence rate with matching upper and lower bounds, sharper than the $\mathcal{O}\big(\frac{\log(T)}{T}\big)$ guarantee obtained under the different problem formulation of Wang et al. (2024) A detailed comparison between the convergence rate is provided in Appendix C. Regarding group-sparse linear prediction, Zhang et al. (2025c) focus primarily on the classification setting, while Theorem 3.1 delivers a complementary result by addressing the regression setting.

The learning guarantee in Theorem 3.1 is established under the assumption that the data input matrix $\mathbf{X}$ is Gaussian, and the target response matrix $\mathbf{Y}$ is provided by the teacher with noises. Here, we can also study the out-of-distribution (OOD) generalization guarantee of the obtained transformer model on data without such assumptions. Specifically, we consider any feature and response matrices $\widetilde{\mathbf{X}} \in \mathbb{R}^{d \times D}$, $\widetilde{\mathbf{Y}} \in \mathbb{R}^{M \times D}$ with bounded second moments, and establish bounds on the OOD loss

$$\mathcal{L}_{\mathbf{OOD}}(\mathbf{W}_V; \mathbf{W}_{KQ}) = \frac{1}{2}\mathbb{E}_{\widetilde{\mathbf{X}}, \widetilde{\mathbf{Y}}}\big[\|\widetilde{\mathbf{Y}} - \mathrm{TF}(\widetilde{\mathbf{Z}}; \mathbf{W}_V; \mathbf{W}_{KQ})\|_F^2\big]$$

by comparing it with the loss achieved by the teacher model. We have the following theorem.

**Theorem 3.2.** Suppose that $D \geq \Omega\big(\text{poly}(M, K)\big)$ and $\eta \leq \mathcal{O}(M^{-1}D^{-5/2})$. In addition, the OOD input pairs $(\widetilde{\mathbf{X}}, \widetilde{\mathbf{Y}})$ satisfy the condition that each column $\widetilde{\mathbf{x}}_i$ and $\widetilde{\mathbf{y}}_i$ has finite second moments, i.e. there exists a constant $\xi > 0$ such that $\mathbb{E}[\|\widetilde{\mathbf{x}}_i\|_2^2], \mathbb{E}[\|\widetilde{\mathbf{y}}_i\|_2^2] \leq \xi$ for all $i \in [D]$. Then for any $\epsilon > 0$, there exists $T_\epsilon = \mathcal{O}\big(\frac{KD^6\xi^2 \sum_{m=1}^M \|\mathbf{v}_m^*\|_2^2}{\eta\epsilon^2}\big)$ such that for any $T > T_\epsilon$, the OOD loss satisfies that:

$$\mathcal{L}_{\mathbf{OOD}}\Big(\mathbf{W}_V^{(T)}; \mathbf{W}_{KQ}^{(T)}\Big) \leq \frac{1}{2}\mathbb{E}\big[\|\widetilde{\mathbf{Y}} - f^*(\widetilde{\mathbf{X}})\|_F^2\big] + \epsilon.$$

Theorem 3.2 requires only the mild assumption that $\widetilde{\mathbf{X}}$ and $\widetilde{\mathbf{Y}}$ have bounded second moments. Notably, the response matrix $\widetilde{\mathbf{Y}}$ need not be generated by or correlated with the output of the teacher model $f^*(\widetilde{\mathbf{X}})$. Therefore, the term $\frac{1}{2}\mathbb{E}\big[\|\widetilde{\mathbf{Y}} - f^*(\widetilde{\mathbf{X}})\|_F^2\big]$ measures the teacher model's O.O.D. test loss, analogous to the role of $\mathcal{L}_{\text{opt}}$ in Theorem 3.1. This shows that the trained transformer's O.O.D. loss exceeds that of the teacher model by at most $\epsilon$, demonstrating its robustness to distribution shift. In addition, although it is challenging to establishing a matching lower bound for all pairs $(\widetilde{\mathbf{X}}, \widetilde{\mathbf{Y}})$ like Theorem 3.1, a worst-case $\widetilde{\mathbf{Y}}$ can be constructed to demonstrate that this upper bound is attainable, thereby validating the tightness of Theorem 3.2. The complete proof of Theorem 3.2 and the worst-case example are provided in Section E.

## 4 EXPERIMENTS

In this section, we present our experimental results. As detailed in Section 2, the teacher model can cover various models, including (i). convolution layer with average pooling, (ii). graph convolution layer on a regular graph, (iii). sparse token selection model, and (iv). group sparse linear predictor. Our experiments also focus on these four cases.

We conduct experiments on both synthetic data and real-world data sets, respectively. For experiments on synthetic data, we follow the exact definitions in Section 2 to build up teacher models $f^*$. For experiments on real-world datasets, we pre-train a teacher CNN on the MNIST dataset, whose first convolution layer is then served as the teacher model to train the student transformer.

### 4.1 SYNTHETIC DATA EXPERIMENTS

We begin by detailing the common experimental setups on synthetic data. Given parameters $d$ and $D$, an fixed orthogonal matrix $\mathbf{P} \in \mathbb{R}^{D \times D}$ serves as the positional encoding matrix We adopt an online gradient descent algorithm to simulate training over the population loss. At each iteration, we sample a new batch of $N = 100$ standard $d \times D$ Gaussian matrices, i.e. $\{\mathbf{X}_n\}_{n=1}^N \subseteq \mathbb{R}^{d \times D}$. For each $\mathbf{X}_n$ with $n \in [N]$, its corresponding label $\mathbf{Y}_n = f^*(\mathbf{X}_n) + \mathcal{E}_n$, where $\mathcal{E}_n \in \mathbb{R}^{M \times D}$ is another independently sampled Gaussian matrix. We concatenate each $\mathbf{X}_n$ with the fixed positional encoding matrix $\mathbf{P}$ to form $\mathbf{Z}_n$ as the inputs to the transformer Subsequently, a gradient descent update is performed using this batch of $N = 100$ data pairs $\{(\mathbf{Z}_n, \mathbf{Y}_n)\}_{n=1}^N$. Furthermore, we also generate another batch of $N = 100$ data pairs $\{(\widetilde{\mathbf{Z}}_n, \widetilde{\mathbf{Y}}_n)\}_{n=1}^N$ following the almost identical procedure, except that each $\widetilde{\mathbf{X}}_n$ is generated from the exponential distribution. This batch of data pairs $\{(\widetilde{\mathbf{Z}}_n, \widetilde{\mathbf{Y}}_n)\}_{n=1}^N$ is prepared for calculating the excess OOD loss, defined as $\mathcal{L}_{\text{OOD}} - \frac{1}{2N}\sum_{n=1}^N \|\widetilde{\mathbf{Y}}_n - f^*(\widetilde{\mathbf{X}}_n)\|_F^2$.

In the next, we introduce the distinct settings for different tasks, specifically the ground-truth softmax score matrices $\mathbf{S}^*$. For the task of learning a convolution layer with average pooling, we set $D = 36$ and $K = 4$, where the pooling groups are partitioned by aggregating the $K$ neighbor patches into a group. Given this partition of pooling groups, the ground truth softmax score of the teacher model can be formulated into a diagonal block matrix as $\mathbf{S}^* = \frac{1}{K}\text{Diag}(\mathbf{1}_{K \times K}, \ldots, \mathbf{1}_{K \times K})$, with totally $D/K$ blocks. For the task of learning a graph convolution layer, we consider a 'cycle-graph' with $D = 20$ nodes, where each node is connected to exactly two other nodes, i.e. the $i$-th node is connected to its adjacent nodes $(i-1)$ and $(i+1)$. Under this setup, the ground-truth softmax score $\mathbf{S}^*$ is constructed as follows: for each column $i$, the entries at rows $(i-1), i$, and $(i+1)$ are set to $1/K$ with $K = 3$, while all other entries are zero. For both the tasks of learning the sparse token selection model and the group sparse linear predictor, we set the total number of

tokens/feature groups $D = 20$, and randomly generate $K$ indices from $[D]$ as indices of target tokens/ label-relevant group, where $K = 4$ and 1 respectively. In these two sets of tasks, the rows representing the target tokens/ label-relevant group equal to $1/K$, while other rows are filled with 0.

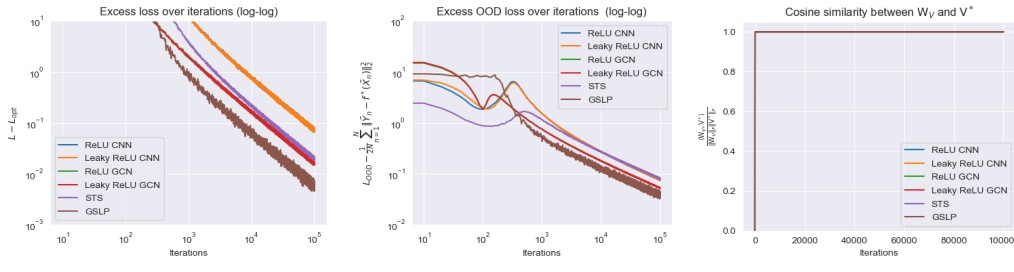

(a) Excess training loss (log-log)   (b) Excess OOD test loss (log-log)   (c) Cosine similarity

Figure 2: Excess training loss, excess OOD test loss (both in log-log scales), and cosine similarity between the value matrix $\mathbf{W}_V$ of one layer transformer (2.4), and ground truth value matrix $\mathbf{V}^*$. These results are presented for six experimental sets, which originate from four distinct tasks.

For the task of learning CNN and GCN, we conduct two sets for each with ReLU and Leaky ReLU respectively. Experiment results are given in Figures 2 and 3. Figure 2(a) and Figure 2(b) demonstrate the convergence curves for the excess training loss and the excess OOD test loss (both in log-log scales). We can clearly observe that both the excess training loss and the OOD test loss converge to a small value on all six sets of experiments. After initial iterations, the curves for excess training loss appear almost straight with slopes equal to $-1$, and excess OOD loss curves have approximate $-0.5$ slopes. These observations validate the $\Theta(1/T)$ convergence rate in Theorem 3.1, and $\mathcal{O}(1/\sqrt{T})$ convergence rate in Theorem 3.2. Figure 2(c) displays the cosine similarity curve between the value matrix $\mathbf{W}_V^{(t)}$, and the ground truth value matrix $\mathbf{V}^*$. It shows that $\mathbf{W}_V^{(t)}$ directionally aligns with the ground truth value matrix $\mathbf{V}^*$ in all six experiments since the very beginning.

Furthermore, Figure 3 provides the heatmaps of the attention scores when the loss converges. Specifically, Figure 3(a) and Figure 3(b) respectively display the attention scores when learning a convolution layer with ReLU and Leaky ReLU. In both figures, the attention scores exhibit a diagonal block matrix pattern, where each diagonal block has approximately equal values $1/4$. Figure 3(c) and Figure 3(d) show the attention scores when learning a graph convolution layer on a cycle graph. Specifically, the attention scores show a pattern of a cyclic tridiagonal matrix, with all the significant entries having approximately equal values $1/3$. Figure 3(e) and Figure 3(f) show the attention scores when learning a sparse token selection task and group sparse linear predictor. We can observe that only the rows corresponding to the target positions are assigned significant values in both tasks. In summary, all these patterns match the ground truth softmax scores, which are described previously.

## 4.2 REAL DATA EXPERIMENTS

We also conduct experiments on the MNIST dataset. Each image is normalized and resized to $27 \times 27$ pixels. We train a two-layer CNN with $M = 16$ convolution kernels, each having a $3 \times 3$ kernel size. Given the $27 \times 27$ image dimensions, each image is divided into $D = 81$ patches. An average pooling layer with a $3 \times 3$ pooling receptive field (i.e $K = 9$) is additive to the first convolution layer, and then cascaded with activation and a linear layer for classification. This two-layer CNN is trained by minimizing the cross-entropy loss, achieving a moderate test accuracy of about $71\%$ on the test set after 20 epochs. After training of this teacher CNN, its first convolution layer with average pooling is extracted as the teacher model $f^*$, with its hidden-layer outputs supervising a one-layer transformer (2.4). The training of the one-layer transformer is still conducted on the MNIST dataset, and the mean-squared loss is employed for optimization.

The experiment results are given in Figure 4 and Figure 5. Figure 4(a) displays the training loss curves. We can observe that for both ReLU and Leaky ReLU, the training loss very quickly converges to a small value. Figure 4(b) demonstrates the cosine similarity curve between the value matrix $\mathbf{W}_V^{(t)}$ of the transformer and the convolution kernel matrix $\mathbf{V}^*$ of the teacher convolution layer. The similarity rises above 0.9, indicating that the transformer successfully learns the ground-

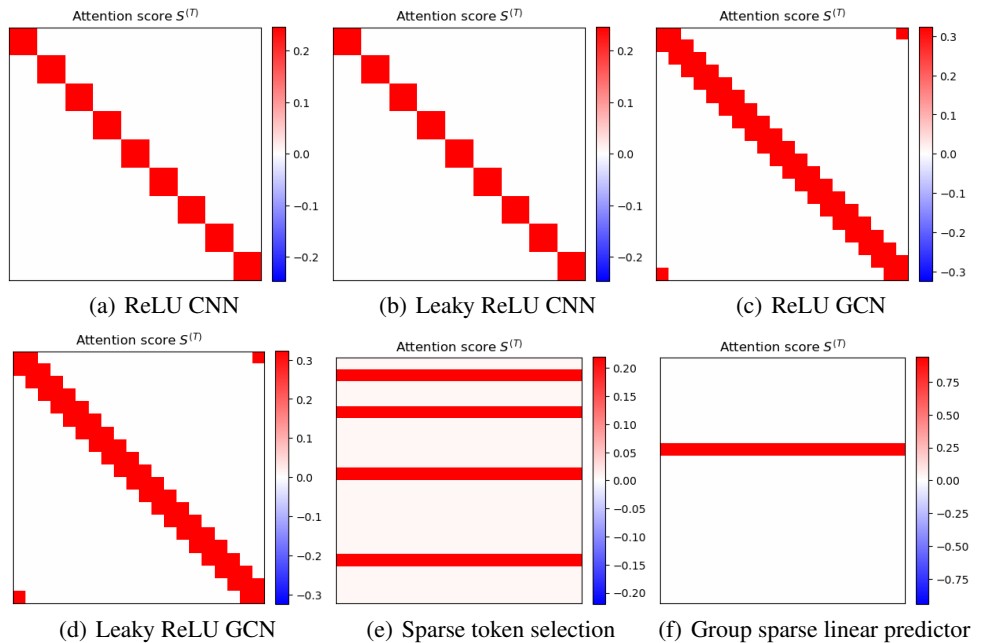

Figure 3: Heatmap of attention score matrix $\mathbf{S}^{(T)}$ when the training loss converges. The results are presented for six different experimental sets, indicated by the captions of sub-figures.

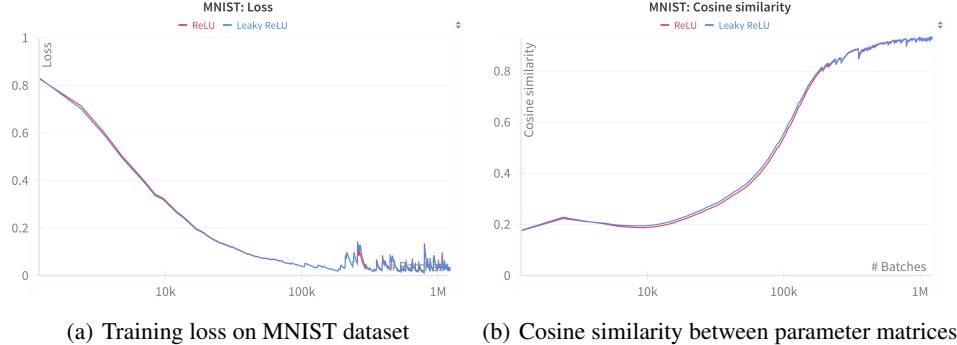

(a) Training loss on MNIST dataset      (b) Cosine similarity between parameter matrices

Figure 4: Training loss and cosine similarity between the value matrix $\mathbf{W}_V$ of the one-layer transformer (2.4), and convolution kernel matrix $\mathbf{V}^*$ of the pre-trained teacher CNN.

truth value matrix of the teacher model. Furthermore, Figure 5(a) provides the heatmap of the ground truth softmax score derived from the teacher CNN's average pooling layer. Figure 5(b) and Figure 5(c) respectively present heatmaps of attention scores at convergence for the transformers with ReLU and Leaky ReLU activations. We can observe that both the attention scores achieved by transformers can capture the pattern of the ground truth softmax scores, with notable exceptions in the first and last nine rows in the softmax heatmap. We remark that the failure in learning these rows of ground-truth softmax scores is due to the fact that they correspond to MNIST image patches that are mostly all background (all zero). Figure 5(d) highlights the image regions corresponding to failed-to-learn softmax scores, marked by yellow rectangles. We can see that they are indeed boundary regions and are mostly pure background. Consequently, they offer minimal informative content to the model, explaining why transformers can not attend to these positions. Overall, it is clear that the real-world data experiments corroborate our theory.

## 5   PROOF SKETCH OF THEOREM 3.1

In this section, we outline the major steps in the proof of Theorem 3.1. For simplicity, here we focus the case where $\sigma(\cdot)$ is the identity map. More details, including more general choices of $\sigma(\cdot)$, are formally proved in Appendix D. The proof consists of three main steps:

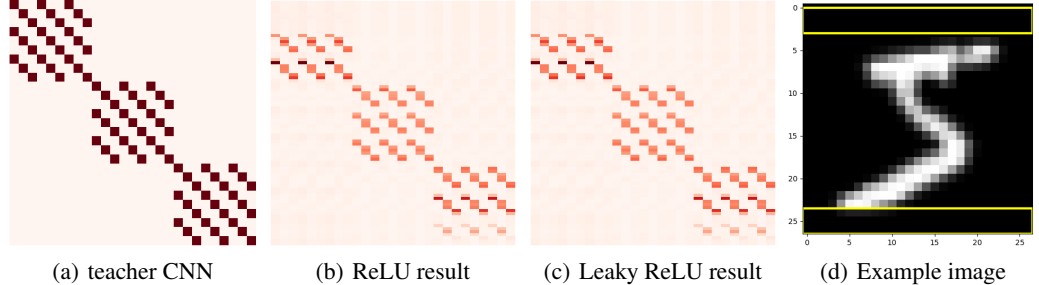

(a) teacher CNN      (b) ReLU result      (c) Leaky ReLU result      (d) Example image

Figure 5: Heatmap of the ground truth softmax scores of average pooling, Heatmap of the attention scores $\mathbf{S}^{(T)}$ of trained one-layer transformer when loss converges, and an image example in MNIST.

**Step 1. Structures of $\mathbf{W}_V$ and $\mathbf{W}_{KQ}$ during training.** A critical step in our proof is to show that throughout training, the parameter matrices $\mathbf{W}_V$ and $\mathbf{W}_{KQ}$ preserve the following decompositions:

$$\mathbf{W}_V^{(t)} = C_1(t)\mathbf{V}^*; \; \mathbf{W}_{KQ}^{(t)} = C_2(t)\sum_{i=1}^{D}\sum_{i' \in G^i}\mathbf{p}_{i'}\mathbf{p}_i^\top - C_3(t)\sum_{i=1}^{D}\sum_{i' \notin G^i}\mathbf{p}_{i'}\mathbf{p}_i^\top,$$

where $G^i$ denotes the index set of entries of value $1/K$ in $i$-th column of $\mathbf{S}^*$. The details of this conclusion are given in Lemma D.2. Based on the decompositions, we can express $\mathbf{S}^{(t)}$ as: $\mathbf{S}^{(t)}_{i',i} = \frac{1}{K+(D-K)\exp(-(C_2(t)+C_3(t))/\sqrt{D})}$ if $i' \in G^i$; $\mathbf{S}^{(t)}_{i',i} = \frac{\exp(-(C_2(t)+C_3(t))/\sqrt{D})}{K+(D-K)\exp(-(C_2(t)+C_3(t))/\sqrt{D})}$ if $i' \notin G^i$. Comparing these results with the definition of the teacher model $f^*(\cdot)$, we can further observe that

$$\mathbf{W}_V^{(t)} \to \mathbf{V}^* \Leftrightarrow C_1(t) \to 1; \qquad \mathbf{S}^{(t)} \to \mathbf{S}^* \Leftrightarrow C_2(t) + C_3(t) \to \infty.$$

In this way, the original optimization analysis regarding full matrices $\mathbf{W}_V$ and $\mathbf{W}_{KQ}$ is simplified into studying the updates of three scalars $C_1(t), C_2(t), C_3(t)$.

**Step 2. Accurate characterization of convergence that $C_1(t) \to 1$ and $C_2(t) + C_3(t) \to \infty$.** The decompositions obtained in **Step 1.** implies that the coefficients $C_1(t), C_2(t), C_3(t)$ essentially follow gradient descent starting from zero initialization minimizing the loss

$$\widetilde{\mathcal{L}}(C_1, C_2, C_3) \propto \frac{D-K}{K}\left[1 - \frac{KC_1}{K + (D-K)e^{-\frac{C_2+C_3}{\sqrt{D}}}}\right]^2 + C_1^2\left[1 - \frac{K}{K + (D-K)e^{-\frac{C_2+C_3}{\sqrt{D}}}}\right]^2,$$

We remark that this expression of $\widetilde{\mathcal{L}}(C_1, C_2, C_3)$ corresponds to the special case where $\sigma(\cdot)$ is the identity map. The general formulation for $\sigma(\cdot)$ is activation is deferred to Lemma D.2. Then by carefully analyzing the training dynamics, we can show that for sufficiently large $T$,

$$C_1(T) - 1 = \Theta\left(\frac{D^2\sqrt{K}}{\|\mathbf{V}^*\|_F\sqrt{\eta T}}\right), \quad C_2(T) + C_3(T) = \frac{\sqrt{D}}{2}\log\left(\Theta\left(\frac{\eta\|\mathbf{V}^*\|_F^2}{K^3D^2}\right)T + e^{\frac{2}{K\sqrt{D}}}\right).$$

The details are provided in Lemmas D.2, D.5, D.15, D.18, and F.12.

**Step 3. Final convergence results.** Combining the convergence rates obtained in **Step 2.** and the formulations of $\mathbf{S}^{(T)}$ and $\mathbf{W}_V^{(T)}$ in **Step 1.**, we can further obtain that $\|\mathbf{S}^{(T)} - \mathbf{S}^*\|_F, \|\mathbf{W}_V^{(T)} - \mathbf{V}^*\|_F = \Theta\left(\frac{1}{\sqrt{T}}\right)$. Under mean-squared loss, the $\Theta\left(\frac{1}{\sqrt{T}}\right)$ convergence of the matrices $\mathbf{S}^{(t)}$ and $\mathbf{V}^{(t)}$ directly suggests that loss will decay at the rate of $\Theta\left(\frac{1}{T}\right)$, which finishes the proof.

## 6   CONCLUSIONS AND LIMITATIONS

In this paper, we provide the theoretical guarantee that a one-layer transformer can learn a class of teacher models, covering a wide range of common models in machine learning. Specifically, we establish a tight convergence bound at the rate of $\Theta\left(\frac{1}{T}\right)$ for the population loss. We also establish out-of-distribution generalization bounds for the obtained transformer model, demonstrating its robustness. To empirically support our findings, we conduct experiments on both synthetic data and real data, and all results align with our theoretical conclusion. Our current theory focuses on one-layer models, and we make certain simplifications and assumptions on the model and data, which present a limitation. We believe establishing teacher-student learning guarantees for more complex models and under milder assumptions is an interesting and promising further work direction.

## ACKNOWLEDGMENTS

We would like to thank the anonymous reviewers and area chairs for their helpful comments. Yuan Cao is supported in part by NSFC 12301657, Hong Kong RGC ECS 27308624, and Hong Kong RGC GRF 17301825.

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

# A    NOTATION

In this section, we introduce the key notations we use throughout paper. We first introduce the following mathematical notations.

**Mathematical notations.** Given two sequences $\{x_n\}$ and $\{y_n\}$, we denote $x_n = \mathcal{O}(y_n)$ if there exist some absolute constant $C_1 > 0$ and $N > 0$ such that $|x_n| \le C_1 |y_n|$ for all $n \ge N$. Similarly, we denote $x_n = \Omega(y_n)$ if there exist $C_2 > 0$ and $N > 0$ such that $|x_n| \ge C_2 |y_n|$ for all $n > N$. We say $x_n = \Theta(y_n)$ if $x_n = \mathcal{O}(y_n)$ and $x_n = \Omega(y_n)$ both holds. We use $\widetilde{\mathcal{O}}(\cdot)$, $\widetilde{\Omega}(\cdot)$, and $\widetilde{\Theta}(\cdot)$ to hide logarithmic factors in these notations respectively. Moreover, we denote $x_n = \text{poly}(y_n)$ if $x_n = O(y_n^D)$ for some positive constant $D$, and $x_n = \text{polylog}(y_n)$ if $x_n = \text{poly}(\log(y_n))$. For two scalars $a$ and $b$, we denote $a \vee b = \max\{a, b\}$ and $a \wedge b = \min\{a, b\}$. For any $n \in \mathbb{N}_+$, we use $[n]$ to denote the set $\{1, 2, \cdots, n\}$. In addition, we use $\mathbf{1}_n$ to denote a $n$-dimensional vector with all 1 entries. For an index set $g$, $\mathbf{1}_g$ denotes a vector whose entries are 1 for indices in $g$, and 0 otherwise. Let $\mathbf{A}_1, \ldots, \mathbf{A}_n$ be $n$ matrices with the same dimensionality $d_1 \times d_2$, then $\text{Diag}(\mathbf{A}_1, \ldots, \mathbf{A}_n)$ is a $nd_1 \times nd_2$ diagonal block matrix, with $\mathbf{A}_1, \ldots, \mathbf{A}_n$ being the block entries.

In addition, we also provide a summary table of the key variables in our study in Table 1.

Table 1: Key variables and their meanings.

| Symbol | Meaning |
|---|---|
| $\mathbf{V}^*$ | Ground truth value matrix in $f^*$, a $M \times D$ matrix. |
| $\mathbf{S}^*$ | Ground truth softmax score matrix in $f^*$, a $D \times D$ column-stochastic matrix. |
| $D$ | Sequence length (number of input tokens). |
| $d$ | Feature dimension of each token. |
| $K$ | Number of none zero entries in each column of $\mathbf{S}^*$. It can represent: |
| | (i) the pooling size in CNN and pooling layer, |
| | (ii) the number of neighbors of GCN layer, |
| | (iii) the number of target tokens in sparse token selection, |
| | (iv) it equals to 1 in group-sparse linear models. |
| $G^i$ | Target index set of $i$-th input token, namely $\mathbf{S}_{i',i} = \frac{1}{K}$ if $i' \in G^i$, and 0 otherwise. |
| $t, T$ | Number of gradient descent iterations. |
| $\eta$ | Learning rate. |
| $\mathbf{W}_V, \mathbf{W}_{KQ}$ | Parameter matrices of the transformer. |
| $\mathcal{L}$ | Population loss (objective function). |
| $\mathcal{L}_{\text{O.O.D.}}$ | Out of distribution loss. |
| $C_1(t), C_2(t), C_3(t)$ | Coefficients of the decompositions of $\mathbf{W}_V$ and $\mathbf{W}_{KQ}$ during the training. |

# B    ADDITIONAL RELATED WORKS

**Optimization of transformers.** There exist multiple recent works studying the optimizations of transformers, most of which focus on the single-layer architecture. Zhang et al. (2020); Kunstner et al. (2023); Pan & Li (2023); Li et al. (2024a) investigate performance comparison between the adaptive methods and SGD under different settings from both theoretical and empirical perspectives. Li et al. (2023b) investigates the optimal parameters of transformers applied to a masked topic structure model similar to the Bert framework through a two-stage training regime. Ildiz et al. (2024); Chen et al. (2024a); Shi & Cao (2025) explain the mechanism of attention from the perspective of Markov chains. Tian et al. (2023; 2024) study the training dynamics of transformers, jointly with a decoder layer and a fully-connected layer, respectively. Li et al. (2024b) analyzes transformer training behavior in the context of one-nearest neighbor selection. Gao et al. (2024) addresses the global convergence of transformers given certain prerequisites. Tarzanagh et al. (2023a;b) demonstrates that single-layer attention mechanisms can converge directionally towards the hard margin solution typical of Support Vector Machines (SVMs). Furthermore, Li et al. (2023a) presents a generalization error bound for vision transformers optimized using stochastic gradient descent. Some works consider the transformers to perform certain algorithms. He et al. (2025c) theoretically characterizes Softmax attention as approximating the Expectation and Maximization updates in EM for Gaussian

mixture models. He et al. (2025b) show that multi-layer Transformers can provably learn and implement spectral methods for Gaussian mixture models via pre-training. Furthermore, many other existing works investigate the optimization of transformers under the so-called "in-context learning" settings (Chen et al., 2024b; Huang et al., 2024; Zhang et al., 2024b;c; Nichani et al., 2024; Huang et al., 2025; Chen et al., 2025; Li et al., 2025; Cao et al., 2025). Based on the framework proposed in (Zhang et al., 2024b), Huang et al. (2024) extends this result to one-layer softmax attention transformers. Siyu et al. (2024) investigates the multi-head self-attention under this setting, and summarizes two distinct patterns among all heads. Nichani et al. (2024) demonstrates that when solving in-context learning tasks with latent causal structure, transformers can encode the latent causal graph. Huang et al. (2025) demonstrates that Chain of Thought (CoT) prompting enables Transformer models to learn to perform multi-step gradient descent and effectively recover true weights. Chen et al. (2025) focuses on the test time computing on the in-context linear regression. Li et al. (2025) studies the context hijacking phenomenon by investigating an optimization procedure with different learning rates. Cao et al. (2025) proves that Transformers can implement in-context maximum likelihood estimation and autoregressive sampling for Bayesian networks, establishing their capability to simulate MLE-based sequence generation.

**Teacher-student framework for training neural networks.** We also introduce some related theoretical works regarding the training of a "student" neural network under the guidance of a "teacher model" (Brutzkus & Globerson, 2017; Tian, 2017; Soltanolkotabi, 2017; Goel et al., 2018; Du et al., 2018b;a; Zhou et al., 2019; Liu et al., 2019; Xu & Du, 2023). Several studies establish convergence guarantees for gradient descent in specific ReLU network settings: Brutzkus & Globerson (2017) demonstrated polynomial-time global convergence for one-hidden-layer non-overlapping convolutional ReLU networks with Gaussian inputs; Tian (2017) characterized critical points and proved gradient descent convergence for two-layer ReLU student-teacher networks under Gaussian inputs; and Du et al. (2018b;a) provided polynomial-time recovery guarantees for learning convolutional ReLU filters and networks, respectively, using (stochastic) gradient descent, even with potential spurious minimizers and for general or Gaussian inputs. Furthermore, Zhou et al. (2019) and Liu et al. (2019) showed that methods like perturbed gradient descent with noise annealing or specific normalizations and initializations can achieve polynomial-time global convergence in convolutional neural networks (including ResNets) despite the presence of spurious local optima. Research focusing on single ReLU scenarios includes Soltanolkotabi (2017)'s analysis of linear convergence for a single ReLU in a high-dimensional Gaussian model with structured weights, and Xu & Du (2023)'s finding that over-parameterizing a student network to learn a single target ReLU neuron under Gaussian inputs can surprisingly slow convergence. Goel et al. (2018) introduced Convotron, a provably efficient algorithm for one-hidden-layer convolutional networks with general patches, achieving global convergence through noise-tolerant stochastic updates without requiring special initialization or learning rate tuning. Zhao & Zhu (2024) studies the statistical limits of knowledge transfer over finite domains, characterizing minimax rates under different levels of teacher supervision.

## C  COMPARISON WITH WANG ET AL. (2024)

In this section, we compare the essential optimization dynamics in Wang et al. (2024) and our works. Wang et al. (2024) and our work both rely on the symmetry of Gaussian data and the uniform distribution among the target tokens expected to be selected. A critical technical step shared by both analyses is to simplify the optimization regarding the full parameter matrices to investigate the evolutions of several specific scalars, as demonstrated in Lemma 3.2 in Wang et al. (2024) and in our Lemma D.2. Specifically, the analysis in Wang et al. (2024) tracks the evolution of two scalars, $\alpha(t)$ and $C(t)$, for which the coefficients are essentially minimizing the loss

$$\widetilde{\mathcal{L}}(\alpha, C) = \frac{d}{2(D-K)} \left[ K(D-K) \left( \frac{\alpha}{K + (D-K)e^{-C}} - \frac{1}{K} \right)^2 + \alpha^2 \left( 1 - \frac{K}{K + (D-K)e^{-C}} \right)^2 \right],$$
(C.1)

as demonstrated on top of Page 31 in Wang et al. (2024).

As demonstrated in Lemma D.2, our analysis focus on the scalars $C_1(t), C_2(t), C_3(t)$. When the teacher model is reduced to the "sparse token selection" task defined in Example 2.3, with $\mathbf{V}^* = \mathbf{I}_d$

and without activation function, the coefficients $C_1(t), C_2(t), C_3(t)$ essentially minimize the loss

$$\widetilde{\mathcal{L}}(C_1, C_2, C_3) = \frac{dD}{2(D-K)}\left[K(D-K)\left(\frac{C_1}{K + (D-K)e^{-\frac{C_2+C_3}{\sqrt{D}}}} - \frac{1}{K}\right)^2\right.$$
$$\left. + C_1^2\left(1 - \frac{K}{K + (D-K)e^{-\frac{C_2+C_3}{\sqrt{D}}}}\right)^2\right] \tag{C.2}$$

Comparing these two loss functions in (C.2) and (C.2), we can observe that they essentially share the same function structure. Specifically, if we regard $\frac{C_2+C_3}{\sqrt{D}}$ in (C.1) as one term, playing the role as $C$ in (C.1), then these two functions only differ by a factor $D$. Therefore, while the setting of the "sparse token selection" task in our work is different from that considered in Wang et al. (2024), they can be formulated into an essentially identical optimization problem. Notably, the loss in (C.2) is only the special case in our setting with $\mathbf{V}^* = \mathbf{I}_d$ and without activation function, while the general case is much more complicated and provided in Lemma D.2. Therefore, the setting considered in our work is more general compared with that in Wang et al. (2024), from a technical perspective. This also highlights that establishing a tight convergence rate with a matching lower bound indeed constitutes a technical advantage of our work.

## D    PROOF OF THEOREM 3.1

In this section, we provide a detailed proof for Theorem 3.1. We first introduce several notations used in the following proof. For each $i \in [D]$, we use $G^i$ to denote the index set to which the entries of $i$-th column of $\mathbf{S}^*$ is $\frac{1}{k}$, i.e. $\mathbf{S}^*_{i',i} = \frac{1}{K}$ if $i' \in G^i$ and 0 otherwise. With this notation, we can express that $\left[f^*(\mathbf{X})\right]_{m,i} = \sigma(\mathbf{v}_m^{*\top}\mathbf{X}\mathbf{1}_{G^i})) = \frac{1}{K}\sigma(\sum_{i'\in G^i}\langle\mathbf{v}_m^*, \mathbf{x}_{i'}\rangle)$. In addition we let $\mathbf{V}^* = [\mathbf{v}_1^*, \mathbf{v}_2^*, \dots, \mathbf{v}_M^*]^\top$, and $\mathbf{W}_V = [\mathbf{w}_{V,1}, \mathbf{w}_{V,2}, \dots, \mathbf{w}_{V,M}]^\top \in \mathbb{R}^{M \times d}$. Based on this notation, it is equivalent to consider the gradient descent updating regarding each $\mathbf{w}_{V,m}$ for all $m \in [M]$, expressed as

$$\mathbf{w}_{V,m}^{(t+1)} = \mathbf{w}_{V,m}^{(t)} - \eta\nabla_{\mathbf{w}_{V,m}}\mathcal{L}(\mathbf{W}_V^{(t)}; \mathbf{W}_{KQ}^{(t)}). \tag{D.1}$$

In the following proof, we will consider the gradient descent updating details for each $\mathbf{w}_{V,m}^{(t)}$, and derive the conclusion for $\mathbf{W}_V^{(t)}$ based on the result of $\mathbf{w}_{V,m}^{(t)}$ for all $m \in [M]$. For simplicity of presentation, we assume that each $\mathbf{v}_m^*$ is normalized in the remaining sections, i.e. $\|\mathbf{v}_m^*\|_2 = 1$ for all $m \in [M]$, without loss of generality (W.L.O.G.). However, our theoretical findings and proofs can be directly extended to the case where $\mathbf{v}_m$ is not normalized. For each $\mathbf{v}_m^*$, let $\mathbf{\Gamma}_m = [\mathbf{v}_m^*, \boldsymbol{\xi}_{m,2}, \dots, \boldsymbol{\xi}_{m,d}] \in \mathbb{R}^{d\times d}$ be an orthogonal matrix with $\mathbf{v}_m$ being its first column. (Actually, if $\mathbf{v}_m^*$ is not normalized, the first column of $\mathbf{\Gamma}_m$ will be $\frac{\mathbf{v}_m^*}{\|\mathbf{v}_m^*\|_2}$.)

Furthermore, we introduce several definitions regarding the expectations of Gaussian random variables. Let $x_1 \sim \mathcal{N}(0, a)$, $x_2 \sim \mathcal{N}(0, b)$, and $x_3 \sim \mathcal{N}(0, c)$ be three independent Gaussian random variables. In addition, $\sigma(\cdot)$ can be the identity map, the ReLU activation function, and the Leaky ReLU activation function, with $\kappa$ denoting the coefficient of the Leaky ReLU activation function when the input is negative. Specifically, when $\sigma(\cdot)$ indicates the Leaky ReLU activation function, $\sigma(x) = x\mathbb{1}_{\{x\geq 0\}} + \kappa x\mathbb{1}_{\{x<0\}}$. Then, based on these notations, we define that

$$F_1(a) = \mathbb{E}[x_1\sigma(x_1)\sigma'(x_1)]; \tag{D.2}$$
$$F_2(a, b) = \mathbb{E}[x_1\sigma(x_1 + x_2)\sigma'(x_1 + x_2)]; \tag{D.3}$$
$$F_3(a, b) = \mathbb{E}[(x_1 + x_2)\sigma(x_1)\sigma'(x_1 + x_2)]; \tag{D.4}$$
$$F_4(a, b, c) = \mathbb{E}[x_1\sigma(x_1 + x_2)\sigma'(x_1 + x_2 + x_3)]; \tag{D.5}$$
$$F_5(a, b, c) = \mathbb{E}[x_2\sigma(x_1)\sigma'(x_1 + x_2 + x_3)]. \tag{D.6}$$

We provide the detailed calculations for these expectations in Section F.1

### D.1    DETAILED GRADIENT DESCENT UPDATING RULES

In this subsection, we introduce and prove several lemmas regarding the calculation details regarding the gradient descent iterative rule (D.1) and (3.4).

**Lemma D.1.** The gradient descent updating regarding $\mathbf{w}_{V,m}^{(t)}$ for all $m \in [M]$ and $\mathbf{W}_{KQ}^{(t)}$, which have been defined in (D.1) and (3.4), can be rewritten as

$$\mathbf{w}_{V,m}^{(t+1)} = \mathbf{w}_{V,m}^{(t)} + \eta \sum_{i=1}^{D} \sum_{i_1=1}^{D} \mathbb{E}\left[ \left[ \mathbf{Y}_{m,i} - \sigma\left( \sum_{i_1=1}^{D} \langle \mathbf{w}_{V,m}^{(t)}, \mathbf{x}_{i_1} \rangle \mathbf{S}_{i_1,i}^{(t)} \right) \right] \sigma'\left( \sum_{i_1=1}^{D} \langle \mathbf{w}_{V,m}^{(t)}, \mathbf{x}_{i_1} \rangle \mathbf{S}_{i_1,i}^{(t)} \right) \mathbf{x}_{i_1} \mathbf{S}_{i_1,i}^{(t)} \right];$$
(D.7)

$$\mathbf{W}_{KQ}^{(t+1)} = \mathbf{W}_{KQ}^{(t)} + \frac{\eta}{\sqrt{D}} \sum_{m=1}^{M} \sum_{i=1}^{D} \mathbb{E}\left[ \left[ \mathbf{Y}_{m,i} - \sigma\left( \sum_{i_1=1}^{D} \langle \mathbf{w}_{V,m}^{(t)}, \mathbf{x}_{i_1} \rangle \mathbf{S}_{i_1,i}^{(t)} \right) \right] \sigma'\left( \sum_{i_1=1}^{D} \langle \mathbf{w}_{V,m}^{(t)}, \mathbf{x}_{i_1} \rangle \mathbf{S}_{i_1,i}^{(t)} \right) \right.$$
$$\left. \cdot \sum_{i_1=1}^{D} \sum_{i_2=1}^{D} \langle \mathbf{w}_{V,m}^{(t)}, \mathbf{x}_{i_1} \rangle \mathbf{S}_{i_1,i}^{(t)} \mathbf{S}_{i_2,i}^{(t)} (\mathbf{p}_{i_1} - \mathbf{p}_{i_2}) \mathbf{p}_i^\top \right].$$
(D.8)

*Proof of Lemma D.1.* By the chain rule of derivatives, we have

$$\mathbf{w}_{V,m}^{(t+1)} = \mathbf{w}_{V,m}^{(t)} - \eta \nabla_{\mathbf{w}_{V,m}} \mathcal{L}(\mathbf{W}_V^{(t)}; \mathbf{W}_{KQ}^{(t)}) = \mathbf{w}_{V,m}^{(t)} - \frac{\eta}{2} \nabla_{\mathbf{w}_{V,m}} \mathbb{E}\left[ \|\mathbf{Y} - \mathrm{TF}(\mathbf{Z}; \mathbf{W}_V; \mathbf{W}_{KQ})\|_F^2 \right]$$

$$= \mathbf{w}_{V,m}^{(t)} - \frac{\eta}{2} \sum_{m'=1}^{M} \sum_{i=1}^{D} \nabla_{\mathbf{w}_{V,m}} \mathbb{E}\left[ (\mathbf{Y}_{m',i} - \sigma(\mathbf{W}_V^{(t)} \mathbf{X} \mathbf{S}^{(t)})_{m',i})^2 \right]$$

$$= \mathbf{w}_{V,m}^{(t)} - \frac{\eta}{2} \sum_{m'=1}^{M} \sum_{i=1}^{D} \nabla_{\mathbf{w}_{V,m}} \mathbb{E}\left[ \left[ \mathbf{Y}_{m',i} - \sigma\left( \sum_{i_1=1}^{D} \langle \mathbf{w}_{V,m'}^{(t)}, \mathbf{x}_{i_1} \rangle \mathbf{S}_{i_1,i}^{(t)} \right) \right]^2 \right]$$

$$= \mathbf{w}_{V,m}^{(t)} + \eta \sum_{i=1}^{D} \sum_{i_1=1}^{D} \mathbb{E}\left[ \left[ \mathbf{Y}_{m,i} - \sigma\left( \sum_{i_1=1}^{D} \langle \mathbf{w}_{V,m}^{(t)}, \mathbf{x}_{i_1} \rangle \mathbf{S}_{i_1,i}^{(t)} \right) \right] \sigma'\left( \sum_{i_1=1}^{D} \langle \mathbf{w}_{V,m}^{(t)}, \mathbf{x}_{i_1} \rangle \mathbf{S}_{i_1,i}^{(t)} \right) \mathbf{x}_{i_1} \mathbf{S}_{i_1,i}^{(t)} \right],$$

where the last equality holds simply by the chain rule of differentiation. This proves (D.7). Next for $\mathbf{W}_{KQ}$, we have [1]

$$\mathbf{W}_{KQ}^{(t+1)} = \mathbf{W}_{KQ}^{(t)} - \eta \nabla_{\mathbf{W}_{KQ}} \mathcal{L}(\mathbf{W}_V^{(t)}; \mathbf{W}_{KQ}^{(t)}) = \mathbf{W}_{KQ}^{(t)} - \frac{\eta}{2} \nabla_{\mathbf{W}_{KQ}} \mathbb{E}\left[ \|\mathbf{Y} - \mathrm{TF}(\mathbf{Z}; \mathbf{W}_V; \mathbf{W}_{KQ})\|_F^2 \right]$$

$$= \mathbf{W}_{KQ}^{(t)} - \frac{\eta}{2} \sum_{m=1}^{M} \sum_{i=1}^{D} \nabla_{\mathbf{W}_{KQ}} \mathbb{E}\left[ (\mathbf{Y}_{m,i} - \sigma(\mathbf{W}_V^{(t)} \mathbf{X} \mathbf{S}^{(t)})_{m,i})^2 \right]$$

$$= \mathbf{W}_{KQ}^{(t)} + \eta \sum_{m=1}^{M} \sum_{i=1}^{D} \mathbb{E}\left[ \left[ \mathbf{Y}_{m,i} - \sigma\left( \sum_{i_1=1}^{D} \langle \mathbf{w}_{V,m}^{(t)}, \mathbf{x}_{i_1} \rangle \mathbf{S}_{i_1,i}^{(t)} \right) \right] \sigma'\left( \sum_{i_1=1}^{D} \langle \mathbf{w}_{V,m}^{(t)}, \mathbf{x}_{i_1} \rangle \mathbf{S}_{i_1,i}^{(t)} \right) \right.$$
$$\left. \cdot \underbrace{\nabla_{\mathbf{W}_{KQ}} (\mathbf{w}_{V,m}^{(t)})^\top \mathbf{X} \mathcal{S}\left( \frac{\mathbf{P} \mathbf{W}_{KQ}^{(t)} \mathbf{P}_i}{\sqrt{D}} \right)}_{I} \right].$$
(D.9)

For the derivative calculation of $I$, we have

$$I = \sum_{i_1=1}^{D} \nabla_{\mathbf{W}_{KQ}} \left[ (\mathbf{w}_{V,m}^{(t)})^\top \mathbf{X} \right]_{i_1} \left[ \mathcal{S}\left( \frac{\mathbf{P} \mathbf{W}_{KQ}^{(t)} \mathbf{P}_i}{\sqrt{D}} \right) \right]_{i_1} = \sum_{i_1=1}^{D} \langle \mathbf{w}_{V,m}^{(t)}, \mathbf{x}_{i_1} \rangle \nabla_{\mathbf{W}_{KQ}} \left[ \mathcal{S}\left( \frac{\mathbf{P} \mathbf{W}_{KQ}^{(t)} \mathbf{P}_i}{\sqrt{D}} \right) \right]_{i_1}$$

$$= \sum_{i_1=1}^{D} \langle \mathbf{w}_{V,m}^{(t)}, \mathbf{x}_{i_1} \rangle \sum_{i_2=1}^{D} \frac{\mathrm{d}\left[ \mathcal{S}\left( \frac{\mathbf{P} \mathbf{W}_{KQ}^{(t)} \mathbf{P}_i}{\sqrt{D}} \right) \right]_{i_1}}{\mathrm{d}\left[ \frac{\mathbf{P} \mathbf{W}_{KQ}^{(t)} \mathbf{P}_i}{\sqrt{D}} \right]_{i_2}} \nabla_{\mathbf{W}_{KQ}} \left[ \frac{\mathbf{P} \mathbf{W}_{KQ}^{(t)} \mathbf{P}_i}{\sqrt{D}} \right]_{i_2}$$

$$= \frac{1}{\sqrt{D}} \sum_{i_1=1}^{D} \langle \mathbf{w}_{V,m}^{(t)}, \mathbf{x}_{i_1} \rangle \sum_{i_2=1}^{D} \left[ \mathcal{S}'\left( \frac{\mathbf{P} \mathbf{W}_{KQ}^{(t)} \mathbf{P}_i}{\sqrt{D}} \right) \right]_{i_1,i_2} \mathbf{p}_{i_2} \mathbf{p}_i^\top = \sum_{i_1=1}^{D} \sum_{i_2 \neq i_1} \langle \mathbf{w}_{V,m}^{(t)}, \mathbf{x}_{i_1} \rangle \mathbf{S}_{i_1,i}^{(t)} \mathbf{S}_{i_2,i}^{(t)} (\mathbf{p}_{i_1} - \mathbf{p}_{i_2}) \mathbf{p}_i^\top.$$
(D.10)

---

[1] Here we slightly abuse the notation of $\mathcal{S}(\cdot)$. If the input is a $D$-dimensional vector, $\mathcal{S}(\cdot)$ denotes the softmax function from $\mathbb{R}^D \mapsto \mathbb{R}^D$. If the input is a $D_1 \times D_2$-dimensional matrix, $\mathcal{S}(\cdot)$ represents the softmax operator which implements the softmax normalization defined above column-wisely.

The last equality holds as $\mathcal{S}'(\mathbf{a}) = \text{diag}(\mathbf{a}) - \mathcal{S}(\mathbf{a})\mathcal{S}(\mathbf{a})^\top \in \mathbb{R}^{d \times d}$ for any vector $\mathbf{a} \in \mathbb{R}^d$, and consequently,

$$\left[\mathcal{S}'\left(\frac{\mathbf{PW}_{KQ}^{(t)}\mathbf{p}_i}{\sqrt{D}}\right)\right]_{i_1,i_2} = \begin{cases} \left[\mathcal{S}\left(\frac{\mathbf{PW}_{KQ}^{(t)}\mathbf{p}_i}{\sqrt{D}}\right)\right]_{i_1}\left(1 - \left[\mathcal{S}\left(\frac{\mathbf{PW}_{KQ}^{(t)}\mathbf{p}_i}{\sqrt{D}}\right)\right]_{i_1}\right) = \mathbf{S}_{i_1,i}^{(t)}(1 - \mathbf{S}_{i_1,i}^{(t)}), & \text{if } i_1 = i_2; \\ -\left[\mathcal{S}\left(\frac{\mathbf{PW}_{KQ}^{(t)}\mathbf{p}_i}{\sqrt{D}}\right)\right]_{i_1}\left[\mathcal{S}\left(\frac{\mathbf{PW}_{KQ}^{(t)}\mathbf{p}_i}{\sqrt{D}}\right)\right]_{i_2} = -\mathbf{S}_{i_1,i}^{(t)}\mathbf{S}_{i_2,i}^{(t)}, & \text{otherwise.} \end{cases}$$

By substituting the result of $I$ from (D.10) into (D.9), we complete the proof of (D.8). $\qquad\square$

The next lemma demonstrates that the training dynamics of $\mathbf{w}_{V,m}^{(t)}$ for all $m \in [M]$ and $\mathbf{W}_{KQ}^{(t)}$ exhibit specific patterns. Analyzing the training processes described in (D.1) and (D.21) can be reframed as an investigation into the coefficients of these patterns.

**Lemma D.2.** Under the same conditions of Theorem 3.1, there exist a time dependent non-negative scalar $C_1(t)$, and non-negative, monotonically increasing scalars $C_2(t)$ and $C_3(t)$, such that

$$\mathbf{w}_{V,m}^{(t)} = C_1(t) \cdot \mathbf{v}_m^*, \text{ for all } m \in [M];$$

$$\mathbf{W}_{KQ}^{(t)} = C_2(t)\sum_{i=1}^{D}\sum_{i_1 \in G^i} \mathbf{p}_{i_1}\mathbf{p}_i^\top - C_3(t)\sum_{i=1}^{D}\sum_{i_1 \notin G^i} \mathbf{p}_{i_1}\mathbf{p}_i^\top.$$

Due to the specific pattern of $\mathbf{W}_{KQ}^{(t)}$ demonstrated above, there exist a time dependent scalar

$$p(t) = \frac{1}{K + (D-K)e^{-\frac{C_2(t)+C_3(t)}{\sqrt{D}}}},$$

such that $\mathbf{S}_{i_1,i}^{(t)} = p(t)$ for all $i \in [D]$ and $i_1 \in G^i$. Otherwise, $\mathbf{S}_{i_1,i}^{(t)} = \frac{1-Kp(t)}{D-K}$. Additionally, $\frac{1}{D} \le p(t) \le \frac{1}{K}$ and $p(t)$ is monotonically increasing. Based on the definition of $p(t)$, $C_1(t)$, $C_2(t)$, and $C_3(t)$ have the following iterative rules:

$$C_1(t+1) = C_1(t) + D\eta\left(\frac{F_3^{(t)}}{Kp(t)} - C_1(t)F_1^{(t)}\right) = C_1(t) + \frac{\eta D F_3^{(t)}}{Kp(t)}\left(1 - \frac{C_1(t)}{C_1^*(t)}\right);$$

$$C_2(t+1) = C_2(t) + \eta\frac{C_1(t)M}{\sqrt{D}}\left(\frac{1}{K}\left(\frac{F_4^{(t)}}{p(t)} - F_3^{(t)}\right) - C_1(t)\left(F_{2,1}^{(t)} + p(t)F_1^{(t)}\right)\right);$$

$$C_3(t+1) = C_3(t) - \eta\frac{C_1(t)M(1-Kp(t))}{\sqrt{D}(D-K)}\left(\left(\frac{F_3^{(t)}}{Kp(t)} - \frac{(D-K)F_5^{(t)}}{Kp(t)(1-Kp(t))}\right) - C_1(t)\left(F_1^{(t)} - \frac{(D-K)F_{2,2}^{(t)}}{1-Kp(t)}\right)\right),$$

where $F_1^{(t)} = F_1\left(Kp(t)^2 + \frac{(1-Kp(t))^2}{D-K}\right)$, $F_{2,1}^{(t)} = F_2\left(p(t)^2, (K-1)p(t)^2 + \frac{(1-Kp(t))^2}{D-K}\right)$, $F_{2,2}^{(t)} = F_2\left(\frac{(1-Kp(t))^2}{(D-K)^2}, Kp(t)^2 + \frac{(D-K-1)(1-Kp(t))^2}{(D-K)^2}\right)$, $F_3^{(t)} = F_3\left(Kp(t)^2, \frac{(1-Kp(t))^2}{D-K}\right)$, $F_4^{(t)} = F_4\left(p(t)^2, (K-1)p(t)^2, \frac{(1-Kp(t))^2}{D-K}\right)$, $F_5^{(t)} = F_5\left(Kp(t)^2, \frac{(1-Kp(t))^2}{(D-K)^2}, \frac{(D-K-1)(1-Kp(t))^2}{(D-K)^2}\right)$, and $C_1^*(t) = \frac{F_3^{(t)}}{Kp(t)F_1^{(t)}}$. In addition, based on all these definitions, the coefficients $C_1(t)$, $C_2(t)$, and $C_3(t)$ are essentially minimizing the following loss function by gradient descent

$$\widetilde{\mathcal{L}}(C_1, C_2, C_3) = \frac{c_\sigma D\|\mathbf{V}^*\|_F^2}{2(D-K)}\left[K(D-K)\left(\frac{1}{K} - C_1p\right)^2 + C_1^2\left(1 - Kp\right)^2\right] - D\|\mathbf{V}^*\|_F^2 F_6(C_1, p).$$

where $c_\sigma$ is an absolute constant such that $c_\sigma = \mathbb{1}_{\{\sigma(\cdot) \text{ is identity map}\}} + \frac{1}{2}\mathbb{1}_{\{\sigma(\cdot) \text{ is ReLU}\}} + \frac{1+\kappa^2}{2}\mathbb{1}_{\{\sigma(\cdot) \text{ is Leaky ReLU}\}}$. In addition, $F_6(C_1, p)$ is defined as

$$F_6 = \begin{cases} 0; & \text{If } \sigma(\cdot) \text{ is identity map} \\ pC_1^2\left(Kp\left(\frac{1}{\pi}\arctan\left(\frac{p\sqrt{K(D-K)}}{1-Kp}\right) - \frac{1}{2}\right) + \frac{(1-Kp)\sqrt{K}}{\pi\sqrt{D-K}}\right); & \text{If } \sigma(\cdot) \text{ is ReLU activation} \\ (1-\kappa)^2pC_1^2\left(Kp\left(\frac{1}{\pi}\arctan\left(\frac{p\sqrt{K(D-K)}}{1-Kp}\right) - \frac{1}{2}\right) + \frac{(1-Kp)\sqrt{K}}{\pi\sqrt{D-K}}\right). & \text{If } \sigma(\cdot) \text{ is Leaky ReLU activation} \end{cases}$$

We establish these conclusions by induction. It can be easily verified that all these conclusions hold at $t = 0$, since the parameters are initialized as $\mathbf{W}_V^{(0)} = \mathbf{0}_{M \times d}$ and $\mathbf{W}_{KQ}^{(0)} = \mathbf{0}_{D \times D}$. However, for the sake of conciseness and coherence in the presentation, we rearrange the contents of Lemma D.2 into Lemma D.4 and Lemma D.8, including the relevant details regarding $\mathbf{W}_{V,m}$ and $\mathbf{W}_{KQ}$ respectively. To prevent the proof of a single Lemma D.2 from becoming overly lengthy, we prove Lemmas D.4 and D.8 separately.

As we use induction, we assume that the conclusions of both Lemma D.4 and Lemma D.8 hold at the current iteration. We then demonstrate that the conclusion of either Lemma D.4 or Lemma D.8 holds at the next iteration, depending on which lemma we are proving. It is important to clarify that this is not circular reasoning; all these contents can indeed be organized into a single Lemma D.2. It is reasonable to assume that all conclusions hold for each iteration and to verify that these conclusions remain valid for the next iteration, as long as we rigorously demonstrate their validity at the outset.

In the following, we introduce and prove Lemma D.4 and Lemma D.8 respectively. Besides, the notations defined in Lemma D.2, containing $p(t)$, $F_1^{(t)}$, $F_{2,1}^{(t)}$, $F_{2,2}^{(t)}$, $F_3^{(t)}$, $F_4^{(t)}$, and $F_5^{(t)}$ will remain consistent unless stated otherwise.

We first introduce and prove a lemma regarding the ratio between $F_1^{(t)}$ and $F_3^{(t)}$, which will be utilized in the proof of Lemma D.4.

**Lemma D.3.** Under the same conditions of Theorem 3.1, for $F_1^{(t)}$ and $F_3^{(t)}$ defined in Lemma D.2, it holds that

$$Kp(t) \leq \frac{F_3^{(t)}}{F_1^{(t)}} \leq \sqrt{DK}p(t).$$

*Proof of Lemma D.3.* By Lemma F.1 and Lemma F.5, we can derive that

- If $\sigma(\cdot)$ is the identity map, then

$$\frac{F_1^{(t)}}{F_3^{(t)}} = \frac{(D-K)Kp(t)^2}{DKp(t)^2 - 2Kp(t) + 1} = Kp(t)\frac{D-K}{DKp(t) + \frac{1}{p(t)} - 2K} \geq Kp(t);$$

$$\frac{F_1^{(t)}}{F_3^{(t)}} = \frac{(D-K)Kp(t)^2}{DKp(t)^2 - 2Kp(t) + 1} = Kp(t)\frac{D-K}{DKp(t) + \frac{1}{p(t)} - 2K} \leq \sqrt{DK}p(t).$$

The last inequality is derived by $2\sqrt{DK} - 2K \leq DKp(t) + \frac{1}{p(t)} - 2K \leq D - K$ as $\frac{1}{D} \leq p(t) < \frac{1}{K}$.

- If $\sigma(\cdot)$ is ReLU activation function, it is also straightforward that

$$\frac{F_1^{(t)}}{F_3^{(t)}} \geq \frac{2(D-K)\frac{Kp(t)^2}{2}}{DKp(t)^2 - 2Kp(t) + 1} = Kp(t)\frac{D-K}{DKp(t) + \frac{1}{p(t)} - 2K} \geq Kp(t).$$

On the other hand, by Lemma F.5, it can be derived that

$$\frac{F_1^{(t)}}{F_3^{(t)}} \leq \frac{2(D-K)\left(\frac{Kp(t)^2}{2} + \frac{1}{2\pi}\sqrt{\frac{K}{D-K}}p(t)(1 - Kp(t))\right)}{DKp(t)^2 - 2Kp(t) + 1}$$

$$= Kp(t)\left(\frac{D-K}{DKp(t) + \frac{1}{p(t)} - 2K} + \sqrt{\frac{D-K}{K}}\frac{1 - Kp(t)}{\pi(DKp(t)^2 - 2Kp(t) + 1)}\right)$$

$$\leq Kp(t)\left(\frac{1}{2}\sqrt{\frac{D}{K}} + \frac{1}{\pi}\sqrt{\frac{D-K}{K}} + \frac{1}{2}\right) \leq \sqrt{DK}p(t),$$

where the penultimate inequality holds since $DKp(t) + \frac{1}{p(t)} - 2K \geq 2\sqrt{DK} - 2K$, and $\frac{1-Kp(t)}{DKp(t)^2 - 2Kp(t) + 1}$ is a decreasing function w.r.t. $p(t)$ as the numerator is decreasing w.r.t. $p(t)$ while denominator is increasing w.r.t. $p(t)$. Therefore, it takes the maximum value when $p(t) = \frac{1}{D}$, and consequently $\frac{1-Kp(t)}{DKp(t)^2 - 2Kp(t) + 1} \leq \sqrt{\frac{D-K}{K}}$.

- If $\sigma(\cdot)$ is Leaky ReLU activation function, by utilizing a similar calculation, it holds that

$$\frac{F_1^{(t)}}{F_3^{(t)}} \geq \frac{2(D-K)\frac{(1+\kappa)^2 Kp(t)^2}{2}}{(1+\kappa)^2\big(DKp(t)^2 - 2Kp(t) + 1\big)} = Kp(t)\frac{(D-K)p(t)}{DKp(t)^2 - 2Kp(t) + 1} \geq Kp(t);$$

$$\frac{F_1^{(t)}}{F_3^{(t)}} \leq \frac{2(D-K)\Big(\frac{(1+\kappa)^2 Kp(t)^2}{2} + \frac{(1-\kappa)^2}{2\pi}\sqrt{\frac{K}{D-K}}p(t)(1 - Kp(t))\Big)}{(1+\kappa)^2\big(DKp(t)^2 - 2Kp(t) + 1\big)} \leq \sqrt{DK}p(t).$$

This completes the proof. $\qquad\square$

**Lemma D.4** (Restatement of Lemma D.2, the first part). *Under the same conditions of Theorem 3.1, there exist time dependent non-negative scalars $C_1(t)$, such that*

$$\mathbf{w}_{V,m}^{(t)} = C_1(t) \cdot \mathbf{v}_m^*, \text{ for all } m \in [M], \tag{D.11}$$

*where $C_1(t)$ has the following iterative rule:*

$$C_1(t+1) = C_1(t) + D\eta\bigg(\frac{F_3^{(t)}}{Kp(t)} - C_1(t)F_1^{(t)}\bigg) = C_1(t) + \frac{\eta D F_3^{(t)}}{Kp(t)}\bigg(1 - \frac{C_1(t)}{C_1^*(t)}\bigg), \tag{D.12}$$

*where $C_1^*(t) = \frac{F_3^{(t)}}{Kp(t)F_1^{(t)}}$.*

*Proof of Lemma D.4.* First at the initialization $t = 0$, we have $\mathbf{W}_V^{(0)} = \mathbf{0}_{M\times d}$, satisfying (D.11). Next, we assume that at $t$-th iteration, the conclusion of (D.11) still holds, and we will prove that it continues to hold at the $t + 1$-th iteration. Actually, it suffices to show that

$$\nabla_{\mathbf{w}_{V,m}}\mathcal{L}(\mathbf{W}_V^{(t)}; \mathbf{W}_{KQ}^{(t)}) = c_1(t) \cdot \mathbf{v}_m^*, \text{ for all } m \in [M], \tag{D.13}$$

where $c_1(t)$ is a time-dependent scalar. By Lemma D.1, we have

$$\nabla_{\mathbf{w}_{V,m}}\mathcal{L}(\mathbf{W}_V^{(t)}; \mathbf{W}_{KQ}^{(t)}) = -\sum_{i=1}^{D}\sum_{i_1=1}^{D}\mathbb{E}\bigg[\Big[\mathbf{Y}_{m,i} - \sigma\Big(\sum_{i_1=1}^{D}\langle\mathbf{w}_{V,m}^{(t)}, \mathbf{x}_{i_1}\rangle\mathbf{S}_{i_1,i}^{(t)}\Big)\Big]\sigma'\Big(\sum_{i_1=1}^{D}\langle\mathbf{w}_{V,m}^{(t)}, \mathbf{x}_{i_1}\rangle\mathbf{S}_{i_1,i}^{(t)}\Big)\mathbf{x}_{i_1}\mathbf{S}_{i_1,i}^{(t)}\bigg]$$

$$= -\underbrace{\sum_{i=1}^{D}\sum_{i_1=1}^{D}\mathbb{E}\bigg[\mathbf{Y}_{m,i}\sigma'\Big(\sum_{i_1=1}^{D}\langle\mathbf{w}_{V,m}^{(t)}, \mathbf{x}_{i_1}\rangle\mathbf{S}_{i_1,i}^{(t)}\Big)\mathbf{x}_{i_1}\mathbf{S}_{i_1,i}^{(t)}\bigg]}_{I_1}$$

$$+ \underbrace{\sum_{i=1}^{D}\sum_{i_1=1}^{D}\mathbb{E}\bigg[\sigma\Big(\sum_{i_1=1}^{D}\langle\mathbf{w}_{V,m}^{(t)}, \mathbf{x}_{i_1}\rangle\mathbf{S}_{i_1,i}^{(t)}\Big)\sigma'\Big(\sum_{i_1=1}^{D}\langle\mathbf{w}_{V,m}^{(t)}, \mathbf{x}_{i_1}\rangle\mathbf{S}_{i_1,i}^{(t)}\Big)\mathbf{x}_{i_1}\mathbf{S}_{i_1,i}^{(t)}\bigg]}_{I_2}$$

$$\tag{D.14}$$

For $I_1$, we have

$$I_1 = \sum_{i=1}^{D}\sum_{i_1=1}^{D}\mathbb{E}\bigg[\mathbf{Y}_{m,i}\sigma'\Big(\sum_{i_1=1}^{D}\langle\mathbf{w}_{V,m}^{(t)}, \mathbf{x}_{i_1}\rangle\mathbf{S}_{i_1,i}^{(t)}\Big)\mathbf{\Gamma}_m\mathbf{\Gamma}_m^\top\mathbf{x}_{i_1}\mathbf{S}_{i_1,i}^{(t)}\bigg]$$

$$= \sum_{i=1}^{D}\sum_{i_1=1}^{D}\mathbb{E}\bigg[\big[f^*(\mathbf{X})\big]_{m,i}\sigma'\Big(\sum_{i_1=1}^{D}\langle\mathbf{w}_{V,m}^{(t)}, \mathbf{x}_{i_1}\rangle\mathbf{S}_{i_1,i}^{(t)}\Big)\langle\mathbf{v}_m^*, \mathbf{x}_{i_1}\rangle\mathbf{S}_{i_1,i}^{(t)}\bigg] \cdot \mathbf{v}_m^*$$

$$+ \sum_{i=1}^{D}\sum_{i_1=1}^{D}\sum_{k=2}^{d}\mathbb{E}\bigg[\big[f^*(\mathbf{X})\big]_{m,i}\sigma'\Big(\sum_{i_1=1}^{D}\langle\mathbf{w}_{V,m}^{(t)}, \mathbf{x}_{i_1}\rangle\mathbf{S}_{i_1,i}^{(t)}\Big)\langle\boldsymbol{\xi}_{m,k}, \mathbf{x}_{i_1}\rangle\mathbf{S}_{i_1,i}^{(t)}\bigg] \cdot \boldsymbol{\xi}_{m,k}$$

$$= \sum_{i=1}^{D}\sum_{i_1=1}^{D}\mathbb{E}\bigg[\big[f^*(\mathbf{X})\big]_{m,i}\sigma'\Big(\sum_{i_1=1}^{D}\langle\mathbf{w}_{V,m}^{(t)}, \mathbf{x}_{i_1}\rangle\mathbf{S}_{i_1,i}^{(t)}\Big)\langle\mathbf{v}_m^*, \mathbf{x}_{i_1}\rangle\mathbf{S}_{i_1,i}^{(t)}\bigg] \cdot \mathbf{v}_m^*.$$

The first quality holds as $\mathcal{E}$ is mean-zero and independent with $\mathbf{X}$, and the last equality holds as the orthogonality between $\mathbf{v}_m^*$ and $\boldsymbol{\xi}_{m,k}$ implies that $\langle \mathbf{v}_m^*, \mathbf{x}_{i_2} \rangle$ is independent with $\langle \boldsymbol{\xi}_{m,k}, \mathbf{x}_{i_1} \rangle$ for all $i_1, i_2 \in [D]$. Notice that $\left[ f^*(\mathbf{X}) \right]_{m,i} = \frac{1}{K} \sigma \left( \sum_{i' \in G^i} \langle \mathbf{v}_m^*, \mathbf{x}_{i'} \rangle \right)$ and $\sigma' \left( \sum_{i_1=1}^{D} \langle \mathbf{w}_{V,m}^{(t)}, \mathbf{x}_{i_1} \rangle \mathbf{S}_{i_1,i}^{(t)} \right) = \sigma' \left( C_1(t) \sum_{i_1=1}^{D} \langle \mathbf{v}_m^*, \mathbf{x}_{i_1} \rangle \mathbf{S}_{i_1,i}^{(t)} \right) = \sigma' \left( \sum_{i_1=1}^{D} \langle \mathbf{v}_m^*, \mathbf{x}_{i_1} \rangle \mathbf{S}_{i_1,i}^{(t)} \right)$. Consequently, $\langle \boldsymbol{\xi}_{m,k}, \mathbf{x}_{i_1} \rangle$ is a mean-zero Gaussian random variable, and independent with both $\left[ f^*(\mathbf{X}) \right]_{m,i}$ and $\sigma' \left( \sum_{i_1=1}^{D} \langle \mathbf{w}_{V,m}^{(t)}, \mathbf{x}_{i_1} \rangle \mathbf{S}_{i_1,i}^{(t)} \right)$ simultaneously, implying that

$$\mathbb{E}\left[ \left[ f^*(\mathbf{X}) \right]_{m,i} \sigma' \left( \sum_{i_1=1}^{D} \langle \mathbf{w}_{V,m}^{(t)}, \mathbf{x}_{i_1} \rangle \mathbf{S}_{i_1,i}^{(t)} \right) \langle \boldsymbol{\xi}_{m,k}, \mathbf{x}_{i_1} \rangle \mathbf{S}_{i_1,i}^{(t)} \right]$$

$$= \mathbb{E}\left[ \left[ f^*(\mathbf{X}) \right]_{m,i} \sigma' \left( \sum_{i_1=1}^{D} \langle \mathbf{w}_{V,m}^{(t)}, \mathbf{x}_{i_1} \rangle \mathbf{S}_{i_1,i}^{(t)} \right) \mathbf{S}_{i_1,i}^{(t)} \right] \mathbb{E}[\langle \boldsymbol{\xi}_{m,k}, \mathbf{x}_{i_1} \rangle] = 0.$$

Based on previous results, by plugging $\left[ f^*(\mathbf{X}) \right]_{m,i} = \frac{1}{K} \sigma(\sum_{i_1 \in G^i} \langle \mathbf{v}_m^*, \mathbf{x}_{i_1} \rangle)$ and utilizing the definition of $F_3(a,b)$ in (D.4), we can further derive that

$$I_1 = \frac{1}{K} \sum_{i=1}^{D} \mathbb{E}\left[ \sigma \left( \sum_{i_1 \in G^i} \langle \mathbf{v}_m^*, \mathbf{x}_{i_1} \rangle \right) \sigma' \left( \sum_{i_1=1}^{D} \langle \mathbf{w}_{V,m}^{(t)}, \mathbf{x}_{i_1} \rangle \mathbf{S}_{i_1,i}^{(t)} \right) \left( \sum_{i_1=1}^{D} \langle \mathbf{v}_m^*, \mathbf{x}_{i_1} \rangle \mathbf{S}_{i_1,i}^{(t)} \right) \right] \cdot \mathbf{v}_m^*$$

$$= \frac{1}{p(t)K} \sum_{i=1}^{D} \mathbb{E}\left[ \sigma \left( \sum_{i_1 \in G^i} \langle \mathbf{v}_m^*, \mathbf{x}_{i_1} \rangle p(t) \right) \sigma' \left( \sum_{i_1 \in G^i} \langle \mathbf{v}_m^*, \mathbf{x}_{i_1} \rangle p(t) + \sum_{i_1 \notin G^i} \langle \mathbf{v}_m^*, \mathbf{x}_{i_1} \rangle \frac{1 - Kp(t)}{D - K} \right) \right.$$

$$\left. \cdot \left( \sum_{i_1 \in G^i} \langle \mathbf{v}_m^*, \mathbf{x}_{i_1} \rangle p(t) + \sum_{i_1 \notin G^i} \langle \mathbf{v}_m^*, \mathbf{x}_{i_1} \rangle \frac{1 - Kp(t)}{D - K} \right) \right] \cdot \mathbf{v}_m^*$$

$$= \frac{D}{Kp(t)} F_3 \left( Kp(t)^2, \frac{(1 - Kp(t))^2}{D - K} \right) \cdot \mathbf{v}_m^* = \frac{D F_3^{(t)}}{Kp(t)} \mathbf{v}_m^*.$$

The second equality is derived by fact that $\sigma(ax) = a\sigma(x)$ and $\sigma'(ax) = \sigma'(x)$ if $a \geq 0$, and the definition of $p(t)$. The penultimate equality holds as $\sum_{i_1 \in G^i} \langle \mathbf{v}_m^*, \mathbf{x}_{i_1} \rangle p(t) \sim \mathcal{N}(0, Kp(t)^2)$, $\sum_{i_1 \notin G^i} \langle \mathbf{v}_m^*, \mathbf{x}_{i_1} \rangle \frac{1 - Kp(t)}{D - K} \sim \mathcal{N}\left(0, \frac{(1 - Kp(t))^2}{D - K}\right)$, and they are independent. Then we can conclude the final result by the definition of $F_3(a,b)$ in (D.4). Similar to the process of handling $I_1$, we have the following for $I_2$:

$$I_2 = \sum_{i=1}^{D} \sum_{i_1=1}^{D} \mathbb{E}\left[ \sigma \left( \sum_{i_1=1}^{D} \langle \mathbf{w}_{V,m}^{(t)}, \mathbf{x}_{i_1} \rangle \mathbf{S}_{i_1,i}^{(t)} \right) \sigma' \left( \sum_{i_1=1}^{D} \langle \mathbf{w}_{V,m}^{(t)}, \mathbf{x}_{i_1} \rangle \mathbf{S}_{i_1,i}^{(t)} \right) \boldsymbol{\Gamma}_m \boldsymbol{\Gamma}_m^\top \mathbf{x}_{i_1} \mathbf{S}_{i_1,i}^{(t)} \right]$$

$$= C_1(t) \sum_{i=1}^{D} \mathbb{E}\left[ \sigma \left( \sum_{i_1 \in G^i} \langle \mathbf{v}_m^*, \mathbf{x}_{i_1} \rangle p(t) + \sum_{i_1 \notin G^i} \langle \mathbf{v}_m^*, \mathbf{x}_{i_1} \rangle \frac{1 - Kp(t)}{D - K} \right) \right.$$

$$\cdot \sigma' \left( \sum_{i_1 \in G^i} \langle \mathbf{v}_m^*, \mathbf{x}_{i_1} \rangle p(t) + \sum_{i_1 \notin G^i} \langle \mathbf{v}_m^*, \mathbf{x}_{i_1} \rangle \frac{1 - Kp(t)}{D - K} \right)$$

$$\left. \cdot \left( \sum_{i_1 \in G^i} \langle \mathbf{v}_m^*, \mathbf{x}_{i_1} \rangle p(t) + \sum_{i_1 \notin G^i} \langle \mathbf{v}_m^*, \mathbf{x}_{i_1} \rangle \frac{1 - Kp(t)}{D - K} \right) \right] \cdot \mathbf{v}_m^*$$

$$= D C_1(t) F_1 \left( Kp(t)^2 + \frac{(1 - Kp(t))^2}{D - K} \right) \cdot \mathbf{v}_m^* = D C_1(t) F_1^{(t)} \cdot \mathbf{v}_m^*.$$

where the last equality holds by Lemma F.1. Plugging the calculation results for $I_1$ and $I_2$ into (D.14), we can immediately derive (D.13), which, as we stated previously, directly conclude (D.11). In addition, we can further calculate that

$$\mathbf{w}_{V,m}^{(t+1)} = C_1(t+1) \cdot \mathbf{v}_m^* = \left( C_1(t) + D\eta \left( \frac{F_3^{(t)}}{Kp(t)} - C_1(t) F_1^{(t)} \right) \right) \cdot \mathbf{v}_m^*,$$

which finishes the proof of (D.12). Next, we prove that $C_1(t)$ is always non-negative by induction. Obviously $C_1(t) \geq 0$, and we prove that $C_1(t+1) \geq 0$ by assuming that $C_1(t) \geq 0$. Firstly, we define that

$$C_1^*(t) = \frac{F_3^{(t)}}{Kp(t)F_1^{(t)}}.$$

Then based on the definition of $C_1^*(t)$, the iterative rule for $C_1(t)$ can be re-written as

$$C_1(t+1) = C_1(t) + \frac{\eta D F_3^{(t)}}{Kp(t)}\left(1 - \frac{C_1(t)}{C_1^*(t)}\right).$$

From the iterative rule above, it is clear that if $C_1(t) \leq C_1^*(t)$, then $C_1(t+1) \geq C_1(t)$, and $C_1(t+1) < C_1(t)$ if $C_1(t) > C_1^*(t)$. Notice that Lemma D.3 immediately implies that $1 \leq C_1^*(t) \leq \sqrt{\frac{D}{K}}$. We can conclude that once $C_1(t)$ surpasses $\sqrt{\frac{D}{K}}$, then it starts to decrease until it becomes lower than $\sqrt{\frac{D}{K}}$. Therefore, we have

$$C_1(t) \leq \sqrt{\frac{D}{K}} + D\eta \frac{F_3^{(t)}}{Kp(t)} \leq \sqrt{\frac{D}{K}} + D\eta \frac{Kp(t)^2 + \sqrt{\frac{K}{D-K}}p(t)\big(1 - Kp(t)\big)}{Kp(t)}$$

$$= \sqrt{\frac{D}{K}} + \eta D\left(p(t) + \sqrt{\frac{1}{K(D-K)}}\big(1 - Kp(t)\big)\right) \leq \sqrt{\frac{D}{K}} + \frac{2\eta D}{K} \leq \sqrt{\frac{D+1}{K}},$$

where the second inequality holds as $F_3^{(t)} \leq Kp(t)^2 + \sqrt{\frac{K}{D-K}}p(t)\big(1 - Kp(t)\big)$ demonstrated in Lemma F.5, and the last inequality holds by the condition of $\eta$ that $\eta \leq \mathcal{O}(D^{-5/2})$ in Theorem 3.1. Now we prove that $C_1(t+1) \geq 0$ holds for both cases: $C_1(t) \leq C_1^*(t)$ and $C_1(t) > C_1^*(t)$. If $C_1(t) \leq C_1^*(t)$, then it is straightforward that $C_1(t+1) \geq C_1(t) \geq 0$. If $C_1(t) > C_1^*(t)$, then we have

$$C_1(t+1) \geq C_1(t) - \eta D C_1(t)F_1(t)$$

$$\geq C_1(t) - \frac{D\eta C_1(t)\big(DKp(t)^2 - 2Kp(t) + 1\big)}{D - K}$$

$$\geq 1 - D\eta\sqrt{\frac{D+1}{K}}\frac{\big(DKp(t)^2 - 2Kp(t) + 1\big)}{D - K}$$

$$\geq 1 - \eta\sqrt{\frac{D+1}{K}}\frac{D}{K} \geq \frac{1}{2}.$$

Here, the second inequality holds as $F_1^{(t)} \leq \frac{DKp(t)^2 - 2Kp(t) + 1}{D - K}$ implied by Lemma F.1. The second inequality holds by $C_1(t) \leq \sqrt{\frac{D+1}{K}}$, and $C_1(t) \geq C_1^*(t) \geq 1$. The third inequality holds as $DKp(t)^2 - 2Kp(t) + 1 \leq \frac{D-K}{K}$ when $\frac{1}{D} \leq p(t) \leq \frac{1}{K}$. The last inequality holds by the condition of $\eta$ that $\eta \leq \mathcal{O}(D^{-5/2})$ in Theorem 3.1. This finishes the proof that $C_1(t)$ is always non-negative. $\square$

In the proof above, we introduce the definition of a proxy $C_1^*(t) = \frac{F_3^{(t)}}{Kp(t)F_1^{(t)}}$, and utilize this proxy to provide an upper bound for $C_1(t)$. In fact, $C_1^*(t)$ can be regarded as a "stationary point" of the iterative rule for $C_1(t)$ in (D.12). Inspired by the proof techniques proposed in Wang et al. (2024), we introduce the following lemma, which offers a more refined upper bound for $C_1(t)$. We demonstrate this lemma prior to Lemma D.8, as its conclusion will be utilized in the proof of Lemma D.8.

**Lemma D.5.** Suppose all conditions of Theorem 3.1 hold, and $C_1(t)$, $C_1^*(t)$ are as defined in Lemma D.4. In addition, define that

$$A(t) = \begin{cases} Kp(t)^2 & \text{if } \sigma(\cdot) \text{ is identity map;} \\ \frac{Kp(t)^2}{4} + \frac{Kp(t)^2}{2\pi}\arctan\left(\frac{\sqrt{K(D-K)}p(t)}{1-Kp(t)}\right) & \text{if } \sigma(\cdot) \text{ is ReLU activation function;} \\ \frac{(1+\kappa)^2 Kp(t)^2}{4} + \frac{(1-\kappa)^2 Kp(t)^2}{2\pi}\arctan\left(\frac{\sqrt{K(D-K)}p(t)}{1-Kp(t)}\right) & \text{if } \sigma(\cdot) \text{ is Leaky ReLU activation function,} \end{cases}$$

$$(\text{D.15})$$

and

$$B(t) = \begin{cases} 0 & \text{if } \sigma(\cdot) \text{ is identity map;} \\ \frac{1}{2\pi}\sqrt{\frac{K}{D-K}}p(t)\big(1-Kp(t)\big) & \text{if } \sigma(\cdot) \text{ is ReLU activation function;} \\ \frac{(1-\kappa)^2}{2\pi}\sqrt{\frac{K}{D-K}}p(t)\big(1-Kp(t)\big) & \text{if } \sigma(\cdot) \text{ is Leaky ReLU activation function.} \end{cases} \tag{D.16}$$

Then it always holds that

$$C_1(t) \le \left(1 + \frac{4A(t)}{5\big(A(t)+B(t)\big)}\frac{1-Kp(t)}{Kp(t)\big(Dp(t)-1\big)}\right)C_1^*(t), \tag{D.17}$$

Specifically, when $p(t) \le \frac{1}{2\sqrt{\pi DK}}$, this upper bound can be tighter as $C_1(t) \le C_1^*(t)$.

**Remark D.6.** In fact, by checking the definition of $F_3^{(t)}$ in Lemma D.2 and its calculated value in Lemma F.5, we can conclude that $F_3^{(t)} = A(t) + B(t)$.

In addition, we also have the following lemma, which provides further calculation results when the conclusion of Lemma D.5 holds. This result will be utilized in the proof of Lemma D.8.

**Lemma D.7.** Suppose $C_1(t), C_1^*(t)$ as defined in Lemma D.4, and satisfying that

$$C_1(t) = \left(1 + \alpha\frac{A(t)}{A(t)+B(t)}\frac{1-Kp(t)}{Kp(t)\big(Dp(t)-1\big)}\right)C_1^*(t)$$

for some scalar $\alpha < 1$, then it holds that

$$\frac{F_4^{(t)}}{p(t)} - F_3^{(t)} - KC_1(t)\Big(F_{2,1}^{(t)} + p(t)F_1^{(t)}\Big) = \frac{\big(1-Kp(t)\big)^2}{Kp(t)\big(DKp(t)^2-2Kp(t)+1\big)}(1-\alpha)A(t);$$

$$\frac{F_3^{(t)}}{Kp(t)} - \frac{(D-K)F_5^{(t)}}{Kp(t)\big(1-Kp(t)\big)} - C_1(t)\left(F_1^{(t)} - \frac{(D-K)F_{2,2}^{(t)}}{1-Kp(t)}\right) = \frac{1-Kp(t)}{Kp(t)\big(DKp(t)^2-2Kp(t)+1\big)}(1-\alpha)A(t).$$

*Proof of Lemma D.7.* We prove this lemma when $\sigma(\cdot)$ is the identity map, ReLU activation function, and Leaky ReLU activation function, respectively. When $\sigma(\cdot)$ is the identity map, utilizing Lemma F.1, Lemma F.2, Lemma F.5, Lemma F.6, and Lemma F.7, we can obtain that

$$F_1^{(t)} = Kp(t)^2 + \frac{\big(1-Kp(t)\big)^2}{D-K}; \quad F_{2,1}^{(t)} = p(t)^2; \quad F_{2,2}^{(t)} = \frac{\big(1-Kp(t)\big)^2}{(D-K)^2};$$

$$F_3^{(t)} = Kp(t)^2; \quad F_4^{(t)} = p(t)^2; \quad F_5^{(t)} = 0. \tag{D.18}$$

Then combined with the definition of $A(t), B(t)$ in Lemma D.5, we can derive that

$$\frac{F_4^{(t)}}{p(t)} - F_3^{(t)} - KC_1(t)\Big(F_{2,1}^{(t)} + p(t)F_1^{(t)}\Big)$$

$$= \frac{F_4^{(t)}}{p(t)} - F_3^{(t)} - \left(1 + \alpha\frac{1-Kp(t)}{Kp(t)\big(Dp(t)-1\big)}\right)KC_1^*(t)\Big(F_{2,1}^{(t)} + p(t)F_1^{(t)}\Big)$$

$$= \frac{1-Kp(t)}{Kp(t)}A(t) - \left(1 + \frac{\alpha A(t)}{A(t)+B(t)}\frac{1-Kp(t)}{Kp(t)\big(Dp(t)-1\big)}\right)\frac{\big(1-Kp(t)\big)\big(Dp(t)-1\big)}{DKp(t)^2-2Kp(t)+1}A(t)$$

$$= \frac{\big(1-Kp(t)\big)^2}{Kp(t)\big(DKp(t)^2-2Kp(t)+1\big)}(1-\alpha)A(t).$$

where the first inequality holds by applying $C_1(t) = \big(1 + \alpha\frac{A(t)}{A(t)+B(t)}\frac{1-Kp(t)}{Kp(t)(Dp(t)-1)}\big)C_1^*(t)$ and $B(t) = 0$, the second inequality holds by applying the definition of $C_1^*(t)$ and the calculation results illustrated in (D.18). Similarly, we can also derive that

$$\frac{F_3^{(t)}}{Kp(t)} - \frac{(D-K)F_5^{(t)}}{Kp(t)\big(1-Kp(t)\big)} - C_1(t)\left(F_1^{(t)} - \frac{(D-K)F_{2,2}^{(t)}}{1-Kp(t)}\right)$$

$$= \frac{A(t)}{Kp(t)} - \left(1 + \alpha \frac{1 - Kp(t)}{Kp(t)\big(Dp(t) - 1\big)}\right) \frac{Dp(t) - 1}{DKp(t)^2 - 2Kp(t) + 1} A(t)$$

$$= \frac{1 - Kp(t)}{Kp(t)\big(DKp(t)^2 - 2Kp(t) + 1\big)}(1 - \alpha)A(t).$$

This finishes the proof when $\sigma(\cdot)$ is identity map. When $\sigma(\cdot)$ is the ReLU activation function, utilizing Lemma F.1, Lemma F.2, Lemma F.5, Lemma F.6, and Lemma F.7, we can obtain that

$$F_1^{(t)} = \frac{Kp(t)^2}{2} + \frac{\big(1 - Kp(t)\big)^2}{2(D - K)}; \quad F_{2,1}^{(t)} = \frac{p(t)^2}{2}; \quad F_{2,2}^{(t)} = \frac{\big(1 - Kp(t)\big)^2}{2(D - K)^2};$$

$$F_3^{(t)} = \frac{Kp(t)^2}{4} + \frac{Kp(t)^2}{2\pi} \arctan\left(\frac{\sqrt{K(D - K)}p(t)}{1 - Kp(t)}\right) + \frac{1}{2\pi}\sqrt{\frac{K}{D - K}}p(t)\big(1 - Kp(t)\big) = A(t) + B(t);$$

$$F_4^{(t)} = \frac{p(t)^2}{4} + \frac{p(t)^2}{2\pi} \arctan\left(\frac{\sqrt{K(D - K)}p(t)}{1 - Kp(t)}\right) + \frac{\sqrt{K(D - K)}p(t)^3\big(1 - Kp(t)\big)}{2\pi(DKp(t)^2 - 2Kp(t) + 1)}$$

$$= A(t) + \frac{(D - K)p(t)^2}{DKp(t)^2 - 2Kp(t) + 1}B(t);$$

$$F_5^{(t)} = \frac{p(t)\big(1 - Kp(t)\big)^3}{2\pi(D - K)\big(DKp(t)^2 - 2Kp(t) + 1\big)}\sqrt{\frac{K}{D - K}}. \tag{D.19}$$

Then combined with the definition of $A(t)$, $B(t)$ in Lemma D.5, we can derive that

$$\frac{F_4^{(t)}}{p(t)} - F_3^{(t)} - KC_1(t)\Big(F_{2,1}^{(t)} + p(t)F_1^{(t)}\Big)$$

$$= \frac{F_4^{(t)}}{p(t)} - F_3^{(t)} - \left(1 + \frac{\alpha A(t)}{A(t) + B(t)}\frac{1 - Kp(t)}{Kp(t)\big(Dp(t) - 1\big)}\right)\frac{C_1^*(t)Kp(t)\big(1 - Kp(t)\big)\big(Dp(t) - 1\big)}{2(D - K)}$$

$$= \frac{F_4^{(t)}}{p(t)} - F_3^{(t)} - \left(1 + \frac{\alpha A(t)}{A(t) + B(t)}\frac{1 - Kp(t)}{Kp(t)\big(Dp(t) - 1\big)}\right)\frac{F_3^{(t)}\big(1 - Kp(t)\big)\big(Dp(t) - 1\big)}{DKp(t)^2 - 2Kp(t) + 1}$$

$$= \frac{1 - Kp(t)}{Kp(t)}A(t) - \left(1 + \frac{\alpha A(t)}{A(t) + B(t)}\frac{1 - Kp(t)}{Kp(t)\big(Dp(t) - 1\big)}\right)\frac{\big(1 - Kp(t)\big)\big(Dp(t) - 1\big)}{DKp(t)^2 - 2Kp(t) + 1}A(t)$$

$$+ \frac{(1 - Kp(t))\big(Dp(t) - 1\big)}{DKp(t)^2 - 2Kp(t) + 1}B(t) - \left(1 + \frac{\alpha A(t)}{A(t) + B(t)}\frac{1 - Kp(t)}{Kp(t)\big(Dp(t) - 1\big)}\right)\frac{(1 - Kp(t))\big(Dp(t) - 1\big)}{DKp(t)^2 - 2Kp(t) + 1}B(t)$$

$$= \frac{\big(1 - Kp(t)\big)^2}{Kp(t)\big(DKp(t)^2 - 2Kp(t) + 1\big)}\left(\left(1 - \frac{\alpha A(t)}{A(t) + B(t)}\right)A(t) - \frac{\alpha A(t)B(t)}{A(t) + B(t)}\right)$$

$$= \frac{\big(1 - Kp(t)\big)^2}{Kp(t)\big(DKp(t)^2 - 2Kp(t) + 1\big)}(1 - \alpha)A(t).$$

Similarly, we also have

$$\frac{F_3^{(t)}}{Kp(t)} - \frac{(D - K)F_5^{(t)}}{Kp(t)\big(1 - Kp(t)\big)} - C_1(t)\left(F_1^{(t)} - \frac{(D - K)F_{2,2}^{(t)}}{1 - Kp(t)}\right)$$

$$= \frac{F_3^{(t)}}{Kp(t)} - \frac{1 - Kp(t)}{Kp(t)\big(DKp(t)^2 - 2Kp(t) + 1\big)}B(t)$$

$$- \left(1 + \frac{\alpha A(t)}{A(t) + B(t)}\frac{1 - Kp(t)}{Kp(t)\big(Dp(t) - 1\big)}\right)\frac{F_1^{(t)}\big(Dp(t) - 1\big)}{DKp(t)^2 - 2Kp(t) + 1}$$

$$= \left(\frac{1}{Kp(t)} - \left(1 + \frac{\alpha A(t)}{A(t) + B(t)}\frac{1 - Kp(t)}{Kp(t)\big(Dp(t) - 1\big)}\right)\frac{Dp(t) - 1}{DKp(t)^2 - 2Kp(t) + 1}\right)A(t)$$

$$+ \left[\frac{1}{Kp(t)} - \frac{1 - Kp(t)}{Kp(t)\big(DKp(t)^2 - 2Kp(t) + 1\big)}\right]$$

$$-\left(1+\frac{\alpha A(t)}{A(t)+B(t)}\frac{1-Kp(t)}{Kp(t)\big(Dp(t)-1\big)}\right)\frac{Dp(t)-1}{DKp(t)^2-2Kp(t)+1}\Bigg]B(t)$$

$$=\frac{1-Kp(t)}{Kp(t)\big(DKp(t)^2-2Kp(t)+1\big)}\left(\left(1-\frac{\alpha A(t)}{A(t)+B(t)}\right)A(t)-\frac{\alpha A(t)B(t)}{A(t)+B(t)}\right)$$

$$=\frac{1-Kp(t)}{Kp(t)\big(DKp(t)^2-2Kp(t)+1\big)}(1-\alpha)A(t).$$

This completes the proof of the scenario that $\sigma(\cdot)$ is ReLU activation function. For the case that $\sigma(\cdot)$ is the Leaky ReLU activation function, utilizing Lemma F.1, Lemma F.2, Lemma F.5, Lemma F.6, and Lemma F.7, we can obtain that

$$F_1^{(t)}=\frac{(1+\kappa^2)Kp(t)^2}{2}+\frac{(1+\kappa^2)\big(1-Kp(t)\big)^2}{2(D-K)};$$

$$F_{2,1}^{(t)}=\frac{(1+\kappa^2)p(t)^2}{2};\quad F_{2,2}^{(t)}=\frac{(1+\kappa^2)\big(1-Kp(t)\big)^2}{2(D-K)^2};$$

$$F_3^{(t)}=\frac{(1+\kappa)^2Kp(t)^2}{4}+\frac{(1-\kappa)^2Kp(t)^2}{2\pi}\arctan\left(\frac{\sqrt{K(D-K)}p(t)}{1-Kp(t)}\right)$$

$$+\frac{(1-\kappa)^2}{2\pi}\sqrt{\frac{K}{D-K}}p(t)\big(1-Kp(t)\big)=A(t)+B(t);$$

$$F_4^{(t)}=\frac{(1+\kappa)^2p(t)^2}{4}+\frac{(1-\kappa)^2p(t)^2}{2\pi}\arctan\left(\frac{\sqrt{K(D-K)}p(t)}{1-Kp(t)}\right)$$

$$+\frac{(1-\kappa)^2\sqrt{K(D-K)}p(t)^3\big(1-Kp(t)\big)}{2\pi(DKp(t)^2-2Kp(t)+1)}=A(t)+\frac{(D-K)p(t)^2}{DKp(t)^2-2Kp(t)+1}B(t);$$

$$F_5^{(t)}=\frac{(1-\kappa)^2p(t)\big(1-Kp(t)\big)^3}{2\pi(D-K)\big(DKp(t)^2-2Kp(t)+1\big)}\sqrt{\frac{K}{D-K}}.\tag{D.20}$$

Then the remaining proof is entirely identical to that of the ReLU activation function, when replacing the values of these terms demonstrated in (D.20). □

Based on the conclusion of Lemma D.5 and Lemma D.7, we are now prepared to prove Lemma D.8. We will address the proof of Lemma D.5 after completing the proof of Lemma D.8.

**Lemma D.8.** Under the same conditions of Theorem 3.1, there exist time dependent non-negative, monotonically increasing scalars $C_2(t)$ and $C_3(t)$, such that

$$\mathbf{W}_{KQ}^{(t)}=C_2(t)\sum_{i=1}^{D}\sum_{i_1\in G^i}\mathbf{p}_{i_1}\mathbf{p}_i^\top-C_3(t)\sum_{i=1}^{D}\sum_{i_1\notin G^i}\mathbf{p}_{i_1}\mathbf{p}_i^\top.\tag{D.21}$$

Due to the specific pattern of $\mathbf{W}_{KQ}^{(t)}$ demonstrated in (D.21), there exist a time dependent scalar $p(t)$, such that $\mathbf{S}_{i_1,i}^{(t)}=p(t)$ for all $i\in[D]$ and $i_1\in G^i$. Otherwise, $\mathbf{S}_{i_1,i}^{(t)}=\frac{1-Kp(t)}{D-K}$. Additionally, $\frac{1}{D}\le p(t)\le\frac{1}{K}$ and $p(t)$ is monotonically increasing. Based on the definition of $p(t)$, $C_2(t)$ and $C_3(t)$ have the following iterative rules

$$C_2(t+1)=C_2(t)+\eta\frac{C_1(t)M}{\sqrt{D}}\left(\frac{1}{K}\left(\frac{F_4^{(t)}}{p(t)}-F_3^{(t)}\right)-C_1(t)\left(F_{2,1}^{(t)}+p(t)F_1^{(t)}\right)\right);\tag{D.22}$$

$$C_3(t+1)=C_3(t)-\eta\frac{C_1(t)M(1-Kp(t))}{\sqrt{D}(D-K)}\left(\left(\frac{F_3^{(t)}}{Kp(t)}-\frac{(D-K)F_5^{(t)}}{Kp(t)\big(1-Kp(t)\big)}\right)-C_1(t)\left(F_1^{(t)}-\frac{(D-K)F_{2,2}^{(t)}}{1-Kp(t)}\right)\right).\tag{D.23}$$

In addition, based on all these definitions, the coefficients $C_1(t)$, $C_2(t)$, and $C_3(t)$ are essentially minimizing the following loss function by gradient descent

$$\widetilde{\mathcal{L}}(C_1,C_2,C_3)=\frac{c_\sigma D\|\mathbf{V}^*\|_F^2}{2(D-K)}\left[K(D-K)\left(\frac{1}{K}-C_1p\right)^2+C_1^2\big(1-Kp\big)^2\right]-D\|\mathbf{V}^*\|_F^2F_6(C_1,p).$$

where $c_\sigma$ is an absolute constant such that $c_\sigma = \mathbb{1}_{\{\sigma(\cdot) \text{ is identity map}\}} + \frac{1}{2}\mathbb{1}_{\{\sigma(\cdot) \text{ is ReLU}\}} + \frac{1+\kappa^2}{2}\mathbb{1}_{\{\sigma(\cdot) \text{ is Leaky ReLU}\}}$. In addition, $F_6(C_1, p)$ is defined as

$$
F_6 = \begin{cases}
0; & \text{If } \sigma(\cdot) \text{ is identity map} \\[2mm]
pC_1^2\left(Kp\left(\frac{1}{\pi}\arctan\left(\frac{p\sqrt{K(D-K)}}{1-Kp}\right) - \frac{1}{2}\right) + \frac{(1-Kp)\sqrt{K}}{\pi\sqrt{D-K}}\right); & \text{If } \sigma(\cdot) \text{ is ReLU activation} \\[2mm]
(1-\kappa)^2 pC_1^2\left(Kp\left(\frac{1}{\pi}\arctan\left(\frac{p\sqrt{K(D-K)}}{1-Kp}\right) - \frac{1}{2}\right) + \frac{(1-Kp)\sqrt{K}}{\pi\sqrt{D-K}}\right). & \text{If } \sigma(\cdot) \text{ is Leaky ReLU activation}
\end{cases}
$$

*Proof of Lemma D.8.* Similarly, it can be easily verified that the initialization $\mathbf{W}_{KQ}^{(0)} = \mathbf{0}_{D\times D}$ satisfies (D.21). Assuming it holds at the $t$-th iteration, we aim to prove that it continues to hold at the $t+1$-th iteration. To do this, it suffices to show that

$$
\nabla_{\mathbf{W}_{KQ}}\mathcal{L}(\mathbf{W}_V^{(t)}; \mathbf{W}_{KQ}^{(t)}) = -c_2(t)\sum_{i=1}^D \sum_{i_1 \in G^i} \mathbf{p}_{i_1}\mathbf{p}_i^\top + c_3(t)\sum_{i=1}^D \sum_{i_1 \notin G^i} \mathbf{p}_{i_1}\mathbf{p}_i^\top, \tag{D.24}
$$

where $c_2(t)$ and $c_3(t)$ are two time-dependent non-positive scalars. By Lemma D.1, we have

$$
\sqrt{D}\nabla_{\mathbf{W}_{KQ}}\mathcal{L}(\mathbf{W}_V^{(t)}; \mathbf{W}_{KQ}^{(t)})
$$

$$
= -\sum_{m=1}^M \sum_{i=1}^D \mathbb{E}\left[\left[\left[f^*(\mathbf{X})\right]_{m,i} - \sigma\left(\sum_{i_1=1}^D \langle \mathbf{w}_{V,m}^{(t)}, \mathbf{x}_{i_1}\rangle \mathbf{S}_{i_1,i}^{(t)}\right)\right]\sigma'\left(\sum_{i_1=1}^D \langle \mathbf{w}_{V,m}^{(t)}, \mathbf{x}_{i_1}\rangle \mathbf{S}_{i_1,i}^{(t)}\right)\right.
$$

$$
\left. \cdot \sum_{i_1=1}^D \sum_{i_2=1}^D \langle \mathbf{w}_{V,m}^{(t)}, \mathbf{x}_{i_1}\rangle \mathbf{S}_{i_1,i}^{(t)}\mathbf{S}_{i_2,i}^{(t)}(\mathbf{p}_{i_1} - \mathbf{p}_{i_2})\mathbf{p}_i^\top\right]
$$

$$
= -\underbrace{\sum_{m=1}^M \sum_{i=1}^D \mathbb{E}\left[\left[f^*(\mathbf{X})\right]_{m,i}\sigma'\left(\sum_{i_1=1}^D \langle \mathbf{w}_{V,m}^{(t)}, \mathbf{x}_{i_1}\rangle \mathbf{S}_{i_1,i}^{(t)}\right)\sum_{i_1=1}^D \sum_{i_2=1}^D \langle \mathbf{w}_{V,m}^{(t)}, \mathbf{x}_{i_1}\rangle \mathbf{S}_{i_1,i}^{(t)}\mathbf{S}_{i_2,i}^{(t)}\mathbf{p}_{i_1}\mathbf{p}_i^\top\right]}_{I_3}
$$

$$
+ \underbrace{\sum_{m=1}^M \sum_{i=1}^D \mathbb{E}\left[\left[f^*(\mathbf{X})\right]_{m,i}\sigma'\left(\sum_{i_1=1}^D \langle \mathbf{w}_{V,m}^{(t)}, \mathbf{x}_{i_1}\rangle \mathbf{S}_{i_1,i}^{(t)}\right)\sum_{i_1=1}^D \sum_{i_2=1}^D \langle \mathbf{w}_{V,m}^{(t)}, \mathbf{x}_{i_1}\rangle \mathbf{S}_{i_1,i}^{(t)}\mathbf{S}_{i_2,i}^{(t)}\mathbf{p}_{i_2}\mathbf{p}_i^\top\right]}_{I_4}
$$

$$
+ \underbrace{\sum_{m=1}^M \sum_{i=1}^D \mathbb{E}\left[\sigma\left(\sum_{i_1=1}^D \langle \mathbf{w}_{V,m}^{(t)}, \mathbf{x}_{i_1}\rangle \mathbf{S}_{i_1,i}^{(t)}\right)\sigma'\left(\sum_{i_1=1}^D \langle \mathbf{w}_{V,m}^{(t)}, \mathbf{x}_{i_1}\rangle \mathbf{S}_{i_1,i}^{(t)}\right)\sum_{i_1=1}^D \sum_{i_2=1}^D \langle \mathbf{w}_{V,m}^{(t)}, \mathbf{x}_{i_1}\rangle \mathbf{S}_{i_1,i}^{(t)}\mathbf{S}_{i_2,i}^{(t)}\mathbf{p}_{i_1}\mathbf{p}_i^\top\right]}_{I_5}
$$

$$
- \underbrace{\sum_{m=1}^M \sum_{i=1}^D \mathbb{E}\left[\sigma\left(\sum_{i_1=1}^D \langle \mathbf{w}_{V,m}^{(t)}, \mathbf{x}_{i_1}\rangle \mathbf{S}_{i_1,i}^{(t)}\right)\sigma'\left(\sum_{i_1=1}^D \langle \mathbf{w}_{V,m}^{(t)}, \mathbf{x}_{i_1}\rangle \mathbf{S}_{i_1,i}^{(t)}\right)\sum_{i_1=1}^D \sum_{i_2=1}^D \langle \mathbf{w}_{V,m}^{(t)}, \mathbf{x}_{i_1}\rangle \mathbf{S}_{i_1,i}^{(t)}\mathbf{S}_{i_2,i}^{(t)}\mathbf{p}_{i_2}\mathbf{p}_i^\top\right]}_{I_6}.
$$
$$\tag{D.25}$$

In the next, we discuss the value of $I_3$, $I_4$, $I_5$, and $I_6$ respectively. For $I_3$, it can be calculated as

$$
I_3 = \frac{1}{K}\sum_{m=1}^M \sum_{i=1}^D \mathbb{E}\left[\sigma\left(\sum_{i_1 \in G^i} \langle \mathbf{v}_m^*, \mathbf{x}_{i_1}\rangle\right)\sigma'\left(\sum_{i_1=1}^D \langle \mathbf{w}_{V,m}^{(t)}, \mathbf{x}_{i_1}\rangle \mathbf{S}_{i_1,i}^{(t)}\right)\sum_{i'=1}^D \langle \mathbf{w}_{V,m}^{(t)}, \mathbf{x}_{i'}\rangle \mathbf{S}_{i',i}^{(t)}\mathbf{p}_{i'}\mathbf{p}_i^\top \sum_{i_2=1}^D \mathbf{S}_{i_2,i}^{(t)}\right]
$$

$$
= \frac{C_1(t)}{Kp(t)}\sum_{m=1}^M \sum_{i=1}^D \sum_{i' \in G^i} \mathbb{E}\left[\sigma\left(\langle \mathbf{v}_m^*, \mathbf{x}_{i'}\rangle p(t) + \sum_{i_1 \in G^i, i_1 \neq i'} \langle \mathbf{v}_m^*, \mathbf{x}_{i_1}\rangle p(t)\right)\right.
$$

$$\cdot \sigma'\bigg(\langle \mathbf{v}_m^*, \mathbf{x}_{i'}\rangle p(t) + \sum_{i_1 \in G^i, i_1 \neq i'} \langle \mathbf{v}_m^*, \mathbf{x}_{i_1}\rangle p(t) + \sum_{i_1 \notin G^i} \langle \mathbf{v}_m^*, \mathbf{x}_{i_1}\rangle \frac{1 - Kp(t)}{D - K}\bigg) \langle \mathbf{v}_m^*, \mathbf{x}_{i'}\rangle p(t)\bigg] \mathbf{p}_{i'} \mathbf{p}_i^\top$$

$$+ \frac{C_1(t)}{Kp(t)} \sum_{m=1}^{M} \sum_{i=1}^{D} \sum_{i' \notin G^i} \mathbb{E}\bigg[\sigma\bigg(\sum_{i_1 \in G^i} \langle \mathbf{v}_m^*, \mathbf{x}_{i_1}\rangle p(t)\bigg)$$

$$\cdot \sigma'\bigg(\langle \mathbf{v}_m^*, \mathbf{x}_{i'}\rangle \frac{1 - Kp(t)}{D - K} + \sum_{i_1 \in G^i} \langle \mathbf{v}_m^*, \mathbf{x}_{i_1}\rangle p(t) + \sum_{i_1 \notin G^i, i_1 \neq i'} \langle \mathbf{v}_m^*, \mathbf{x}_{i_1}\rangle \frac{1 - Kp(t)}{D - K}\bigg) \langle \mathbf{v}_m^*, \mathbf{x}_{i'}\rangle p(t)\bigg] \mathbf{p}_{i'} \mathbf{p}_i^\top$$

$$= \frac{C_1(t)M}{Kp(t)} F_4\bigg(p(t)^2, (K-1)p(t)^2, \frac{(1 - Kp(t))^2}{D - K}\bigg) \sum_{i=1}^{D} \sum_{i' \in G^i} \mathbf{p}_{i'} \mathbf{p}_i^\top$$

$$+ \frac{C_1(t)M}{Kp(t)} F_5\bigg(Kp(t)^2, \frac{(1 - Kp(t))^2}{(D - K)^2}, \frac{(D - K - 1)(1 - Kp(t))^2}{(D - K)^2}\bigg) \sum_{i=1}^{D} \sum_{i' \notin G^i} \mathbf{p}_{i'} \mathbf{p}_i^\top$$

$$= \frac{C_1(t)M F_4^{(t)}}{Kp(t)} \sum_{i=1}^{D} \sum_{i' \in G^i} \mathbf{p}_{i'} \mathbf{p}_i^\top + \frac{C_1(t)M F_5^{(t)}}{Kp(t)} \sum_{i=1}^{D} \sum_{i' \notin G^i} \mathbf{p}_{i'} \mathbf{p}_i^\top$$

Notice that $\langle \mathbf{v}_m^*, \mathbf{x}_{i'}\rangle p(t) \sim \mathcal{N}(0, p(t)^2)$, $\sum_{i_1 \in G^i, i_1 \neq i'} \langle \mathbf{v}_m^*, \mathbf{x}_{i_1}\rangle p(t) \sim \mathcal{N}(0, (K - 1)p(t)^2)$, and $\sum_{i_1 \notin G^i} \langle \mathbf{v}_m^*, \mathbf{x}_{i_1}\rangle \frac{1 - Kp(t)}{D - K} \sim \mathcal{N}(0, \frac{(1 - Kp(t))^2}{D - K})$ are three independent Gaussian random variables. Consequently, the first term in the penultimate equality is derived by the definition of $F_4(a, b, c)$ in (D.5). Similarly, $\sum_{i_1 \in G^i} \langle \mathbf{v}_m^*, \mathbf{x}_{i_1}\rangle p(t) \sim \mathcal{N}(0, Kp(t)^2)$, $\langle \mathbf{v}_m^*, \mathbf{x}_{i'}\rangle \frac{1 - Kp(t)}{D - K} \sim \mathcal{N}(0, \frac{(1 - Kp(t))^2}{(D - K)^2})$, and $\sum_{i_1 \notin G^i, i_1 \neq i'} \langle \mathbf{v}_m^*, \mathbf{x}_{i_1}\rangle \frac{1 - Kp(t)}{D - K} \sim \mathcal{N}(0, \frac{(D - K - 1)(1 - Kp(t))^2}{(D - K)^2})$ are three independent Gaussian random variables. Therefore, the second term in the penultimate equality is derived by the definition of $F_5(a, b, c)$ in (D.6). Similarly, for $I_4$, we can calculate it as

$$I_4 = \frac{1}{K} \sum_{m=1}^{M} \sum_{i=1}^{D} \mathbb{E}\bigg[\sigma\bigg(\sum_{i_1 \in G^i} \langle \mathbf{v}_m^*, \mathbf{x}_{i_1}\rangle\bigg) \sigma'\bigg(\sum_{i_1=1}^{D} \langle \mathbf{w}_{V,m}^{(t)}, \mathbf{x}_{i_1}\rangle \mathbf{S}_{i_1,i}^{(t)}\bigg) \bigg(\sum_{i_1=1}^{D} \langle \mathbf{w}_{V,m}^{(t)}, \mathbf{x}_{i_1}\rangle \mathbf{S}_{i_1,i}^{(t)}\bigg)\bigg] \sum_{i_2=1}^{D} \mathbf{S}_{i_2,i}^{(t)} \mathbf{p}_{i_2} \mathbf{p}_i^\top$$

$$= \frac{C_1(t)}{Kp(t)} \sum_{m=1}^{M} \sum_{i=1}^{D} \mathbb{E}\bigg[\sigma\bigg(\sum_{i_1 \in G^i} \langle \mathbf{v}_m^*, \mathbf{x}_{i_1}\rangle p(t)\bigg) \sigma'\bigg(\sum_{i_1 \in G^i} \langle \mathbf{v}_m^*, \mathbf{x}_{i_1}\rangle p(t) + \sum_{i_1 \notin G^i} \langle \mathbf{v}_m^*, \mathbf{x}_{i_1}\rangle \frac{1 - Kp(t)}{D - K}\bigg)$$

$$\cdot \bigg(\sum_{i_1 \in G^i} \langle \mathbf{v}_m^*, \mathbf{x}_{i_1}\rangle p(t) + \sum_{i_1 \notin G^i} \langle \mathbf{v}_m^*, \mathbf{x}_{i_1}\rangle \frac{1 - Kp(t)}{D - K}\bigg)\bigg] \sum_{i_2=1}^{D} \mathbf{S}_{i_2,i}^{(t)} \mathbf{p}_{i_2} \mathbf{p}_i^\top$$

$$= \frac{C_1(t)M}{K} F_3\bigg(Kp(t)^2, \frac{(1 - Kp(t))^2}{D - K}\bigg) \sum_{i=1}^{D} \sum_{i_2 \in G^i} \mathbf{p}_{i_2} \mathbf{p}_i^\top$$

$$+ \frac{C_1(t)M(1 - Kp(t))}{(D - K)Kp(t)} F_3\bigg(Kp(t)^2, \frac{(1 - Kp(t))^2}{D - K}\bigg) \sum_{i=1}^{D} \sum_{i_2 \notin G^i} \mathbf{p}_{i_2} \mathbf{p}_i^\top$$

$$= \frac{C_1(t)M F_3^{(t)}}{K} \sum_{i=1}^{D} \sum_{i_2 \in G^i} \mathbf{p}_{i_2} \mathbf{p}_i^\top + \frac{C_1(t)M(1 - Kp(t)) F_3^{(t)}}{(D - K)Kp(t)} \sum_{i=1}^{D} \sum_{i_2 \notin G^i} \mathbf{p}_{i_2} \mathbf{p}_i^\top$$

The penultimate equality holds by the definition of $F_3(a, b)$ in (D.4), as $\sum_{i_1 \in G^i} \langle \mathbf{v}_m^*, \mathbf{x}_{i_1}\rangle p(t) \sim \mathcal{N}(0, Kp(t)^2)$, and $\sum_{i_1 \notin G^i} \langle \mathbf{v}_m^*, \mathbf{x}_{i_1}\rangle \frac{1 - Kp(t)}{D - K} \sim \mathcal{N}\big(0, \frac{(1 - Kp(t))^2}{D - K}\big)$ are two independent Gaussian random variables. Additionally, $I_5$ can be calculated as

$$I_5 = C_1(t)^2 \sum_{m=1}^{M} \sum_{i=1}^{D} \sum_{i'=1}^{D} \mathbb{E}\bigg[\sigma\bigg(\sum_{i_1=1}^{D} \langle \mathbf{v}_m^*, \mathbf{x}_{i_1}\rangle \mathbf{S}_{i_1,i}^{(t)}\bigg) \sigma'\bigg(\sum_{i_1=1}^{D} \langle \mathbf{v}_m^*, \mathbf{x}_{i_1}\rangle \mathbf{S}_{i_1,i}^{(t)}\bigg) \langle \mathbf{v}_m^*, \mathbf{x}_{i'}\rangle \mathbf{S}_{i',i}^{(t)}\bigg] \mathbf{p}_{i'} \mathbf{p}_i^\top \sum_{i_2=1}^{D} \mathbf{S}_{i_2,i}^{(t)}$$

$$= C_1(t)^2 \sum_{m=1}^{M} \sum_{i=1}^{D} \sum_{i' \in G^i} \mathbb{E}\bigg[\sigma\bigg(\sum_{i_1=1}^{D} \langle \mathbf{v}_m^*, \mathbf{x}_{i_1}\rangle \mathbf{S}_{i_1,i}^{(t)}\bigg) \sigma'\bigg(\sum_{i_1=1}^{D} \langle \mathbf{v}_m^*, \mathbf{x}_{i_1}\rangle \mathbf{S}_{i_1,i}^{(t)}\bigg) \langle \mathbf{v}_m^*, \mathbf{x}_{i'}\rangle p(t)\bigg] \mathbf{p}_{i'} \mathbf{p}_i^\top$$

$$+ C_1(t)^2 \sum_{m=1}^{M} \sum_{i=1}^{D} \sum_{i' \in G^i} \mathbb{E}\left[ \sigma\left( \sum_{i_1=1}^{D} \langle \mathbf{v}_m^*, \mathbf{x}_{i_1} \rangle \mathbf{S}_{i_1,i}^{(t)} \right) \sigma'\left( \sum_{i_1=1}^{D} \langle \mathbf{v}_m^*, \mathbf{x}_{i_1} \rangle \mathbf{S}_{i_1,i}^{(t)} \right) \langle \mathbf{v}_m^*, \mathbf{x}_{i'} \rangle \frac{1 - Kp(t)}{D - K} \right] \mathbf{p}_{i'} \mathbf{p}_i^\top$$

$$= MC_1(t)^2 F_2\left( p(t)^2, (K-1)p(t)^2 + \frac{(1 - Kp(t))^2}{D - K} \right) \sum_{i=1}^{D} \sum_{i' \in G^i} \mathbf{p}_{i'} \mathbf{p}_i^\top$$

$$+ MC_1(t)^2 F_2\left( \frac{(1 - Kp(t))^2}{(D - K)^2}, Kp(t)^2 + \frac{(D - K - 1)(1 - Kp(t))^2}{(D - K)^2} \right) \sum_{i=1}^{D} \sum_{i' \notin G^i} \mathbf{p}_{i'} \mathbf{p}_i^\top$$

$$= MC_1(t)^2 F_{2,1}^{(t)} \sum_{i=1}^{D} \sum_{i' \in G^i} \mathbf{p}_{i'} \mathbf{p}_i^\top + MC_1(t)^2 F_{2,2}^{(t)} \sum_{i=1}^{D} \sum_{i' \notin G^i} \mathbf{p}_{i'} \mathbf{p}_i^\top,$$

where the penultimate equality utilize the definition of $F_2(a, b)$ in (D.3). Similarly, for $I_6$, we have

$$I_6 = C_1(t)^2 \sum_{m=1}^{M} \sum_{i=1}^{D} \mathbb{E}\left[ \sigma\left( \sum_{i_1=1}^{D} \langle \mathbf{v}_m^*, \mathbf{x}_{i_1} \rangle \mathbf{S}_{i_1,i}^{(t)} \right) \sigma'\left( \sum_{i_1=1}^{D} \langle \mathbf{v}_m^*, \mathbf{x}_{i_1} \rangle \mathbf{S}_{i_1,i}^{(t)} \right) \left( \sum_{i_1=1}^{D} \langle \mathbf{v}_m^*, \mathbf{x}_{i_1} \rangle \mathbf{S}_{i_1,i}^{(t)} \right) \right] \sum_{i_2=1}^{D} \mathbf{S}_{i_2,i}^{(t)} \mathbf{p}_{i_2} \mathbf{p}_i^\top$$

$$= MC_1(t)^2 p(t) F_1^{(t)} \sum_{i=1}^{D} \sum_{i_2 \in G^i} \mathbf{p}_{i_2} \mathbf{p}_i^\top + MC_1(t)^2 F_1^{(t)} \frac{1 - Kp(t)}{D - K} \sum_{i=1}^{D} \sum_{i_2 \notin G^i} \mathbf{p}_{i_2} \mathbf{p}_i^\top.$$

Combining all these results of $I_3$, $I_4$, $I_5$, and $I_6$, and plugging them into (D.25), we obtain that

$$\nabla_{\mathbf{W}_{KQ}} \mathcal{L}(\mathbf{W}_V^{(t)}; \mathbf{W}_{KQ}^{(t)})$$

$$= -\frac{C_1(t)M}{\sqrt{D}}\left( \frac{1}{K}\left( \frac{F_4^{(t)}}{p(t)} - F_3^{(t)} \right) - C_1(t)\left( F_{2,1}^{(t)} + p(t) F_1^{(t)} \right) \right) \sum_{i=1}^{D} \sum_{i' \in G^i} \mathbf{p}_{i'} \mathbf{p}_i^\top$$

$$+ \frac{C_1(t)M(1 - Kp(t))}{(D - K)\sqrt{D}}\left( \left( \frac{F_3^{(t)}}{Kp(t)} - \frac{(D - K)F_5^{(t)}}{Kp(t)(1 - Kp(t))} \right) - C_1(t)\left( F_1^{(t)} - \frac{(D - K)F_{2,2}^{(t)}}{1 - Kp(t)} \right) \right) \sum_{i=1}^{D} \sum_{i' \notin G^i} \mathbf{p}_{i'} \mathbf{p}_i^\top$$

$$= -c_2(t) \sum_{i=1}^{D} \sum_{i' \in G^i} \mathbf{p}_{i'} \mathbf{p}_i^\top + c_3(t) \sum_{i=1}^{D} \sum_{i' \notin G^i} \mathbf{p}_{i'} \mathbf{p}_i^\top.$$

It remains to show that $c_2(t)$ and $c_3(t)$ are always non-negative. Notice that Lemma D.5 guarantee the assumption of Lemma D.7. By carefully compare the formulas and applying Lemma D.7, we can obtain that

$$\frac{K\sqrt{D}c_2(t)}{MC_1(t)} \geq \frac{(1 - Kp(t))^2}{5Kp(t)(DKp(t)^2 - 2Kp(t) + 1)} A(t) \geq 0.$$

Since we have proved that $C_1(t)$ is always non-negative in Lemma D.4, this result implies that $c_2(t) \geq 0$. Similarly, for $c_3(t)$, we have

$$\frac{\sqrt{D}(D - K)c_3(t)}{MC_1(t)(1 - Kp(t))} \geq \frac{1 - Kp(t)}{5Kp(t)(DKp(t)^2 - 2Kp(t) + 1)} A(t) \geq 0.$$

This proves that $c_3(t) \geq 0$, and we conclude that

$$C_2(t+1) = C_2(t) + \eta \frac{C_1(t)M}{\sqrt{D}}\left( \frac{1}{K}\left( \frac{F_4^{(t)}}{p(t)} - F_3^{(t)} \right) - C_1(t)\left( F_{2,1}^{(t)} + p(t) F_1^{(t)} \right) \right);$$

$$C_3(t+1) = C_3(t) - \eta \frac{C_1(t)M(1 - Kp(t))}{\sqrt{D}(D - K)}\left( \left( \frac{F_3^{(t)}}{Kp(t)} - \frac{(D - K)F_5^{(t)}}{Kp(t)(1 - Kp(t))} \right) - C_1(t)\left( F_1^{(t)} - \frac{(D - K)F_{2,2}^{(t)}}{1 - Kp(t)} \right) \right),$$

which completes the proof of (D.21), (D.22) and (D.23). It remains to prove the conclusions regarding $\mathbf{S}_{i_1,i}^{(t)}$ and $p(t)$. By the orthogonality among the positional encodings $\mathbf{p}_i$'s, it is straightforward

that for all $i, i_1 \in [D]$,

$$\mathbf{p}_{i_1}^{\top} \mathbf{W}_{KQ}^{(t)} \mathbf{p}_i = \begin{cases} C_2(t) & \text{if } i_1 \in G^i; \\ -C_3(t) & \text{if } i_1 \notin G^i. \end{cases}$$

Then by the definition of $\mathbf{S}^{(t)}$, when $i_1 \in G^i$

$$\mathbf{S}_{i_1,i}^{(t)} = \frac{\exp\left(\frac{\mathbf{p}_{i_1}^{\top} \mathbf{W}_{KQ}^{(t)} \mathbf{p}_i}{\sqrt{D}}\right)}{\sum_{i_2=1}^{D} \exp\left(\frac{\mathbf{p}_{i_2}^{\top} \mathbf{W}_{KQ}^{(t)} \mathbf{p}_i}{\sqrt{D}}\right)} = \frac{\exp\left(\frac{C_2(t)}{\sqrt{D}}\right)}{K \exp\left(\frac{C_2(t)}{\sqrt{D}}\right) + (D-K) \exp\left(-\frac{C_3(t)}{\sqrt{D}}\right)}$$

$$= \frac{1}{K + (D-K) \exp\left(-\frac{C_2(t)+C_3(t)}{\sqrt{D}}\right)} = p(t).$$

Since $C_2(t)$ and $C_3(t)$ are non-negative and monotonically increasing scalars, we immediately conclude that $\frac{1}{D} \leq p(t) \leq \frac{1}{K}$, and $p(t)$ is also monotonically increasing. Lastly, it remains to formulate excess loss into an expression of excess loss. By the parameter forms in Lemma D.2, we have

$$\widetilde{\mathcal{L}}(C_1, C_2, C_3) = \mathcal{L}(\mathbf{W}_V, \mathbf{W}_{KQ}) - \mathcal{L}_{\text{opt}}$$

$$= \mathbb{E}\left[\sum_{m=1}^{M} \sum_{i=1}^{D} \left(\sigma\left(\frac{\sum_{i' \in G^i} \langle \mathbf{v}_m^*, \mathbf{x}_{i'} \rangle}{K}\right) - \sigma\left(C_1 p \sum_{i' \in G^i} \langle \mathbf{v}_m^*, \mathbf{x}_{i'} \rangle + \frac{C_1(1-Kp)}{D-K} \sum_{i' \notin G^i} \langle \mathbf{v}_m^*, \mathbf{x}_{i'} \rangle\right)\right)^2\right]$$

$$= \sum_{m=1}^{M} \sum_{i=1}^{D} \underbrace{\mathbb{E}\left[\sigma\left(\frac{\sum_{i' \in G^i} \langle \mathbf{v}_m^*, \mathbf{x}_{i'} \rangle}{K}\right)^2\right]}_{I_7} + \sum_{m=1}^{M} \sum_{i=1}^{D} \underbrace{\mathbb{E}\left[\sigma\left(C_1 p \sum_{i' \in G^i} \langle \mathbf{v}_m^*, \mathbf{x}_{i'} \rangle + \frac{C_1(1-Kp)}{D-K} \sum_{i' \notin G^i} \langle \mathbf{v}_m^*, \mathbf{x}_{i'} \rangle\right)^2\right]}_{I_8}$$

$$- 2 \sum_{m=1}^{M} \sum_{i=1}^{D} \underbrace{\mathbb{E}\left[\sigma\left(\frac{\sum_{i' \in G^i} \langle \mathbf{v}_m^*, \mathbf{x}_{i'} \rangle}{K}\right) \sigma\left(C_1 p \sum_{i' \in G^i} \langle \mathbf{v}_m^*, \mathbf{x}_{i'} \rangle + \frac{C_1(1-Kp)}{D-K} \sum_{i' \notin G^i} \langle \mathbf{v}_m^*, \mathbf{x}_{i'} \rangle\right)\right]}_{I_9}.$$

For the term $I_7$ and $I_8$, by the fact that $\mathbb{E}[\sigma(x)^2] = c_\sigma a$ when $x \sim \mathcal{N}(0, a)$, we can directly calculate that

$$I_7 = \frac{c_\sigma \|\mathbf{v}_m^*\|_2^2}{K};$$

$$I_8 = c_\sigma K p^2 C_1^2 \|\mathbf{v}_m^*\|_2^2 + \frac{c_\sigma C_1^2 (1-Kp)^2 \|\mathbf{v}_m^*\|_2^2}{D-K}.$$

In addition, by utilizing the conclusions in Lemma F.8, we can conclude that

$$I_9 = c_\sigma C_1 p \|\mathbf{v}_m^*\|_2^2 - \frac{D \|\mathbf{v}_m^*\|_2^2 F_6(C_1, p)}{2}.$$

Plugging all these results, we completes the proof that

$$\widetilde{\mathcal{L}}(C_1, C_2, C_3) = \frac{c_\sigma D \|\mathbf{V}^*\|_F^2}{2(D-K)} \left[K(D-K)\left(\frac{1}{K} - C_1 p\right)^2 + C_1^2 \left(1 - Kp\right)^2\right] - D \|\mathbf{V}^*\|_F^2 F_6(C_1, p)$$

Now, we successfully prove all the conclusions of Lemma D.8. $\qquad\square$

Lastly, before we prove Lemma D.5, we first introduce and prove the following Lemma D.9, Lemma D.10, Lemma D.11, and Lemma D.12, which will be utilized for proof of Lemma D.5.

**Lemma D.9.** Under the same conditions as Theorem 3.1 and $p(t)$ as defined in Lemma D.2, it holds that

$$\frac{p(t)\left(1 - Kp(t)\right)}{2\sqrt{D}} \left(\Delta C_2(t) + \Delta C_3(t)\right) \leq \Delta p(t) \leq \frac{D^2 p(t)\left(1 - Kp(t)\right)}{\sqrt{D}(D^2 - 1)} \left(\Delta C_2(t) + \Delta C_3(t)\right);$$

$$\Delta C_2(t) + \Delta C_3(t) \leq \eta \frac{MD}{K^2(D-K)} \sqrt{\frac{D+1}{K}} \leq \frac{1}{D^2},$$

where $\Delta p(t) = p(t+1) - p(t)$, $\Delta C_2(t) = C_2(t+1) - C_2(t)$, and $\Delta C_3(t) = C_3(t+1) - C_3(t)$.

*Proof of Lemma D.9.* By the definition of $p(t)$ in Lemma D.2, it can be derived that

$$\Delta p(t) = p(t+1) - p(t) = \frac{1}{K + (D-K)\exp\left(-\frac{C_2(t+1)+C_3(t+1)}{\sqrt{D}}\right)} - \frac{1}{K + (D-K)\exp\left(-\frac{C_2(t)+C_3(t)}{\sqrt{D}}\right)}$$

$$\leq \frac{1}{K + (D-K)\exp\left(-\frac{C_2(t)+C_3(t)}{\sqrt{D}}\right)\left(1 - \frac{\Delta C_2(t)+\Delta C_3(t)}{\sqrt{D}}\right)} - \frac{1}{K + (D-K)\exp\left(-\frac{C_2(t)+C_3(t)}{\sqrt{D}}\right)}$$

$$= \frac{\left(\Delta C_2(t) + \Delta C_3(t)\right)(D-K)\exp\left(-\frac{C_2(t)+C_3(t)}{\sqrt{D}}\right)}{\sqrt{D}\left[K + (D-K)\exp\left(-\frac{C_2(t)+C_3(t)}{\sqrt{D}}\right)\left(1 - \frac{\Delta C_2(t)+\Delta C_3(t)}{\sqrt{D}}\right)\right]\left[K + (D-K)\exp\left(-\frac{C_2(t)+C_3(t)}{\sqrt{D}}\right)\right]}$$

$$\leq \frac{\Delta C_2(t) + \Delta C_3(t)}{\sqrt{D} - \Delta C_2(t) - \Delta C_3(t)} \frac{(D-K)\exp\left(-\frac{C_2(t)+C_3(t)}{\sqrt{D}}\right)}{\left[K + (D-K)\exp\left(-\frac{C_2(t)+C_3(t)}{\sqrt{D}}\right)\right]^2}$$

$$= \frac{\Delta C_2(t) + \Delta C_3(t)}{\sqrt{D} - \Delta C_2(t) - \Delta C_3(t)} p(t)\left(1 - Kp(t)\right)$$

Additionally, applying the update rules for $C_2(t)$ and $C_3(t)$ derived in Lemma D.8, along with a similar calculation to the one used in the proof of Lemma D.7, we obtain that

$$\Delta C_2(t) \leq \eta \frac{MC_1(t)}{K\sqrt{D}}\left(\frac{F_4^{(t)}}{p(t)} - F_3^{(t)}\right)$$

$$= \eta \frac{MC_1(t)}{K\sqrt{D}}\left(\left(\frac{1}{Kp(t)} - 1\right)A(t) + \left(\frac{(D-K)p(t)}{DKp(t)^2 - 2Kp(t) + 1} - 1\right)B(t)\right)$$

$$= \eta \frac{MC_1(t)}{K\sqrt{D}}\left(\frac{1 - Kp(t)}{Kp(t)}A(t) + \frac{\left(1 - Kp(t)\right)\left(Dp(t) - 1\right)}{Kp(t)\left(Dp(t) - 1\right) + 1 - Kp(t)}B(t)\right)$$

$$\leq \eta \frac{MC_1(t)}{K\sqrt{D}}\frac{1 - Kp(t)}{Kp(t)}F_3^{(t)} \leq \eta \frac{MC_1(t)}{K\sqrt{D}}\frac{1 - Kp(t)}{Kp(t)}\left(Kp(t)^2 + \frac{1}{2\pi}\sqrt{\frac{K}{D-K}}p(t)\left(1 - Kp(t)\right)\right)$$

$$\leq \eta \frac{M}{K^2}\sqrt{\frac{D+1}{DK}}.$$

Here, the penultimate inequality holds as $C_1(t) \leq \sqrt{\frac{D+1}{K}}$ and $\frac{1}{D} \leq p(t) \leq \frac{1}{K}$. Similarly, we can also derive that

$$\Delta C_3(t) \leq \eta \frac{MC_1(t)\left(1 - Kp(t)\right)}{\sqrt{D}(D-K)}\frac{F_3^{(t)}}{Kp(t)} \leq \eta \frac{M}{K(D-K)}\sqrt{\frac{D+1}{DK}}.$$

Combining these results, we have

$$\Delta C_2(t) + \Delta C_3(t) \leq \eta \frac{MD}{K^2(D-K)}\sqrt{\frac{D+1}{K}} \leq \frac{1}{D^2},$$

where the last inequality holds by the condition that $\eta \leq \mathcal{O}(M^{-1}D^{-5/2})$ in Theorem 3.1. Replacing these results, we finally prove that

$$\Delta p(t) \leq \frac{D^2 p(t)\left(1 - Kp(t)\right)}{\sqrt{D}(D^2 - 1)}\left(\Delta C_2(t) + \Delta C_3(t)\right).$$

On the other hand, since $\Delta C_2(t) + \Delta C_3(t)$ is sufficiently small, we can also have

$$\Delta p(t) \geq \frac{1}{K + (D-K)\exp\left(-\frac{C_2(t)+C_3(t)}{\sqrt{D}}\right)\left(1 - \frac{\Delta C_2(t)+\Delta C_3(t)}{2\sqrt{D}}\right)}$$

$$- \frac{1}{K + (D-K)\exp\left(-\frac{C_2(t)+C_3(t)}{\sqrt{D}}\right)}$$

$$\geq \frac{\Delta C_2(t) + \Delta C_3(t)}{2\sqrt{D}} \frac{(D-K)\exp\left(-\frac{C_2(t)+C_3(t)}{\sqrt{D}}\right)}{\left[K + (D-K)\exp\left(-\frac{C_2(t)+C_3(t)}{\sqrt{D}}\right)\right]^2}$$

$$= \frac{p(t)\left(1 - Kp(t)\right)}{2\sqrt{D}}\left(\Delta C_2(t) + \Delta C_3(t)\right).$$

This completes the proof. $\square$

**Lemma D.10.** For $C_1^*(t)$ defined in Lemma D.2, it hols that $C_1^*(t)$ is monotonically increasing w.r.t $t$ when $p(t) \leq \frac{1}{2\sqrt{\pi DK}}$.

*Proof of Lemma D.10.* As Lemma D.8 demonstrates that $p(t)$ is always monotonically increasing. Consequently, it suffices to show that $C_1^*(t)$ is monotonically increasing w.r.t. $p(t)$ when $p(t) \leq \frac{1}{2\sqrt{\pi DK}}$. In the following, we discuss the three scenarios where $\sigma(\cdot)$ is the identity map, ReLU activation function, and Leaky ReLU activation function, respectively. When $\sigma(\cdot)$ is the identity map,

$$C_1^*(t) = \frac{(D-K)p(t)}{DKp(t)^2 - 2Kp(t) + 1} = \frac{D-K}{DKp(t) + \frac{1}{p(t)} - 2K}.$$

It is straightforward that $C_1^*(t)$ is monotonically increasing when $p(t) \leq \frac{1}{\sqrt{DK}}$, as the denominator is decreasing. When $\sigma(\cdot)$ is the ReLU activation function, we have

$$C_1^*(t) = \frac{\pi(D-K)p(t) + 2(D-K)p(t)\arctan\left(\sqrt{K(D-K)}\frac{p(t)}{1-Kp(t)}\right) + 2(D-K)\frac{1-Kp(t)}{\sqrt{K(D-K)}}}{2\pi(DKp(t)^2 - 2Kp(t) + 1)}.$$

By applying basic calculus, we can derive that

$$\frac{\mathrm{d}C_1^*(t)}{\mathrm{d}p(t)} \geq \frac{2\pi(\pi-1)(D-K)\left(DKp(t)^2 - 2Kp(t) + 1\right)}{4\pi^2\left(DKp(t)^2 - 2Kp(t) + 1\right)^2}$$

$$- \frac{2(D-K)\left(\pi p(t) + \frac{1}{\sqrt{K(D-K)}}\right)4\pi K\left(Dp(t) - 1\right)}{4\pi^2\left(DKp(t)^2 - 2Kp(t) + 1\right)^2}$$

$$\geq \frac{3\pi(D-K)\left(1 - 4\pi DKp(t)^2\right)}{4\pi^2\left(DKp(t)^2 - 2Kp(t) + 1\right)^2} + \frac{3\pi(D-K)\left(1 - 2\sqrt{\pi DK}p(t)\right)}{4\pi^2\left(DKp(t)^2 - 2Kp(t) + 1\right)^2},$$

which is positive when $p(t) \leq \frac{1}{2\sqrt{\pi DK}}$. Therefore, we can conclude that when $p(t) \leq \frac{1}{2\sqrt{\pi DK}}$, $C_1^*(t)$ is monotonically increasing w.r.t. $p(t)$. Similarly, when $\sigma(\cdot)$ is the Leaky ReLU activation function, we also have,

$$C_1^*(t) = \frac{(1+\kappa)^2\pi(D-K)p(t) + 2(1-\kappa)^2(D-K)p(t)\arctan\left(\sqrt{K(D-K)}\frac{p(t)}{1-Kp(t)}\right)}{2(1+\kappa^2)\pi(DKp(t)^2 - 2Kp(t) + 1)}$$

$$+ \frac{(1-\kappa)^2(D-K)(1-Kp(t))}{\sqrt{K(D-K)}(1+\kappa^2)\pi(DKp(t)^2 - 2Kp(t) + 1)},$$

and

$$\frac{\mathrm{d}C_1^*(t)}{\mathrm{d}p(t)} \geq \frac{2\pi(\pi-1)(D-K)\left(DKp(t)^2 - 2Kp(t) + 1\right)}{4\pi^2\left(DKp(t)^2 - 2Kp(t) + 1\right)^2}$$

$$= \frac{2(D-K)\left(\pi p(t) + \frac{1}{\sqrt{K(D-K)}}\right)K\left(Dp(t) - 1\right)}{\pi\left(DKp(t)^2 - 2Kp(t) + 1\right)^2}$$

$$\geq \frac{3\pi(D-K)\left(1 - 4\pi DKp(t)^2\right)}{4\pi^2\left(DKp(t)^2 - 2Kp(t) + 1\right)^2} + \frac{3\pi(D-K)\left(1 - 2\sqrt{\pi DK}p(t)\right)}{4\pi^2\left(DKp(t)^2 - 2Kp(t) + 1\right)^2},$$

which proves that $C_1^*(t)$ is monotonically increasing w.r.t. $p(t)$ when $p(t) \leq \frac{1}{2\sqrt{\pi DK}}$. $\square$

**Lemma D.11.** For $C_1^*(t)$ defined in Lemma D.2, it holds that

$$C_1^*(t+1) \geq C_1^*(t) - \frac{3DKp(t)\Delta p(t)}{DKp(t)^2 - 2Kp(t) + 1} C_1^*(t), \tag{D.26}$$

*Proof of Lemma D.11.* We prove (D.26) for $\sigma(\cdot)$ is identity map, ReLU activation function, and Leaky ReLU activation function, respectively. When $\sigma(\cdot)$ is identity map,

$$C_1^*(t+1) = \frac{(D-K)p(t+1)}{DKp(t+1)^2 - 2Kp(t+1) + 1} \geq \frac{(D-K)p(t)}{DKp(t)^2 - 2Kp(t) + 1 + DK\Delta p(t)(2p(t) + \Delta p(t))}$$

$$\geq C_1^*(t) - \frac{DK\Delta p(t)(2p(t) + \Delta p(t))}{DKp(t)^2 - 2Kp(t) + 1} C_1^*(t) \geq C_1^*(t) - \frac{3DKp(t)\Delta p(t)}{DKp(t)^2 - 2Kp(t) + 1} C_1^*(t),$$

where the second inequality holds by Lemma F.9, and the last inequality holds by $\Delta p(t) \leq p(t)$ implied by Lemma D.9. When $\sigma(\cdot)$ is ReLU activation function,

$$C_1^*(t+1) = \frac{\pi(D-K)p(t+1) + 2(D-K)p(t+1)\arctan\left(\sqrt{K(D-K)}\frac{p(t+1)}{1-Kp(t+1)}\right)}{2\pi(DKp(t+1)^2 - 2Kp(t+1) + 1)}$$

$$+ \frac{(D-K)(1-Kp(t+1))}{\sqrt{K(D-K)}\pi(DKp(t+1)^2 - 2Kp(t+1) + 1)}$$

$$\geq \frac{\pi(D-K)p(t) + 2(D-K)p(t)\arctan\left(\sqrt{K(D-K)}\frac{p(t)}{1-Kp(t)}\right) + 2(D-K)\frac{1-Kp(t)}{\sqrt{K(D-K)}}}{2\pi(DKp(t)^2 - 2Kp(t) + 1) + 2\pi DK\Delta p(t)(2p(t) + \Delta p(t))}$$

$$\geq C_1^*(t) - \frac{3DKp(t)\Delta p(t)}{DKp(t)^2 - 2Kp(t) + 1} C_1^*(t),$$

where the first inequality holds as the numerator is a monotonically increasing function w.r.t. $p(t)$. Furthermore, the second inequality holds by Lemma F.9, and $\Delta p(t) \leq p(t)$ implied by Lemma D.9. Similarly, when $\sigma(\cdot)$ is the Leaky ReLU activation function,

$$C_1^*(t+1) = \frac{\frac{(1+\kappa)^2\pi(D-K)}{(1-\kappa)^2}p(t+1) + 2(D-K)p(t+1)\arctan\left(\sqrt{K(D-K)}\frac{p(t+1)}{1-Kp(t+1)}\right) + 2(D-K)\frac{1-Kp(t+1)}{\sqrt{K(D-K)}}}{\frac{2\pi(1+\kappa^2)}{(1-\kappa)^2}(DKp(t+1)^2 - 2Kp(t+1) + 1)}$$

$$\geq \frac{\frac{(1+\kappa)^2\pi(D-K)}{(1-\kappa)^2}p(t) + 2(D-K)p(t)\arctan\left(\sqrt{K(D-K)}\frac{p(t)}{1-Kp(t)}\right) + 2(D-K)\frac{1-Kp(t)}{\sqrt{K(D-K)}}}{\frac{2\pi(1+\kappa^2)}{(1-\kappa)^2}(DKp(t)^2 - 2Kp(t) + 1) + \frac{2\pi(1+\kappa^2)}{(1-\kappa)^2}DK\Delta p(t)(2p(t) + \Delta p(t))}$$

$$\geq C_1^*(t) - \frac{3DKp(t)\Delta p(t)}{DKp(t)^2 - 2Kp(t) + 1} C_1^*(t).$$

This completes the proof □

**Lemma D.12.** For $A(t)$, $B(t)$ defined in Lemma D.5, it holds that

$$\frac{A(t+1)}{A(t+1) + B(t+1)} \frac{1 - Kp(t+1)}{Kp(t+1)(Dp(t+1) - 1)}$$

$$\geq \frac{A(t)}{A(t) + B(t)} \frac{1 - Kp(t)}{Kp(t)(Dp(t) - 1)} - \frac{A(t)}{A(t) + B(t)} \frac{2DKp(t)\Delta p(t)}{K^2 p(t)^2 (Dp(t) - 1)^2}. \tag{D.27}$$

*Proof of Lemma D.12.* Notice that $\frac{B(t)}{A(t)}$ is a non-increasing function w.r.t. $p(t)$. Therefore, we can derive that

$$\frac{A(t+1)}{A(t+1) + B(t+1)} \frac{1 - Kp(t+1)}{Kp(t+1)(Dp(t+1) - 1)}$$

$$= \frac{A(t+1)}{A(t+1) + B(t+1)} \left(\frac{1}{Kp(t+1)(Dp(t+1) - 1)} - \frac{1}{Dp(t+1) - 1}\right)$$

$$\geq \frac{A(t)}{A(t)+B(t)} \left( \frac{1}{Kp(t)\big(Dp(t)-1\big) + \Delta p(t)\big(2DKp(t)+DK\Delta p(t)-K\big)} - \frac{1}{Dp(t)-1} \right)$$

$$\geq \frac{A(t)}{A(t)+B(t)} \frac{1-Kp(t)}{Kp(t)\big(Dp(t)-1\big)} - \frac{A(t)}{A(t)+B(t)} \frac{\Delta p(t)\big(2DKp(t)+DK\Delta p(t)-K\big)}{K^2 p(t)^2 \big(Dp(t)-1\big)^2}$$

$$\geq \frac{A(t)}{A(t)+B(t)} \frac{1-Kp(t)}{Kp(t)\big(Dp(t)-1\big)} - \frac{A(t)}{A(t)+B(t)} \frac{2DKp(t)\Delta p(t)}{K^2 p(t)^2 \big(Dp(t)-1\big)^2},$$

where the last inequality holds $\delta p(t) \leq \frac{1}{D}$ implied by Lemma D.9. This completes the proof. $\square$

Now, we are ready to prove Lemma D.5.

*Proof of Lemma D.5.* As Lemma D.10 guarantees that $C_1^*(t)$ is monotonically increasing when $p(t) \leq \frac{1}{2\sqrt{\pi DK}}$. Consequently, when $p(t) \leq \frac{1}{2\sqrt{\pi DK}}$,

$$C_1^*(t+1) - C_1(t+1) \geq C_1^*(t) - C_1(t+1) = \left( 1 - \frac{\eta D F_3^{(t)}}{Kp(t)C_1^*(t)} \right) \left( C_1^*(t) - C_1(t) \right)$$

$$\geq \left( 1 - \frac{\eta D}{K} \right) \left( C_1^*(t) - C_1(t) \right) \geq \left( 1 - \frac{\eta D}{K} \right)^{t+1} \left( C_1^*(0) - C_1(0) \right) \geq 0.$$

The second inequality holds by $\frac{F_3^{(t)}}{Kp(t)} \leq \frac{1}{K}$, and $C_1^*(t) \geq 1$. The last inequality holds by the assumption of $\eta$ in Theorem 3.1, and $C_1(0) = 0$. In the next, we prove that (D.17) holds when $p(t) \geq \frac{1}{2\sqrt{\pi DK}}$ by induction. We assume (D.17) holds at $t$-th iteration and examine the $t+1$-th iteration. Inspired by the separating strategy in Wang et al. (2024), we consider the following two cases: (i). when $C_1(t) \leq \left( 1 + \frac{2A(t)}{5(A(t)+B(t))} \frac{1-Kp(t)}{Kp(t)(Dp(t)-1)} \right) C_1^*(t)$ and (ii). when $\left( 1 + \frac{2A(t)}{5(A(t)+B(t))} \frac{1-Kp(t)}{Kp(t)(Dp(t)-1)} \right) C_1^*(t) \leq C_1(t) \leq \left( 1 + \frac{4A(t)}{5(A(t)+B(t))} \frac{1-Kp(t)}{Kp(t)(Dp(t)-1)} \right) C_1^*(t)$. For the first case, it suffices to show that

$$\left( 1 + \frac{4A(t+1)}{5\big(A(t+1)+B(t+1)\big)} \frac{1-Kp(t+1)}{Kp(t+1)\big(Dp(t+1)-1\big)} \right) C_1^*(t+1)$$

$$\geq \left( 1 + \frac{2A(t)}{5\big(A(t)+B(t)\big)} \frac{1-Kp(t)}{Kp(t)\big(Dp(t)-1\big)} \right) C_1^*(t). \tag{D.28}$$

This is because if $C_1(t) \leq C_1^*(t)$, we have

$$C_1^*(t) - C_1(t+1) = \left( 1 - \frac{\eta D F_3^{(t)}}{Kp(t)C_1^*(t)} \right) \left( C_1^*(t) - C_1(t) \right) \geq 0,$$

which implies that

$$C_1(t+1) \leq C_1^*(t) \leq \left( 1 + \frac{4A(t+1)}{5\big(A(t+1)+B(t+1)\big)} \frac{1-Kp(t+1)}{Kp(t+1)(Dp(t+1)-1)} \right) C_1^*(t+1).$$

The last inequality is guaranteed by (D.28). On the other hand, if $C_1^*(t) < C_1(t) \leq \left( 1 + \frac{2A(t)}{5(A(t)+B(t))} \frac{1-Kp(t)}{Kp(t)(Dp(t)-1)} \right) C_1^*(t)$, then we can also obtain that

$$C_1(t+1) \leq C_1(t) \leq \left( 1 + \frac{2A(t)}{5\big(A(t)+B(t)\big)} \frac{1-Kp(t)}{Kp(t)\big(Dp(t)-1\big)} \right) C_1^*(t)$$

$$\leq \left( 1 + \frac{4A(t+1)}{5\big(A(t+1)+B(t+1)\big)} \frac{1-Kp(t+1)}{Kp(t+1)(Dp(t+1)-1)} \right) C_1^*(t+1).$$

In the next, we show that (D.28) holds. By applying the lower bounds derived in Lemma D.11 and Lemma D.12, we can derive that

$$\left( 1 + \frac{4A(t+1)}{5\big(A(t+1)+B(t+1)\big)} \frac{1-Kp(t+1)}{Kp(t+1)(Dp(t+1)-1)} \right) C_1^*(t+1)$$

$$\geq \left(1+\frac{2A(t)}{5(A(t)+B(t))}\frac{1-Kp(t)}{Kp(t)\big(Dp(t)-1\big)}\right)C_1^*(t) + \frac{2A(t)}{5(A(t)+B(t))}\frac{1-Kp(t)}{Kp(t)\big(Dp(t)-1\big)}C_1^*(t)$$

$$-\left(1+\frac{4A(t+1)}{5(A(t+1)+B(t+1))}\frac{1-Kp(t+1)}{Kp(t+1)(Dp(t+1)-1)}\right)\frac{3DKp(t)\Delta p(t)}{DKp(t)^2-2Kp(t)+1}C_1^*(t)$$

$$-\frac{A(t)}{A(t)+B(t)}\frac{2DKp(t)\Delta p(t)}{K^2p(t)^2\big(Dp(t)-1\big)^2}C_1^*(t)$$

$$\geq \left(1+\frac{2A(t)}{5(A(t)+B(t))}\frac{1-Kp(t)}{Kp(t)\big(Dp(t)-1\big)}\right)C_1^*(t) + \frac{2\big(1-Kp(t)\big)}{15Kp(t)\big(Dp(t)-1\big)}C_1^*(t)$$

$$-\frac{3(4\pi+1)DKp(t)\Delta p(t)}{Kp(t)\big(Dp(t)-1\big)+1-Kp(t)}C_1^*(t) - \frac{10\pi DKp(t)\Delta p(t)}{Kp(t)\big(Dp(t)-1\big)}C_1^*(t)$$

$$\geq \left(1+\frac{2A(t)}{5(A(t)+B(t))}\frac{1-Kp(t)}{Kp(t)\big(Dp(t)-1\big)}\right)C_1^*(t) + \frac{1-Kp(t)}{Kp(t)\big(Dp(t)-1\big)}\left(\frac{2}{15}-\frac{(22\pi+3)Kp(t)^2}{D}\right)C_1^*(t)$$

$$\geq \left(1+\frac{2A(t)}{5(A(t)+B(t))}\frac{1-Kp(t)}{Kp(t)\big(Dp(t)-1\big)}\right)C_1^*(t),$$

which finishes the proof of (D.28). In the derivation above, the second inequality holds as $\frac{A(t)}{A(t)+B(t)} \geq \frac{1}{3}$ and $\frac{1}{Kp(t)(Dp(t)-1)} \leq 5\pi$ when $p(t) \geq \frac{1}{2\sqrt{\pi DK}}$. The penultimate inequality is derived by Lemma D.9. As we demonstrated previously, (D.28) implies that (D.17) holds at the $t+1$-th iteration for the first case. In the following, we consider the second case, where $\left(1+\frac{2A(t)}{5(A(t)+B(t))}\frac{1-Kp(t)}{Kp(t)(Dp(t)-1)}\right)C_1^*(t) \leq C_1(t) \leq \left(1+\frac{4A(t)}{5(A(t)+B(t))}\frac{1-Kp(t)}{Kp(t)(Dp(t)-1)}\right)C_1^*(t)$. For this case, it suffices to show that

$$\left(1+\frac{4A(t)}{5(A(t)+B(t))}\frac{1-Kp(t)}{Kp(t)\big(Dp(t)-1\big)}\right)C_1^*(t) - \frac{\eta D\big(1-Kp(t)\big)}{10K\big(Dp(t)-1\big)}$$

$$\leq \left(1+\frac{4A(t+1)}{5\big(A(t+1)+B(t+1)\big)}\frac{1-Kp(t+1)}{Kp(t+1)(Dp(t+1)-1)}\right)C_1^*(t+1) \qquad (D.29)$$

This is because

$$C_1(t+1) = C_1(t) + \frac{\eta DF_3^{(t)}}{Kp(t)}\left(1-\frac{C_1(t)}{C_1^*(t)}\right)$$

$$\leq \left(1+\frac{4A(t)}{5(A(t)+B(t))}\frac{1-Kp(t)}{Kp(t)\big(Dp(t)-1\big)}\right)C_1^*(t) - \frac{\eta DF_3^{(t)}}{Kp(t)}\frac{2A(t)}{5(A(t)+B(t))}\frac{1-Kp(t)}{Kp(t)(Dp(t)-1)}$$

$$\leq \left(1+\frac{4A(t)}{5(A(t)+B(t))}\frac{1-Kp(t)}{Kp(t)\big(Dp(t)-1\big)}\right)C_1^*(t) - \frac{\eta D\big(1-Kp(t)\big)}{10K\big(Dp(t)-1\big)}$$

$$\leq \left(1+\frac{4A(t+1)}{5\big(A(t+1)+B(t+1)\big)}\frac{1-Kp(t+1)}{Kp(t+1)(Dp(t+1)-1)}\right)C_1^*(t+1),$$

where the penultimate inequality is derived by $F_3^{(t)} = A(t)+B(t)$ and $A(t) \geq \frac{Kp(t)^2}{4}$, and the last inequality is guaranteed by (D.29). To show (D.29) holds, by applying Lemma D.7, we derive an refined upper bound for $\Delta C_2(t)$ and $\Delta C_3(t)$ as follows:

$$\Delta C_2(t) = \frac{\eta MC_1(t)}{K\sqrt{D}}\left(\frac{F_4^{(t)}}{p(t)}-F_3^{(t)}-KC_1(t)\big(F_{2,1}^{(t)}+p(t)F_1^{(t)}\big)\right)$$

$$\leq \frac{3\eta MC_1(t)}{5K\sqrt{D}}\frac{\big(1-Kp(t)\big)^2}{Kp(t)\big(DKp(t)^2-2Kp(t)+1\big)}A(t) \leq \frac{3(4\pi+1)\eta Mp(t)\big(1-Kp(t)\big)^2}{5K\sqrt{D}\big(DKp(t)^2-2Kp(t)+1\big)}C_1^*(t),$$

and

$$\Delta C_3(t) = \frac{\eta MC_1(t)\big(1-Kp(t)\big)}{\sqrt{D}(D-K)}\left(\frac{F_3^{(t)}}{Kp(t)}-\frac{(D-K)F_5^{(t)}}{Kp(t)\big(1-Kp(t)\big)}-C_1(t)\left(F_1^{(t)}-\frac{(D-K)F_{2,2}^{(t)}}{1-Kp(t)}\right)\right)$$

$$\leq \frac{3\eta M C_1(t)\big(1-Kp(t)\big)}{5\sqrt{D}(D-K)}\frac{1-Kp(t)}{Kp(t)\big(DKp(t)^2-2Kp(t)+1\big)}A(t)$$

$$\leq \frac{3(4\pi+1)\eta M p(t)\big(1-Kp(t)\big)^2}{5(D-K)\sqrt{D}\big(DKp(t)^2-2Kp(t)+1\big)}C_1^*(t).$$

Based on these refined upper bounds for $\Delta C_2(t), \Delta C_3(t)$, and the lower bounds obtained previously, we can derive that

$$\left(1+\frac{4A(t)}{5(A(t)+B(t))}\frac{1-Kp(t)}{Kp(t)\big(Dp(t)-1\big)}\right)C_1^*(t)$$

$$-\left(1+\frac{4A(t+1)}{5\big(A(t+1)+B(t+1)\big)}\frac{1-Kp(t+1)}{Kp(t+1)(Dp(t+1)-1)}\right)C_1^*(t+1)$$

$$\leq\left(1+\frac{4A(t)}{5(A(t)+B(t))}\frac{1-Kp(t)}{Kp(t)\big(Dp(t)-1\big)}\right)C_1^*(t)-\left(1+\frac{4A(t)}{5(A(t)+B(t))}\frac{1-Kp(t)}{Kp(t)\big(Dp(t)-1\big)}\right)C_1^*(t)$$

$$+\frac{(22\pi+3)D\Delta p(t)}{Dp(t)-1}C_1^*(t)$$

$$\leq\frac{(22\pi+3)D}{Dp(t)-1}\frac{D^2p(t)\big(1-Kp(t)\big)}{\sqrt{D}(D^2-1)}\big(\Delta C_2(t)+\Delta C_3(t)\big)C_1^*(t)$$

$$\leq\frac{3(22\pi+3)(4\pi+1)}{5}\frac{\eta D\big(1-Kp(t)\big)}{K\big(Dp(t)-1\big)}\frac{D^3Mp(t)^2\big(1-Kp(t)\big)^2}{(D^2-1)D(D-K)\big(DKp(t)^2-2Kp(t)+1\big)}C_1^*(t)^2$$

$$\leq\frac{\eta D\big(1-Kp(t)\big)}{K\big(Dp(t)-1\big)}\frac{60\pi M}{DK^2}\leq\frac{\eta D\big(1-Kp(t)\big)}{10K\big(Dp(t)-1\big)}.$$

Here, the first inequality is derived by applying the upper bound of $\big(1+\frac{4A(t+1)}{5(A(t+1)+B(t+1))}\frac{1-Kp(t+1)}{Kp(t+1)(Dp(t+1)-1)}\big)C_1^*(t+1)$ obtained previously. The second inequality holds by applying Lemma D.9. The third inequality is derived by replacing the refined upper bound of $\Delta C_2(t)$ and $\Delta C_3(t)$. The penultimate inequality holds as $C_1^*(t)\leq\frac{1}{Kp(t)}$, and the last inequality is guaranteed by $D\geq\Omega(M)$ in the condition of Theorem 3.1. This demonstrates that (D.29) holds in the second case, which completes the proof of (D.17). □

## D.2 THREE PHASES TRAINING

In the previous section, Lemma D.2 accurately characterizes the training dynamics of $\mathbf{W}_V^{(t)}$ and $\mathbf{W}_{KQ}^{(t)}$. Specifically, it demonstrates that $\mathbf{W}_V^{(t)}=C_1(t)\mathbf{V}^*$, where $C_1(t)$ is always upper bounded by $\big(1+\frac{4A(t)}{5(A(t)+B(t))}\frac{1-Kp(t)}{Kp(t)(Dp(t)-1)}\big)C_1^*(t)$. Next, we will show that the update pattern of $C_1(t)$ differs across three distinct phases. In the first phase, $C_1(t)$ monotonically increases, approaching $C_1^*(t)$ while $p(t)$ remains close to $\frac{1}{D}$. In the second phase, $C_1(t)$ remains in a neighborhood of $C_1^*(t)$, while $p(t)$ monotonically increases. This increase exhibits modes characteristic of a tensor power progression, continuing until $p(t)$ reaches $\frac{1}{2K}$. In the third phase, the $p(t)$ will eventually converges to $\frac{1}{K}$, and $C_1^*(t)$ converges to 1, leading the loss also to converge. The formal proof is provided as follows.

**Lemma D.13.** Under the same conditions with Theorem 3.1, there exist $t_1=\Theta(\eta^{-1})$, such that $C_1(t_1)\geq 0.95\cdot C_1^*(t_1)$, and $p(t)\leq\frac{1+D^{-1/4}}{D}$ for all $t\leq t_1$.

*Proof of Lemma D.13.* Notice that when $C_1(t)\leq C_1^*(t)$, $C_1(t)$ is monotonically increasing. Let $t_1$ be the first time such that $C_1(t)\geq 0.95\cdot C_1^*(t)$. For the conclusion regarding $p(t)$ with $t\leq t_1$, we first assume it holds and utilize it to demonstrate other conclusions, and lastly prove it by induction. Since $p(t)$ almost remain unchanged for all $t\leq t_1$, we can obtain that $0.975\cdot C_1^*(t')\geq 0.95\cdot C_1^*(t'')$ for all $t',t''\leq t_1$ (This conclusion is proved in following Lemma D.14). Therefore for all $t<t_1$

$$C_1(t+1)-C_1(t)=\frac{\eta DF_3^{(t)}}{Kp(t)}\left(1-\frac{C_1(t)}{C_1^*(t)}\right)\geq\frac{\eta DF_3^{(t)}}{Kp(t)}\left(1-\frac{C_1(t_1-1)}{\frac{0.95}{0.975}C_1^*(t_1-1)}\right)\geq\frac{\eta DF_3^{(t)}}{40Kp(t)},$$

where the last inequality holds by $\frac{C_1(t_1-1)}{C_1^*(t_1-1)} \le 0.95$. On the other hand, it is straightforward that $C(t+1) - C(t) \le \frac{\eta D F_3^{(t)}}{Kp(t)}$. Additionally, when $\frac{1}{D} \le p(t), p(t_1) \le \frac{1+D^{-1/4}}{D}$, we can obtain that

- If $\sigma(\cdot)$ is the identity map, then

$$\frac{F_3^{(t)}}{Kp(t)} = p(t) = \Theta\left(\frac{1}{D}\right);$$

$$C_1^*(t_1) = \frac{F_3^{(t_1)}}{Kp(t_1)F_1^{(t_1)}} = \frac{\Theta(1)}{Kp(t_1)(Dp(t_1)-1)+1-Kp(t_1)} = \Theta(1).$$

- If $\sigma(\cdot)$ is the ReLU activation function, then

$$\frac{F_3^{(t)}}{Kp(t)} = \frac{p(t)}{4} + \frac{p(t)}{2\pi}\arctan\left(\frac{\sqrt{K(D-K)}p(t)}{1-Kp(t)}\right) + \frac{1}{2\pi\sqrt{K(D-K)}}\left(1-Kp(t)\right) = \Theta\left(\frac{1}{\sqrt{DK}}\right);$$

$$C_1^*(t_1) = \frac{F_3^{(t_1)}}{Kp(t_1)F_1^{(t_1)}} = \frac{\Theta\left(\sqrt{\frac{D}{K}}\right)}{Kp(t_1)(Dp(t_1)-1)+1-Kp(t_1)} = \Theta\left(\sqrt{\frac{D}{K}}\right).$$

- If $\sigma(\cdot)$ is the Leaky ReLU activation function, then

$$\frac{F_3^{(t)}}{Kp(t)} = \frac{(1+\kappa)^2 p(t)}{4} + \frac{(1-\kappa)^2 p(t)}{2\pi}\arctan\left(\frac{\sqrt{K(D-K)}p(t)}{1-Kp(t)}\right) + \frac{(1-\kappa)^2}{2\pi\sqrt{K(D-K)}}\left(1-Kp(t)\right)$$

$$= \Theta\left(\frac{1}{\sqrt{DK}}\right);$$

$$C_1^*(t_1) = \frac{F_3^{(t_1)}}{Kp(t_1)F_1^{(t_1)}} = \frac{\Theta\left(\sqrt{\frac{D}{K}}\right)}{Kp(t_1)(Dp(t_1)-1)+1-Kp(t_1)} = \Theta\left(\sqrt{\frac{D}{K}}\right).$$

Therefore, we conclude that

$$t_1 = \frac{0.95 \cdot C_1^*(t_1)}{\frac{1}{t_1}\sum_{t=0}^{t_1-1}\Delta C_1(t)} = \begin{cases} \frac{\Theta(1)}{\Theta(\eta)} = \Theta(\eta^{-1}), & \text{if } \sigma(\cdot)\text{is identity map;} \\[2mm] \frac{\Theta\left(\sqrt{\frac{D}{K}}\right)}{\Theta\left(\eta\sqrt{\frac{D}{K}}\right)} = \Theta(\eta^{-1}), & \text{if } \sigma(\cdot)\text{is ReLU activation function;} \\[2mm] \frac{\Theta\left(\sqrt{\frac{D}{K}}\right)}{\Theta\left(\eta\sqrt{\frac{D}{K}}\right)} = \Theta(\eta^{-1}), & \text{if } \sigma(\cdot)\text{is Leaky ReLU activation function.} \end{cases}$$

Next we prove that $p(t_1) \le \frac{1+D^{-1/4}}{D}$ by induction. Assume it holds at $t$-th iteration, then by Lemma D.9 we can derive that

$$\Delta C_2(t) + \Delta C_3(t) \le \eta\frac{MD}{K^2(D-K)}\sqrt{\frac{D+1}{DK}},$$

and consequently

$$\Delta p(t) \le \frac{D^2 p(t)\left(1-Kp(t)\right)}{\sqrt{D}(D^2-1)}\left(\Delta C_2(t)+\Delta C_3(t)\right) \le \frac{3M\eta}{\sqrt{K^5 D^3}}.$$

Therefore, we can eventually conclude that

$$p(t_1) \le p(0) + \sum_{t=0}^{t_1-1}\Delta p(t) \le \frac{1}{D} + \Theta\left(\frac{M}{\sqrt{K^5 D^3}}\right) \le \frac{1+D^{-1/4}}{D},$$

where the last inequality is derived by our condition that $D = \Omega\left(\text{poly}(M)\right)$ in Theorem 3.1. This completes the proof. $\qquad\square$

**Lemma D.14.** For all $t', t'' \leq t_1$, where $t_1$ is defined in Lemma D.13, it holds that $0.975 \cdot C_1^*(t') \geq 0.95 \cdot C_1^*(t'')$.

*Proof of Lemma D.14.* Notice that by the definition of $C_1^*(t)$, it is entirely determined by $p(t)$. And for all $t', t'' \leq t_1$, we all have $\frac{1}{D} \leq p(t'), p(t'') \leq \frac{1+D^{-1/4}}{D}$. With Lemma F.1 and Lemma F.5, we can further derive that

- If $\sigma(\cdot)$ is the identity map, then

$$C_1^*(t') = \frac{(D-K)p(t')}{DKp(t')^2 - 2Kp(t') + 1} \leq \frac{D-K}{\frac{D}{1+D^{-1/4}} - K(1 - D^{-1/4})} \leq 1 + D^{-\frac{1}{4}}$$

$$C_1^*(t') = \frac{(D-K)p(t')}{DKp(t')^2 - 2Kp(t') + 1} \geq \frac{D-K}{D-K} = 1.$$

  It immediately concludes that

$$0.975 \cdot C_1^*(t'') - 0.95 \cdot C_1^*(t') \geq \frac{1}{40} - D^{-\frac{1}{4}} \geq 0,$$

  as $D \geq \Omega(1)$.

- If $\sigma(\cdot)$ is the ReLU activation function, then

$$C_1^*(t') = \frac{2(D-K)F_3^{(t')}}{Kp(t')\big(DKp(t')^2 - 2Kp(t') + 1\big)} \leq \frac{\frac{1}{\pi}\sqrt{\frac{D-K}{K}} + \frac{(D-K)p(t')}{2}}{1 - Kp(t')}$$

$$\leq \frac{1}{\pi}\sqrt{\frac{D-K}{K}} + \frac{(D-K)p(t')}{2} + \frac{Kp(t')\big(\frac{1}{\pi}\sqrt{\frac{D-K}{K}} + \frac{(D-K)p(t')}{2}\big)}{\big(1 - Kp(t')\big)^2}$$

$$\leq \frac{1}{\pi}\sqrt{\frac{D-K}{K}} + 1 + \frac{K}{D} + \frac{2}{\pi}\sqrt{\frac{K}{D}} \leq \frac{1}{\pi}\sqrt{\frac{D-K}{K}} + 2,$$

  where the penultimate and last inequalities hold by utilizing Lemma F.10, and $\frac{1}{D} \leq p(t') \leq \frac{1+D^{-1/4}}{D}$, $D \geq \Omega\big(\mathrm{poly}(K)\big)$ in the conditions of Theorem 3.1. Similarly, we can also obtain that

$$C_1^*(t'') = \frac{2(D-K)F_3^{(t'')}}{Kp(t'')\big(DKp(t'')^2 - 2Kp(t'') + 1\big)} \geq \frac{\frac{1}{\pi}\sqrt{\frac{D-K}{K}}}{1 + DKp(t'')^2}$$

$$\geq \frac{1}{\pi}\sqrt{\frac{D-K}{K}} - \frac{1}{\pi}\sqrt{\frac{D-K}{K}}DKp(t'')^2 \geq \frac{1}{\pi}\sqrt{\frac{D-K}{K}} - 1,$$

  where the penultimate and last inequalities hold by utilizing Lemma F.9, and $\frac{1}{D} \leq p(t') \leq \frac{1+D^{-1/4}}{D}$, $D \geq \Omega\big(\mathrm{poly}(K)\big)$ in the conditions of Theorem 3.1. Based on these two results, it is straightforward that

$$0.975 \cdot C_1^*(t'') - 0.95 \cdot C_1^*(t') \geq \frac{1}{40\pi}\sqrt{\frac{D-K}{K}} - 3 \geq 0,$$

  as $D \geq \Omega\big(\mathrm{poly}(K)\big)$.

- If $\sigma(\cdot)$ is the ReLU activation function, then

$$C_1^*(t') = \frac{2(D-K)F_3^{(t')}}{(1+\kappa^2)Kp(t')\big(DKp(t')^2 - 2Kp(t') + 1\big)} \leq \frac{\frac{(1-\kappa)^2}{\pi}\sqrt{\frac{D-K}{K}} + \frac{(1+\kappa^2)(D-K)p(t')}{2}}{(1+\kappa^2)\big(1 - Kp(t')\big)}$$

$$\leq \frac{(1-\kappa)^2}{(1+\kappa^2)\pi}\sqrt{\frac{D-K}{K}} + 2;$$

$$C_1^*(t'') = \frac{2(D-K)F_3^{(t')}}{(1+\kappa^2)Kp(t')\big(DKp(t')^2 - 2Kp(t') + 1\big)} \geq \frac{\frac{(1-\kappa)^2}{\pi}\sqrt{\frac{D-K}{K}}}{(1+\kappa)^2\big(1 + DKp(t'')^2\big)}$$

$$\geq \frac{(1-\kappa)^2}{(1+\kappa^2)\pi}\sqrt{\frac{D-K}{K}}-1.$$

Combining these results directly leads to

$$0.975 \cdot C_1^*(t'') - 0.95 \cdot C_1^*(t') \geq \frac{(1-\kappa)^2}{40(1+\kappa^2)\pi}\sqrt{\frac{D-K}{K}}-3 \geq 0,$$

as $D \geq \Omega\big(\mathrm{poly}(K)\big)$.

This completes the proof. $\qquad\square$

Lemma D.13 successfully demonstrate that at the initial phase of training, $C_1(t)$ will monotonically increases until $0.95 \cdot C_1^*(t)$, while $p(t)$ remains smaller that $\frac{1+D^{-1/4}}{D}$. Furthermore, once $C_1(t)$ reaches $0.95 \cdot C_1^*(t)$, it never falls below this threshold again. Combined with the conclusion demonstrated in Lemma D.5 that $C_1(t)$ is always upper bounded by $\big(1 + \frac{4A(t)}{5(A(t)+B(t))}\frac{1-Kp(t)}{Kp(t)(Dp(t)-1)}\big)C_1^*(t)$, we can claim that $C_1(t)$ will always remain inner a neighborhood around $C_1^*(t)$. The following lemma provides a formal illustration.

**Lemma D.15.** Under the same conditions as Theorem 3.1 and with $t_1$ as defined in Lemma D.13, for all $t \geq t_1$, the following holds:

$$C_1(t) \geq \left[0.95 \vee \left(1 - \frac{4A(t)}{5(A(t)+B(t))}\frac{1-Kp(t)}{Kp(t)\big(Dp(t)-1\big)}\right)\right]C_1^*(t), \qquad (\text{D.30})$$

where $A(t)$ and $B(t)$ are defined same as in Lemma D.5.

Before we prove Lemma D.15, we first introduce the following lemma, which will be utilized in the proof of Lemma D.15.

**Lemma D.16.** For $C_1^*(t)$ defined in Lemma D.2, it always holds that

$$C_1^*(t+1) \leq C_1^*(t) + \frac{3\big(D-K+KC_1^*(t)\big)}{2\big(DKp(t)^2-2Kp(t)+1-K\Delta p(t)\big)}\Delta p(t). \qquad (\text{D.31})$$

In addition, $C_1^*(t)$ is monotonically decreasing when $p(t) \geq \frac{2}{\sqrt{DK}}$.

*Proof of Lemma D.16.* We prove this lemma by considering $\sigma(\cdot)$ as the identity map, ReLU activation function, and Leaky ReLU activation function, respectively.

- If $\sigma(\cdot)$ is the identity map, then

$$C_1^*(t+1) = \frac{(D-K)p(t+1)}{DKp(t+1)^2-2Kp(t+1)+1} \leq \frac{(D-K)p(t)+(D-K)\Delta p(t)}{DKp(t)^2-2Kp(t)+1-2K\Delta p(t)}$$

$$\leq C_1^*(t) + \frac{D-K+KC_1^*(t)}{DKp(t)^2-2Kp(t)+1-K\Delta p(t)}\Delta p(t).$$

In addition,

$$C_1^*(t) = \frac{D-K}{DKp(t)+\frac{1}{p(t)}-2K},$$

which is obviously decreasing when $p(t) \geq \frac{2}{\sqrt{DK}}$.

- If $\sigma(\cdot)$ is the ReLU activation function, then

$$C_1^*(t+1) = \frac{\pi(D-K)p(t+1)+2(D-K)p(t+1)\arctan\left(\frac{\sqrt{K(D-K)}p(t+1)}{1-Kp(t+1)}\right)+2(D-K)\frac{1-Kp(t+1)}{\sqrt{K(D-K)}}}{2\pi(DKp(t+1)^2-2Kp(t+1)+1)}$$

$$\leq \frac{\pi(D-K)p(t)+2(D-K)p(t)\arctan\left(\frac{\sqrt{K(D-K)}p(t)}{1-Kp(t)}\right)}{2\pi(DKp(t)^2-2Kp(t)+1)-2\pi K\Delta p(t)}$$

$$+\frac{2(D-K)\frac{1-Kp(t)}{\sqrt{K(D-K)}}+3\pi(D-K)\Delta p(t)}{2\pi(DKp(t)^2-2Kp(t)+1)-2\pi K\Delta p(t)}$$

$$\leq C_1^*(t)+\frac{3\big(D-K+KC_1^*(t)\big)}{2\big(DKp(t)^2-2Kp(t)+1-K\Delta p(t)\big)}\Delta p(t).$$

In addition,

$$C_1^*(t)=\frac{D-K}{2\big(DKp(t)+\frac{1}{p(t)}-2K\big)}+\frac{(D-K)\arctan\left(\frac{\sqrt{K(D-K)}p(t)}{1-Kp(t)}\right)}{\pi\big(DKp(t)+\frac{1}{p(t)}-2K\big)}+\frac{\sqrt{\frac{D-K}{K}}\big(1-Kp(t)\big)}{\pi\big(DKp(t)+\frac{1}{p(t)}-2K\big)},$$

where all these three terms are monotonically decreasing w.r.t. $p(t)$, when $p(t)\geq\frac{2}{\sqrt{DK}}$. This demonstrates that $C_1^*(t)$ is monotonically decreasing when $p(t)\geq\frac{2}{\sqrt{DK}}$.

- If $\sigma(\cdot)$ is the Leaky ReLU activation function, then by a similar calculation process,

$$C_1^*(t+1)$$

$$=\frac{\pi(1+\kappa)^2(D-K)p(t+1)+2(1-\kappa)^2(D-K)p(t+1)\arctan\left(\frac{\sqrt{K(D-K)}p(t+1)}{1-Kp(t+1)}\right)}{2(1+\kappa^2)\pi(DKp(t+1)^2-2Kp(t+1)+1)}$$

$$+\frac{(1-\kappa)^2(D-K)(1-Kp(t+1))}{(1+\kappa^2)\pi(DKp(t+1)^2-2Kp(t+1)+1)\sqrt{K(D-K)}}$$

$$\leq\frac{\pi(1+\kappa)^2(D-K)p(t)+2(1-\kappa)^2(D-K)p(t)\arctan\left(\frac{\sqrt{K(D-K)}p(t)}{1-Kp(t)}\right)}{2(1+\kappa^2)\pi(DKp(t)^2-2Kp(t)+1)-2\pi(1+\kappa^2)K\Delta p(t)}$$

$$+\frac{2(1-\kappa)^2(D-K)\frac{1-Kp(t)}{\sqrt{K(D-K)}}+3\pi(1+\kappa)^2(D-K)\Delta p(t)}{2(1+\kappa^2)\pi(DKp(t)^2-2Kp(t)+1)-2\pi(1+\kappa^2)K\Delta p(t)}$$

$$\leq C_1^*(t)+\frac{3\big(D-K+KC_1^*(t)\big)}{2\big(DKp(t)^2-2Kp(t)+1-K\Delta p(t)\big)}\Delta p(t).$$

In addition,

$$C_1^*(t)=\frac{(1+\kappa)^2(D-K)}{2(1+\kappa^2)\big(DKp(t)+\frac{1}{p(t)}-2K\big)}+\frac{(1-\kappa)^2(D-K)\arctan\left(\frac{\sqrt{K(D-K)}p(t)}{1-Kp(t)}\right)}{\pi(1+\kappa^2)\big(DKp(t)+\frac{1}{p(t)}-2K\big)}$$

$$+\frac{(1-\kappa)^2\sqrt{\frac{D-K}{K}}\big(1-Kp(t)\big)}{\pi(1+\kappa^2)\big(DKp(t)+\frac{1}{p(t)}-2K\big)},$$

where all these three terms are monotonically decreasing w.r.t. $p(t)$, when $p(t)\geq\frac{2}{\sqrt{DK}}$. This demonstrates that $C_1^*(t)$ is monotonically decreasing when $p(t)\geq\frac{2}{\sqrt{DK}}$.

This completes the proof. □

Now, we are ready to prove Lemma D.15

*Proof of Lemma D.15.* We first prove the first part of (D.30), i.e. $C_1(t)\geq 0.95\cdot C_1^*(t)$ for all $t>t_1$, by induction. To establish the conclusion, we consider two cases at the $t$-th iteration: (i). when $C_1(t)\geq 0.975\cdot C_1^*(t)$. (ii). when $0.95\cdot C_1^*(t)\leq C_1(t)<0.975\cdot C_1^*(t)$. For the first case,

when $p(t) \leq \frac{1}{2\sqrt{\pi D K}}$, Lemma D.5 shows that $C_1(t) \leq C_1^*(t)$, implying that $C_1(t+1) \geq C_1(t)$. Then we can derive that

$$C_1(t+1) \geq C_1(t) \geq 0.975 \cdot C_1^*(t) \geq 0.975 \cdot C_1^*(t+1) - \frac{3\big(D - K + K C_1^*(t)\big)}{2\big(DKp(t)^2 - 2Kp(t) + 1 - K\Delta p(t)\big)}\Delta p(t)$$

$$\geq 0.975 \cdot C_1^*(t+1) - \frac{3(D-K)}{D^2} \geq 0.975 \cdot C_1^*(t+1) - 0.025 \geq 0.95 \cdot C_1^*(t+1),$$

where the third inequality holds by applying the lower bound of $C_1^*(t)$ demonstrated in Lemma D.16. The forth inequality holds as $C_1^*(t) \leq \sqrt{\frac{D}{K}}$, and $\Delta p(t) \leq \frac{1}{D^2}$ guaranteed by Lemma D.9. The penultimate inequality holds as $D \geq \Omega\big(\text{poly}(K)\big)$, and the last inequality holds as $C_1^*(t) \geq 1$. When $p(t) \geq \frac{1}{2\sqrt{\pi D K}}$, the upper bound of $C_1(t)$ established in Lemma D.5 can help to derive that

$$C_1(t+1) \geq C_1(t) - \frac{4\eta D A(t)(1 - Kp(t))}{5K^2 p(t)^2 \big(Dp(t) - 1\big)}$$

$$\geq 0.975 \cdot C_1^*(t+1) - \frac{3(D-K)}{D^2} - \frac{4\eta D A(t)}{5K^2 p(t)^2 \big(Dp(t) - 1\big)}$$

$$\geq 0.975 \cdot C_1^*(t+1) - \frac{3(D-K)}{D^2} - \eta\sqrt{\frac{D}{K^3}} \geq 0.975 \cdot C_1^*(t+1) - 0.025 \geq 0.95 \cdot C_1^*(t+1).$$

Here, the second inequality applies the previously obtained lower bound for $C_1(t)$. The third inequality holds as $A(t) \leq Kp(t)^2$, and $p(t) \geq \frac{1}{2\sqrt{\pi D K}}$. The penultimate inequality is derived by $D \geq \Omega\big(\text{poly}(K)\big)$ and $\eta \leq \mathcal{O}(MD^{-5/2})$ in the condition of Theorem 3.1. These results demonstrate that under the first case, $C_1(t+1) \geq 0.95 \cdot C_1^*(t+1)$. Let's consider the second case, where $0.95 \cdot C_1^*(t) \leq C_1(t) < 0.975 \cdot C_1^*(t)$. Under this case, it is obvious that $C_1(t+1)$ would be larger than $C_1(t)$, and by the updating rule, we have

$$C_1(t+1) \geq C_1(t) + \frac{\eta D F_3^{(t)}}{40Kp(t)} \geq 0.95 \cdot C_1^*(t) + \frac{\eta D F_3^{(t)}}{40Kp(t)} - \frac{3\big(D - K + K C_1^*(t)\big)}{2\big(DKp(t)^2 - 2Kp(t) + 1 - K\Delta p(t)\big)}\Delta p(t)$$

$$\geq 0.95 \cdot C_1^*(t) + \eta p(t)\left(\frac{D}{80} - \frac{3MD^3(1 - Kp(t))}{\sqrt{D}(D^2 - 1)(D - K)K^2}\sqrt{\frac{D+1}{K}}\right)$$

$$\geq 0.95 \cdot C_1^*(t) + \eta p(t)\left(\frac{D}{80} - \frac{4M}{K^{\frac{5}{2}}}\right) \geq 0.95 \cdot C_1^*(t).$$

Here, the second inequality holds by (D.31), the third inequality holds since $F_3^{(t)} \geq \frac{Kp(t)^2}{2}$ by Lemma F.5, $\frac{D - K + K C_1^*(t)}{DKp(t)^2 - 2Kp(t) + 1 - K\Delta p(t)} \leq 2D$, and applying the conclusion of upper bound of $\Delta p(t)$ demonstrated in Lemma D.9. Besides, the last two inequalities is guaranteed by $D \geq \Omega\big(\text{poly}(M, K)\big)$. This finishes the proof of $C_1(t) \geq 0.95 \cdot C_1^*(t)$ for all $t \geq t_1$. In the next, we prove the second part of (D.30), i.e. $C_1(t) \geq \big(1 - \frac{4A(t)}{5(A(t)+B(t))}\frac{1 - Kp(t)}{Kp(t)(Dp(t)-1)}\big)C_1^*(t)$. In fact, we only need to consider the scenario where $p(t) \geq \frac{2}{\sqrt{DK}}$. This is because when $p(t) \leq \frac{2}{\sqrt{DK}}$,

$$\frac{4A(t)}{5(A(t)+B(t))}\frac{1 - Kp(t)}{Kp(t)(Dp(t) - 1)} \geq \frac{4(1 - Kp(t))}{5\big(Kp(t) + \frac{2}{\pi}\sqrt{\frac{K}{D-K}}(1 - Kp(t))\big)(Dp(t) - 1)}$$

$$\geq \frac{1}{10\sqrt{DK}p(t)} \geq 0.05.$$

Therefore, $C_1(t) \geq 0.95 \cdot C_1^*(t)$ guarantee that $C_1(t) \geq \big(1 - \frac{4A(t)}{5(A(t)+B(t))}\frac{1 - Kp(t)}{Kp(t)(Dp(t)-1)}\big)C_1^*(t)$ holds when $p(t) \leq \frac{2}{\sqrt{DK}}$. When $p(t) \geq \frac{2}{\sqrt{DK}}$, we also consider two cases: (i). when $C_1(t) > \big(1 - \frac{2A(t)}{5(A(t)+B(t))}\frac{1 - Kp(t)}{Kp(t)(Dp(t)-1)}\big)C_1^*(t)$. (ii). when $\big(1 - \frac{4A(t)}{5(A(t)+B(t))}\frac{1 - Kp(t)}{Kp(t)(Dp(t)-1)}\big)C_1^*(t) \leq C_1(t) \leq \big(1 - \frac{2A(t)}{5(A(t)+B(t))}\frac{1 - Kp(t)}{Kp(t)(Dp(t)-1)}\big)C_1^*(t)$. Then, for the first case, at the $t+1$-th iteration, we have

$$C_1(t+1) \geq C_1(t) - \frac{4\eta D A(t)(1 - Kp(t))}{5K^2 p(t)^2 \big(Dp(t) - 1\big)}$$

$$\geq \left(1 - \frac{2A(t)}{5(A(t) + B(t))} \frac{1 - Kp(t)}{Kp(t)(Dp(t) - 1)}\right) C_1^*(t) - \frac{4\eta DA(t)(1 - Kp(t))}{5K^2 p(t)^2 (Dp(t) - 1)}$$

$$\geq \left(1 - \frac{4A(t+1)}{5(A(t+1) + B(t+1))} \frac{1 - Kp(t+1)}{Kp(t+1)(Dp(t+1) - 1)}\right) C_1^*(t+1)$$

$$+ \frac{2A(t)}{5(A(t) + B(t))} \frac{1 - Kp(t)}{Kp(t)(Dp(t) - 1)} C_1^*(t) - \frac{4\eta DA(t)(1 - Kp(t))}{5K^2 p(t)^2 (Dp(t) - 1)}$$

$$- \frac{A(t)}{A(t) + B(t)} \frac{\Delta p(t) (2DKp(t) + DK\Delta p(t) - K)}{K^2 p(t)^2 (Dp(t) - 1)^2} C_1^*(t)$$

$$\geq \left(1 - \frac{4A(t+1)}{5(A(t+1) + B(t+1))} \frac{1 - Kp(t+1)}{Kp(t+1)(Dp(t+1) - 1)}\right) C_1^*(t+1)$$

$$+ \frac{1 - Kp(t)}{Kp(t)(Dp(t) - 1)} \left(\frac{C_1^*(t)}{5} - \eta \frac{4D}{5} - \frac{DC_1^*(t)\Delta p(t)}{Dp(t) - 1}\right)$$

$$\geq \left(1 - \frac{4A(t+1)}{5(A(t+1) + B(t+1))} \frac{1 - Kp(t+1)}{Kp(t+1)(Dp(t+1) - 1)}\right) C_1^*(t+1)$$

$$+ \frac{1 - Kp(t)}{Kp(t)(Dp(t) - 1)} \left(\frac{1}{5} - \frac{4}{5\sqrt{D^3}} - \frac{2}{\sqrt{D^3}}\right)$$

$$\geq \left(1 - \frac{4A(t+1)}{5(A(t+1) + B(t+1))} \frac{1 - Kp(t+1)}{Kp(t+1)(Dp(t+1) - 1)}\right) C_1^*(t+1).$$

In particular, the third inequality is obtained by replacing the the lower bound of $\frac{A(t)}{A(t)+B(t)} \frac{1-Kp(t)}{Kp(t)(Dp(t)-1)}$ in Lemma D.12, and utilizing $C_1^*(t) \geq C_1^*(t+1)$ when $p(t) \geq \frac{2}{\sqrt{DK}}$, which is demonstrated in Lemma D.16. The forth inequality is derived by the facts $\frac{A(t)}{A(t)+B(t)} \geq \frac{1}{2}$ when $p(t) \geq \frac{2}{\sqrt{DK}}$, $A(t) \leq Kp(t)^2$, and utilizing the upper bound of $\Delta p(t)$ in Lemma D.9. Lastly, the penultimate inequality is derived as $1 \leq C_1^*(t) \leq \sqrt{\frac{D}{K}}$, $\Delta p(t) \leq \frac{1}{D^{5/2}}$, and $\eta \leq \mathcal{O}(D^{-5/2})$. This demonstrates that the second part of (D.30) holds at $t + 1$-th iteration for the first case. On the other hand, for the second case, $C_1(t + 1)$ would be strictly larger than $C_1(t)$, and it can be demonstrated that

$$C_1(t+1) \geq C_1(t) + \frac{2\eta DA(t)(1 - Kp(t))}{5K^2 p(t)^2 (Dp(t) - 1)}$$

$$\geq \left(1 - \frac{4A(t)}{5(A(t) + B(t))} \frac{1 - Kp(t)}{Kp(t)(Dp(t) - 1)}\right) C_1^*(t) + \frac{2\eta DA(t)(1 - Kp(t))}{5K^2 p(t)^2 (Dp(t) - 1)}$$

$$\geq \left(1 - \frac{4A(t+1)}{5(A(t+1) + B(t+1))} \frac{1 - Kp(t+1)}{Kp(t+1)(Dp(t+1) - 1)}\right) C_1^*(t+1)$$

$$+ \frac{2\eta DA(t)(1 - Kp(t))}{5K^2 p(t)^2 (Dp(t) - 1)} - \frac{A(t)}{A(t) + B(t)} \frac{\Delta p(t) (2DKp(t) + DK\Delta p(t) - K)}{K^2 p(t)^2 (Dp(t) - 1)^2} C_1^*(t)$$

$$\geq \left(1 - \frac{4A(t+1)}{5(A(t+1) + B(t+1))} \frac{1 - Kp(t+1)}{Kp(t+1)(Dp(t+1) - 1)}\right) C_1^*(t+1)$$

$$+ \frac{\eta(1 - Kp(t))}{Kp(t)(Dp(t) - 1)} \left(\frac{D}{5} - \frac{2DM(1 - Kp(t))}{\sqrt{K^7}(Dp(t) - 1)}\right)$$

$$\geq \left(1 - \frac{4A(t+1)}{5(A(t+1) + B(t+1))} \frac{1 - Kp(t+1)}{Kp(t+1)(Dp(t+1) - 1)}\right) C_1^*(t+1),$$

where the last inequality holds as $\frac{2DM(1 - Kp(t))}{\sqrt{K^7}(Dp(t)-1)} \leq \mathcal{O}\left(\frac{\sqrt{D}M}{K^3}\right) \leq \mathcal{O}(D)$. This demonstrates that under the second case, we still have

$$C_1(t+1) \geq \left(1 - \frac{4A(t+1)}{5(A(t+1) + B(t+1))} \frac{1 - Kp(t+1)}{Kp(t+1)(Dp(t+1) - 1)}\right) C_1^*(t+1),$$

which finishes the proof of (D.30). □

Lemmas D.15 and D.5 together establish matching lower and upper bounds for $C_1(t)$ after $t_1$. Based on these bounds, we can derive a precise training time at which $p(t)$ achieves $\frac{1}{2K}$. This result is formally presented in the following lemma.

**Lemma D.17.** Under the same conditions as Theorem 3.1, there exists $T^* = \Theta\big(\frac{KD^2}{\eta \sum_{m=1}^{M} \|\mathbf{v}_m^*\|_2^2}\big)$, such that $p(T^*) \geq \frac{1}{2}$.

*Proof of Lemma D.17.* Notice that Lemma D.15 and Lemma D.5 guarantee that

$$0.95 \cdot C_1^*(t) \leq C_1(t) \leq (4\pi + 1) \cdot C_1^*(t)$$

for all $t \geq t_1$. The left hand side inequality is straightforward, and the right hand side holds because: when $p(t) \leq \frac{1}{2\sqrt{\pi DK}}$, $C_1(t) \leq C_1^*(t) < (4\pi + 1) \cdot C_1^*(t)$; when $p(t) \geq \frac{1}{2\sqrt{\pi DK}}$,

$$C_1(t) \leq \left(1 + \frac{4A(t)}{5\big(A(t) + B(t)\big)} \frac{1 - Kp(t)}{Kp(t)\big(Dp(t) - 1\big)}\right) C_1^*(t) \leq (4\pi + 1) \cdot C_1^*(t).$$

On the other hand, Lemma D.15 and Lemma D.5 also guarantee that

$$C_1(t) \leq \left(1 + \frac{4A(t)}{5\big(A(t) + B(t)\big)} \frac{1 - Kp(t)}{Kp(t)\big(Dp(t) - 1\big)}\right) C_1^*(t);$$

$$C_1(t) \geq \left(1 - \frac{4A(t)}{5\big(A(t) + B(t)\big)} \frac{1 - Kp(t)}{Kp(t)\big(Dp(t) - 1\big)}\right) C_1^*(t) \tag{D.32}$$

These two lower and upper bounds of $C_1(t)$ allow us to apply Lemma D.7 to derive lower and upper bounds for $\Delta C_2(t) + \Delta C_3(t)$ as

$$\Delta C_2(t) + \Delta C_3(t) \leq \eta \frac{9(4\pi + 1)Dp(t)\big(1 - Kp(t)\big)^2 \sum_{m=1}^{M} \|\mathbf{v}_m^*\|_2^2}{10K(D - K)\sqrt{D}\big(DKp(t)^2 - 2Kp(t) + 1\big)} C_1^*(t);$$

$$\Delta C_2(t) + \Delta C_3(t) \geq \eta \frac{19Dp(t)\big(1 - Kp(t)\big)^2 \sum_{m=1}^{M} \|\mathbf{v}_m^*\|_2^2}{200K(D - K)\sqrt{D}\big(DKp(t)^2 - 2Kp(t) + 1\big)} C_1^*(t), \tag{D.33}$$

where we replacing $M$ with $\sum_{m=1}^{M} \|\mathbf{v}_m^*\|_2^2$ to match the presentation in our Theorem 3.1. With these bounds in hand, we denote $T^*$ as the first time such that $p(t) \geq \frac{1}{2K}$. Then for all $t_1 \leq t \leq T^*$, by applying Lemma D.9 and the upper and lower bounds of $\Delta C_2(t) + \Delta C_3(t)$ obtained in (D.33), it can be derived that

$$\Delta p(t) \leq \frac{D^2 p(t)\big(1 - Kp(t)\big)}{\sqrt{D}(D^2 - 1)}\big(\Delta C_2(t) + \Delta C_3(t)\big) \leq \eta \frac{(8\pi + 2) \sum_{m=1}^{M} \|\mathbf{v}_m^*\|_2^2}{K\sqrt{DK}} p(t)^2;$$

$$\Delta p(t) \geq \frac{p(t)\big(1 - Kp(t)\big)}{2\sqrt{D}}\big(\Delta C_2(t) + \Delta C_3(t)\big) \geq \eta \frac{\sum_{m=1}^{M} \|\mathbf{v}_m^*\|_2^2}{50K\sqrt{DK}} p(t)^2.$$

Notice that the iterative rules for $p(t)$ satisfying the assumptions in Lemma F.11. By applying Lemma F.11 with the initialization that $\frac{1}{D} \leq p(t_1) \leq \frac{2}{D}$, we can obtained that

$$T^* - t_1 \leq \frac{50D^2 K}{\eta \sum_{m=1}^{M} \|\mathbf{v}_m^*\|_2^2} + 100(8\pi + 2)\big(\log D - \log K\big) \leq \Theta\left(\frac{D^2 K}{\eta \sum_{m=1}^{M} \|\mathbf{v}_m^*\|_2^2}\right)$$

$$T^* - t_1 \geq \frac{D^2 K}{35\pi\eta \sum_{m=1}^{M} \|\mathbf{v}_m^*\|_2^2} - \big(\log D - \log K\big) \geq \Theta\left(\frac{D^2 K}{\eta \sum_{m=1}^{M} \|\mathbf{v}_m^*\|_2^2}\right).$$

This results demonstrates that $T^* = t_1 + \Theta\big(\frac{D^2 K}{\eta \sum_{m=1}^{M} \|\mathbf{v}_m^*\|_2^2}\big) = \Theta\big(\frac{D^2 K}{\eta \sum_{m=1}^{M} \|\mathbf{v}_m^*\|_2^2}\big)$. This finishes the proof. □

In the next, we provide the analysis for the last stage that $p(t)$ eventually converges to $\frac{1}{K}$. This result is formally presented in the following lemma.

**Lemma D.18.** Under the same conditions as Theorem 3.1, for any $T \geq T^*$, where $T^* = \Theta\big(\frac{D^2 K}{\eta \sum_{m=1}^{M} \|\mathbf{v}_m^*\|_2^2}\big)$ as defined in Lemma D.17, it holds that

$$\frac{1}{K} - \frac{20D(D-K)}{\sqrt{\eta K \sum_{m=1}^{M} \|\mathbf{v}_m^*\|_2^2 (T - T^*)}} \leq p(T) \leq \frac{1}{K} - \frac{D(D-K)}{2e\sqrt{\eta K \sum_{m=1}^{M} \|\mathbf{v}_m^*\|_2^2 (T - T^*)}}. \quad \text{(D.34)}$$

In addition, it holds that

$$\left| p(T)C_1(T) - \frac{1}{K} \right| \leq \frac{2}{D}\big(1 - Kp(T)\big) + \frac{1}{K}\big(1 - Kp(T)\big)^2$$

$$\leq \frac{40K(D-K)}{\sqrt{\eta K \sum_{m=1}^{M} \|\mathbf{v}_m^*\|_2^2 (T - T^*)}} + \frac{400D^2(D-K)^2}{\eta \sum_{m=1}^{M} \|\mathbf{v}_m^*\|_2^2 (T - T^*)} \quad \text{(D.35)}$$

*Proof of Lemma D.18.* With the bounds established in Lemma D.17 and the fact that $\frac{1-Kp(t)}{p(t)} = \exp\big(-\frac{C_2(t)+C_3(t)}{\sqrt{D}}\big)$, the upper and lower bounds of $\Delta C_2(t) + \Delta C_3(t)$ obtained in (D.33) can be rewritten as

$$\Delta C_2(t) + \Delta C_3(t) \leq \eta \frac{8\pi \sum_{m=1}^{M} \|\mathbf{v}_m^*\|_2^2}{K^3 \sqrt{D^3}} e^{-\frac{2}{\sqrt{D}}\big(C_2(t)+C_3(t)\big)};$$

$$\Delta C_2(t) + \Delta C_3(t) \geq \eta \frac{\sum_{m=1}^{M} \|\mathbf{v}_m^*\|_2^2}{200K^3 \sqrt{D^3}} e^{-\frac{2}{\sqrt{D}}\big(C_2(t)+C_3(t)\big)}.$$

The upper and lower bounds of $\Delta C_2(t) + \Delta C_3(t)$ match the assumptions of Lemma F.12. By applying the lemma, we can obtain that for all $T \geq T^*$,

$$C_2(T) + C_3(T) \geq \frac{\sqrt{D}}{2} \log\left(\frac{\eta \sum_{m=1}^{M} \|\mathbf{v}_m^*\|_2^2}{200K^3 D^2}(T - T^*) + e^{\frac{2}{K\sqrt{D}}}\right);$$

$$C_2(T) + C_3(T) \leq \eta \frac{8\pi \sum_{m=1}^{M} \|\mathbf{v}_m^*\|_2^2}{K^3 \sqrt{D^3}} + \frac{\sqrt{D}}{2} \log\left(\frac{8\pi\eta \sum_{m=1}^{M} \|\mathbf{v}_m^*\|_2^2}{K^3 D^2}(T - T^*) + e^{\frac{2}{K\sqrt{D}}}\right).$$

Replacing this result into the formula of $p(T)$, we have

$$p(T) = \frac{1}{K + (D-K)\exp\big(-\frac{C_2(T)+C_3(T)}{\sqrt{D}}\big)} \geq \frac{1}{K + (D-K)\exp\big(-\frac{1}{2}\log\big(\frac{\eta \sum_{m=1}^{M} \|\mathbf{v}_m^*\|_2^2}{200K^3 D^2}(T - T^*)\big)\big)}$$

$$\geq \frac{1}{K} - \frac{20D(D-K)}{\sqrt{\eta K \sum_{m=1}^{M} \|\mathbf{v}_m^*\|_2^2 (T - T^*)}}$$

On the other hand, we can also derive that

$$p(T^*) \leq \frac{1}{K + (D-K)\exp\big(-\frac{1}{2}\log\big(\frac{8\pi \sum_{m=1}^{M} \|\mathbf{v}_m^*\|_2^2}{K^3 D^2}(T - T^*) + e^{\frac{2}{K\sqrt{D}}}\big) - \frac{\eta 4\pi \sum_{m=1}^{M} \|\mathbf{v}_m^*\|_2^2}{K^3 D^3}\big)}$$

$$\leq \frac{1}{K} - \frac{D(D-K)}{2e\sqrt{\eta K \sum_{m=1}^{M} \|\mathbf{v}_m^*\|_2^2 (T - T^*)}}.$$

This finishes the proof of (D.34). With this condition holds, by checking the definition of $C_1^*(T^*)$, we can obtain that

$$1 + \big(1 - Kp(T)\big)\left(\frac{1}{Kp(T)} - \frac{1 - Kp(T)}{(D-K)Kp(T)^2}\right) \leq C_1^*(T) \leq 1 + \frac{1 - Kp(T)}{Kp(T)}.$$

Plugging this result into (D.32), we derive that

$$1 + \big(1 - Kp(T)\big)\left(\frac{1}{Kp(T)} - \frac{2K}{D}\right) \leq C_1(T) \leq 1 + \big(1 - Kp(T)\big)\left(\frac{1}{Kp(T)} + \frac{2K}{D}\right),$$

which immediately leads to the final conclusion of (D.35).

$\square$

Now, we are ready to prove Theorem 3.1.

*Proof of Theorem 3.1.* We first prove the first conclusion.

$$\left\| \mathbf{S}^{(T)} - \mathbf{S}^* \right\|_F = \sqrt{\sum_{i_1=1}^{D} \sum_{i=1}^{D} \left( \mathbf{S}_{i_1,i}^{(T)} - \mathbf{S}_{i_1,i}^* \right)^2} = \sqrt{DK\left( \frac{1}{K} - p(T) \right)^2 + D(D-K)\frac{\left(1 - Kp(T)\right)^2}{(D-K)^2}}$$

$$= \frac{D}{\sqrt{K(D-K)}}\left(1 - Kp(T)\right) = \Theta\left( \frac{D^{\frac{5}{2}}}{\sqrt{\eta \sum_{m=1}^{M} \|\mathbf{v}_m^*\|_2^2 (T - T^*)}} \right),$$

where the last inequality holds by applying the upper and lower bounds of $p(T)$ derived in Lemma D.18. This finishes the first conclusion of Theorem 3.1. Notice that in Lemma D.18, we have derived that $|C_1(T) - 1| = \Theta(1 - Kp(T))$, which directly imply that

$$\left\| \mathbf{W}_V^{(T)} - \mathbf{V}^* \right\|_F = |C_1(T) - 1|\|\mathbf{V}^*\|_F = \Theta\left( D^2 \sqrt{\frac{K}{\eta \sum_{m=1}^{M} \|\mathbf{v}_m^*\|_2^2}(T - T^*)} \right) \cdot \|\mathbf{V}^*\|_F,$$

where the last inequality holds by applying the upper and lower bounds of $p(T)$ derived in Lemma D.18. This finishes the second conclusion of Theorem 3.1. For the third conclusion, notice that

$$\mathcal{L}(\mathbf{W}_V^{(T)}; \mathbf{W}_{KQ}^{(T)}) = \frac{1}{2} \sum_{m=1}^{M} \sum_{i=1}^{D} \mathbb{E}\left[ \left( \mathbf{Y}_{m,i} - \sigma\left( \sum_{i_1=1}^{D} \langle \mathbf{w}_{V,m}^{(T)}, \mathbf{x}_{i_1} \rangle \mathbf{S}_{i_1,i}^{(T)} \right) \right)^2 \right]$$

$$= \frac{1}{2} \sum_{m=1}^{M} \sum_{i=1}^{D} \mathbb{E}\left[ \left( [f^*(\mathbf{X})]_{m,i} - \sigma\left( \sum_{i_1=1}^{D} \langle \mathbf{w}_{V,m}^{(T)}, \mathbf{x}_{i_1} \rangle \mathbf{S}_{i_1,i}^{(T)} \right) \right)^2 \right] + \frac{1}{2}\mathbb{E}\left[ \|\mathcal{E}\|_F^2 \right],$$

where the last term is essential $\mathcal{L}_{\mathbf{opt}}$, and the last inequality holds by the independence between $\mathbf{X}$ and $\mathcal{E}$ and the fact that $\mathcal{E}$ is zero-mean. Since this equation holds, in the next, we directly deal with $\mathcal{L}(\mathbf{W}_V^{(T)}; \mathbf{W}_{KQ}^{(T)}) - \mathcal{L}_{\mathbf{opt}}$. We first prove the upper bound. By utilizing the fact that $|\sigma(x) - \sigma(y)| \leq |x - y|$ for all $x, y \in \mathbb{R}$, we can derive that

$$\mathcal{L}(\mathbf{W}_V^{(T)}; \mathbf{W}_{KQ}^{(T)}) - \mathcal{L}_{\mathbf{opt}} = \frac{1}{2} \sum_{m=1}^{M} \sum_{i=1}^{D} \mathbb{E}\left[ \left( [f^*(\mathbf{X})]_{m,i} - \sigma\left( \sum_{i_1=1}^{D} \langle \mathbf{w}_{V,m}^{(T)}, \mathbf{x}_{i_1} \rangle \mathbf{S}_{i_1,i}^{(T)} \right) \right)^2 \right]$$

$$\leq \frac{1}{2} \sum_{m=1}^{M} \sum_{i=1}^{D} \mathbb{E}\left[ \left( \underbrace{\left( \frac{1}{K} - C_1(T)p(T) \right) \sum_{i_1 \in G^i} \langle \mathbf{v}_m^*, \mathbf{x}_{i_1} \rangle}_{Z_{1,i,m}^{(T)}} + \underbrace{\frac{C_1(T)\left(1 - Kp(T)\right)}{D-K} \sum_{i_1 \notin G^i} \langle \mathbf{v}_m^*, \mathbf{x}_{i_1} \rangle}_{Z_{2,i,m}^{(T)}} \right)^2 \right].$$

Notice that $Z_{1,i,m}^{(T)} \sim \mathcal{N}(0, \sigma_{1,m}^2)$, where $\sigma_{1,m}^2 = K\|\mathbf{v}_m^*\|_2^2\left( \frac{1}{K} - C_1(T)p(T) \right)^2$, and $Z_{2,i,m}^{(T)} \sim \mathcal{N}(0, \sigma_{2,m}^2)$, where $\sigma_{2,m}^2 = \frac{\|\mathbf{v}_m^*\|_2^2 C_1(T)^2 (1 - Kp(T))^2}{D-K}$, and they are independent. Based on the upper bounds derived in Lemma D.18, we can finally derive that

$$\mathcal{L}(\mathbf{W}_V^{(T)}; \mathbf{W}_{KQ}^{(T)}) - \mathcal{L}_{\mathbf{opt}} \leq \frac{1}{2} \sum_{m=1}^{M} \sum_{i=1}^{D} \mathbb{E}\left[ \left( Z_{1,i,m}^{(T)} + Z_{2,i,m}^{(T)} \right)^2 \right] = \frac{D}{2} \sum_{m=1}^{M} \sigma_{1,m}^2 + \frac{D}{2} \sum_{m=1}^{M} \sigma_{2,m}^2$$

$$= \frac{DK}{2}\left( \frac{1}{K} - C_1(T)p(T) \right)^2 \sum_{m=1}^{M} \|\mathbf{v}_m\|_2^2 + \frac{DC_1(T)^2\left(1 - Kp(T)\right)^2}{2(D-K)} \sum_{m=1}^{M} \|\mathbf{v}_m\|_2^2$$

$$\leq \bar{c}\frac{KD^4}{\eta(T - T^*)}.$$

where the last inequality holds by applying the upper bounds for $\left( \frac{1}{K} - C_1(T)p(T) \right)^2$, and $p(T)$ derived in Lemma D.18 This completes the proof for upper bound. On the other hand, denote $Z_{3,m,i} = \sum_{i_1 \in G^i} \langle \mathbf{v}_m^*, \mathbf{x}_i \rangle \sim \mathcal{N}(0, K\|\mathbf{v}_m\|_2^2)$ and $Z_{4,m,i} = \sum_{i_1 \notin G^i} \langle \mathbf{v}_m^*, \mathbf{x}_i \rangle \sim \mathcal{N}\left(0, (D-K)\|\mathbf{v}_m\|_2^2\right)$,

and $Z_{5,m,i}^{(T)} = p(T)Z_{3,m,i} + \frac{1-Kp(T)}{D-K}Z_{4,m,i}$. Then, by utilizing the fact that $|\sigma(x) - \sigma(y)| \geq |x-y| \cdot \mathbb{1}_{\{x \geq 0, y \geq 0\}}$, we can further derive that

$$
\mathcal{L}(\mathbf{W}_V^{(T)}; \mathbf{W}_{KQ}^{(T)}) - \mathcal{L}_{\mathbf{opt}}
$$

$$
= \frac{1}{2}\sum_{m=1}^{M}\sum_{i=1}^{D}\mathbb{E}\left[\left(\left[f^*(\mathbf{X})\right]_{m,i} - \sigma\left(\sum_{i_1=1}^{D}\langle\mathbf{w}_{V,m}^{(T)}, \mathbf{x}_{i_1}\rangle\mathbf{S}_{i_1,i}^{(T)}\right)\right)^2\right]
$$

$$
= \frac{1}{2}\sum_{m=1}^{M}\sum_{i=1}^{D}\mathbb{E}\left[\left(\sigma\left(\frac{Z_{3,m,i}}{K}\right) - \sigma\left(C_1(T)Z_{5,m,i}^{(T)}\right)\right)^2\right]
$$

$$
\geq \frac{1}{2}\sum_{m=1}^{M}\sum_{i=1}^{D}\mathbb{E}\left[\left(\left(\frac{1}{K} - C_1(T)p(T)\right)Z_{3,m,i} - \frac{C_1(T)(1-Kp(T))}{D-K}Z_{4,m,i}\right)^2\mathbb{1}_{\{Z_{3,m,i}\geq 0\}}\mathbb{1}_{\{Z_{5,m,i}^{(T)}\geq 0\}}\right]
$$

$$
\geq \frac{1}{2}\sum_{m=1}^{M}\sum_{i=1}^{D}\mathbb{E}\left[\left(\left(\frac{1}{K} - C_1(T)p(T)\right)Z_{3,m,i} - \frac{C_1(T)(1-Kp(T))}{D-K}Z_{4,m,i}\right)^2\mathbb{1}_{\{Z_{3,m,i}\geq 0\}}\mathbb{1}_{\{Z_{4,m,i}\geq 0\}}\mathbb{1}_{\{Z_{5,m,i}^{(T)}\geq 0\}}\right]
$$

$$
= \frac{1}{2}\sum_{m=1}^{M}\sum_{i=1}^{D}\mathbb{E}\left[\left(\left(\frac{1}{K} - C_1(T)p(T)\right)Z_{3,m,i} - \frac{C_1(T)(1-Kp(T))}{D-K}Z_{4,m,i}\right)^2\mathbb{1}_{\{Z_{3,m,i}\geq 0\}}\mathbb{1}_{\{Z_{4,m,i}\geq 0\}}\right]
$$

$$
= \frac{1}{2}\sum_{m=1}^{M}\sum_{i=1}^{D}\frac{\left(\frac{1}{K} - C_1(T)p(T)\right)^2}{4}\mathbb{E}[Z_{3,m,i}^2] + \frac{1}{2}\sum_{m=1}^{M}\sum_{i=1}^{D}\frac{C_1(T)^2(1-Kp(T))^2}{4(D-K)^2}\mathbb{E}[Z_{4,m,i}^2]
$$

$$
- \sum_{m=1}^{M}\sum_{i=1}^{D}\frac{C_1(T)\left|\frac{1}{K} - C_1(T)p(T)\right|(1-Kp(T))}{D-K}\mathbb{E}\left[Z_{3,m,i}\mathbb{1}_{\{Z_{3,m,i}\geq 0\}}\right]\mathbb{E}\left[Z_{4,m,i}\mathbb{1}_{\{Z_{4,m,i}\geq 0\}}\right]
$$

$$
\geq \frac{\sum_{m=1}^{M}\|\mathbf{v}_m\|_2^2 DC_1(T)^2(1-Kp(T))^2}{8(D-K)}
$$

$$
- \frac{\sum_{m=1}^{M}\|\mathbf{v}_m\|_2^2 DC_1(T)\left|\frac{1}{K} - C_1(T)p(T)\right|(1-Kp(T))\sqrt{K(D-K)}}{2\pi(D-K)}
$$

$$
\geq \frac{\sum_{m=1}^{M}\|\mathbf{v}_m\|_2^2 DC_1(T)(1-Kp(T))^2}{2(D-K)}\left(\frac{C_1(T)}{4} - \frac{2\sqrt{K(D-K)}}{\pi D}\right)
$$

$$
\geq \frac{\sum_{m=1}^{M}\|\mathbf{v}_m\|_2^2 D(1-Kp(T))^2}{16(D-K)} \geq c\frac{KD^4}{\eta(T-T^*)},
$$

where the last inequality holds by applying the lower bound of $1 - Kp(T)$ demonstrated in Lemma D.18. This completes the proof. □

# E  PROOF OF THEOREM 3.2 AND DISCUSSION OF THE WORST CASE EXAMPLE

In this section, we provide a complete proof for Theorem 3.2, and a worst-case example can attain the upper bound in Theorem 3.2. We first prove Theorem 3.2 in the following.

*Proof of Theorem 3.2.* We first upper bound the OOD loss by the sum of three terms as

$$
\mathcal{L}_{\mathbf{OOD}}(\mathbf{W}_V^{(T)}; \mathbf{W}_{KQ}^{(T)}) = \frac{1}{2}\mathbb{E}\left[\|\widetilde{\mathbf{Y}} - \mathrm{TF}(\widetilde{\mathbf{Z}}; \mathbf{W}_V^{(T)}; \mathbf{W}_{KQ}^{(T)})\|_F^2\right]
$$

$$
= \frac{1}{2}\mathbb{E}\left[\|\widetilde{\mathbf{Y}} - f^*(\widetilde{\mathbf{X}}) + f^*(\widetilde{\mathbf{X}}) - \mathrm{TF}(\widetilde{\mathbf{Z}}; \mathbf{W}_V^{(T)}; \mathbf{W}_{KQ}^{(T)})\|_F^2\right]
$$

$$
= \frac{1}{2}\mathbb{E}\left[\|\widetilde{\mathbf{Y}} - f^*(\widetilde{\mathbf{X}})\|_F^2\right] + \frac{1}{2}\mathbb{E}\left[\|f^*(\widetilde{\mathbf{X}}) - \mathrm{TF}(\widetilde{\mathbf{Z}}; \mathbf{W}_V^{(T)}; \mathbf{W}_{KQ}^{(T)})\|_F^2\right]
$$

$$
+ \mathbb{E}\left[\langle\widetilde{\mathbf{Y}} - f^*(\widetilde{\mathbf{X}}), f^*(\widetilde{\mathbf{X}}) - \mathrm{TF}(\widetilde{\mathbf{Z}}; \mathbf{W}_V^{(T)}; \mathbf{W}_{KQ}^{(T)})\rangle\right]
$$

$$
\leq \frac{1}{2}\mathbb{E}\left[\|\widetilde{\mathbf{Y}} - f^*(\widetilde{\mathbf{X}})\|_F^2\right] + \frac{1}{2}\mathbb{E}\left[\|f^*(\widetilde{\mathbf{X}}) - \mathrm{TF}(\widetilde{\mathbf{Z}}; \mathbf{W}_V^{(T)}; \mathbf{W}_{KQ}^{(T)})\|_F^2\right]
$$

$$+ \sqrt{\mathbb{E}\big[\|\widetilde{\mathbf{Y}} - f^*(\widetilde{\mathbf{X}})\|_F^2\big] \mathbb{E}\big[\|f^*(\widetilde{\mathbf{X}}) - \mathrm{TF}(\widetilde{\mathbf{Z}}; \mathbf{W}_V^{(T)}; \mathbf{W}_{KQ}^{(T)})\|_F^2\big]},$$

where the last inequality holds by Cauchy-Schwarz inequality. Based on this decomposition, it is critical to derive the upper bound for $\mathbb{E}\big[\|\widetilde{\mathbf{Y}} - f^*(\widetilde{\mathbf{X}})\|_F^2\big]$ and $\mathbb{E}\big[\|f^*(\widetilde{\mathbf{X}}) - \mathrm{TF}(\widetilde{\mathbf{Z}}; \mathbf{W}_V^{(T)}; \mathbf{W}_{KQ}^{(T)})\|_F^2\big]$. For the first term $\mathbb{E}\big[\|\widetilde{\mathbf{Y}} - f^*(\widetilde{\mathbf{X}})\|_F^2\big]$, we have

$$\mathbb{E}\big[\|\widetilde{\mathbf{Y}} - f^*(\widetilde{\mathbf{X}})\|_F^2\big] \leq 2\mathbb{E}\big[\|\widetilde{\mathbf{Y}}\|_F^2\big] + 2\mathbb{E}\big[\|f^*(\widetilde{\mathbf{X}})\|_F^2\big].$$

By the assumption that each column of $\widetilde{\mathbf{Y}}$ satisfying that $\mathbb{E}[\|\widetilde{\mathbf{y}}_m\|_2^2] \leq \xi$, it is straightforward that $\mathbb{E}\big[\|\widetilde{\mathbf{Y}}\|_F^2\big] \leq D\xi$. On the other hand, we have

$$\mathbb{E}\big[\|f^*(\widetilde{\mathbf{X}})\|_F^2\big] = \sum_{i=1}^{D} \sum_{m=1}^{M} \mathbb{E}\Big[\big[f^*(\widetilde{\mathbf{X}})\big]_{m,i}^2\Big] \leq \sum_{i=1}^{D} \sum_{m=1}^{M} \sum_{i' \in G^i} \frac{\|\mathbf{v}_m^*\|_2^2}{K} \mathbb{E}[\langle \mathbf{v}_m^* / \|\mathbf{v}_m^*\|_2, \widetilde{\mathbf{x}}_{i'} \rangle^2]$$

$$\leq \sum_{i=1}^{D} \sum_{m=1}^{M} \sum_{i' \in G^i} \frac{\mathbb{E}[\|\widetilde{\mathbf{x}}_{i'}\|_2^2]\|\mathbf{v}_m^*\|_2^2}{K} \leq D\xi \sum_{m=1}^{M} \|\mathbf{v}_m^*\|_2^2.$$

For the second term $\mathbb{E}\big[\|f^*(\widetilde{\mathbf{X}}) - \mathrm{TF}(\widetilde{\mathbf{Z}}; \mathbf{W}_V^{(T)}; \mathbf{W}_{KQ}^{(T)})\|_F^2\big]$, we can derive that

$$\mathbb{E}\big[\|f^*(\widetilde{\mathbf{X}}) - \mathrm{TF}(\widetilde{\mathbf{Z}}; \mathbf{W}_V^{(T)}; \mathbf{W}_{KQ}^{(T)})\|_F^2\big]$$

$$\leq \sum_{m=1}^{M} \sum_{i=1}^{D} \mathbb{E}\Bigg[\bigg(\Big(\frac{1}{K} - C_1(T)p(T)\Big) \sum_{i_1 \in G^i} \langle \mathbf{v}_m^*, \widetilde{\mathbf{x}}_{i_1} \rangle + \frac{C_1(T)\big(1 - Kp(T)\big)}{D - K} \sum_{i_1 \notin G^i} \langle \mathbf{v}_m^*, \widetilde{\mathbf{x}}_{i_1} \rangle\bigg)^2\Bigg]$$

$$\leq D \sum_{m=1}^{M} \sum_{i=1}^{D} \|\mathbf{v}_m^*\|_2^2 \Big(\frac{1}{K} - C_1(T)p(T)\Big)^2 \sum_{i_1 \in G^i} \mathbb{E}[\|\widetilde{\mathbf{x}}_{i_1}\|_2^2]$$

$$+ D \sum_{m=1}^{M} \sum_{i=1}^{D} \|\mathbf{v}_m^*\|_2^2 \frac{C_1(T)^2\big(1 - Kp(T)\big)^2}{(D - K)^2} \sum_{i_1 \notin G^i} \mathbb{E}[\|\widetilde{\mathbf{x}}_{i_1}\|_2^2]$$

$$\leq \mathcal{O}\bigg(\frac{KD^5\xi}{\eta(T - T^*)}\bigg).$$

Here the first inequality holds by $|\sigma(x) - \sigma(y)| \leq |x - y|$. The second inequality is established by the fact $(\sum_{i=1}^{D} a_i)^2 \leq D \sum_{i=1}^{D} a_i^2$ for all scalar $a_i$'s and $\langle \mathbf{v}_m^*, \widetilde{\mathbf{x}}_{i_1} \rangle^2 \leq \|\mathbf{v}_m^*\|^2 \|\widetilde{\mathbf{x}}_{i_1}\|_2^2$. Lastly, the third inequality is derived by replacing the conclusions in Lemma D.18. Combining all these derived terms into the three terms derived as the upper bound for OOD loss, we have,

$$\mathcal{L}_{\mathbf{OOD}}(\mathbf{W}_V^{(T)}; \mathbf{W}_{KQ}^{(T)}) - \frac{1}{2}\mathbb{E}\big[\|\widetilde{\mathbf{Y}} - f^*(\widetilde{\mathbf{X}})\|_F^2\big] \leq \mathcal{O}\bigg(D^3\xi\sqrt{\frac{K\sum_{m=1}^{M}\|\mathbf{v}_m^*\|_2^2}{\eta(T - T^*)}} + \frac{KD^5\xi}{\eta(T - T^*)}\bigg).$$

Let the upper bound derived above smaller than $\epsilon$, we can derive that

$$T_\epsilon = T^* + \mathcal{O}\bigg(\frac{KD^6\xi^2\sum_{m=1}^{M}\|\mathbf{v}_m^*\|_2^2}{\eta\epsilon^2}\bigg) = \mathcal{O}\bigg(\frac{KD^6\xi^2\sum_{m=1}^{M}\|\mathbf{v}_m^*\|_2^2}{\eta\epsilon^2}\bigg).$$

This completes the proof. $\qquad\square$

In the next, we discuss the construction of the worst case $\widetilde{\mathbf{Y}}$, such that $\mathcal{L}_{\mathbf{OOD}}(\mathbf{W}_V^{(T_\epsilon)}; \mathbf{W}_{KQ}^{(T_\epsilon)}) - \frac{1}{2}\mathbb{E}\big[\|\widetilde{\mathbf{Y}} - f^*(\widetilde{\mathbf{X}})\|_F^2\big] \geq \epsilon$ for some $T_\epsilon = \Theta\big(\frac{MKD^6}{\eta\epsilon^2}\big)$ (assuming $\|\mathbf{v}_m\|_2 = 1$ and $\xi = \Theta(1)$ for simplicity). In fact, this $T_\epsilon$ can be different with the $T_\epsilon$ defined in Theorem 3.2, but at the same order w.r.t. $\epsilon$, hence a matching result.

By the conclusions in Lemma D.18, we know that $\frac{1}{K} - p(T) = \Theta(\frac{D^2}{\sqrt{\eta KMT}})$ and $\big|p(T)C_1(T) - \frac{1}{K}\big| \leq \mathcal{O}(\frac{D}{\sqrt{\eta KMT}})$. Therefore, there exists an absolute constant $c'$ such that $\frac{1 - Kp(T)}{|p(T)C_1(T) - \frac{1}{K}|} \geq c'D$. In addition, we let $A_{m,i}$ to denote the event such that $\big|\sum_{i_1 \notin G^i} \langle \mathbf{v}_m^*, \widetilde{\mathbf{x}}_{i_1} \rangle\big| \geq$

$\max\{\frac{2}{c'}|\sum_{i_1 \in G^i}\langle \mathbf{v}_m^*, \widetilde{\mathbf{x}}_{i_1}\rangle|, 1\}$. We can assume the probability of $A_{m,i}$ is larger than an absolute constant. In fact, such an assumption can be easily verified on many specific distributions like Gaussian distributions. With these notations in hand, we can design $\widetilde{\mathbf{Y}}$ such that its $(m, i)$-th entry is generates as $\widetilde{\mathbf{Y}}_{m,i} = \text{sign}(\sum_{i_1 \notin G^i}\langle \mathbf{v}_m^*, \widetilde{\mathbf{x}}_{i_1}\rangle) \cdot \mathbb{1}_{\{A_{m,i}\}} + f^*(\widetilde{\mathbf{X}})_{m,i}$. Given this construction, we can deduce that

$$\mathbb{E}\big[\langle \widetilde{\mathbf{Y}} - f^*(\widetilde{\mathbf{X}}), f^*(\widetilde{\mathbf{X}}) - \text{TF}(\widetilde{\mathbf{Z}}; \mathbf{W}_V^{(T)}; \mathbf{W}_{KQ}^{(T)})\rangle\big]$$

$$= \sum_{m=1}^M \sum_{i=1}^D \mathbb{E}\Bigg[\Big(\widetilde{\mathbf{Y}}_{m,i} - f^*(\widetilde{\mathbf{X}})_{m,i}\Big)\Big(\Big(\frac{1}{K} - C_1(T)p(T)\Big)\sum_{i_1 \in G^i}\langle \mathbf{v}_m^*, \widetilde{\mathbf{x}}_{i_1}\rangle$$

$$+ \frac{C_1(T)\big(1 - Kp(T)\big)}{D - K}\sum_{i_1 \notin G^i}\langle \mathbf{v}_m^*, \widetilde{\mathbf{x}}_{i_1}\rangle\Big)\Bigg]$$

$$\geq \frac{1}{2}\sum_{m=1}^M \sum_{i=1}^D \mathbb{E}\Bigg[\frac{C_1(T)\big(1 - Kp(T)\big)}{D - K}\Big|\sum_{i_1 \notin G^i}\langle \mathbf{v}_m^*, \widetilde{\mathbf{x}}_{i_1}\rangle\Big|\mathbb{1}_{\{A_{m,i}\}}\Bigg]$$

$$\geq \frac{D^3}{2}\sqrt{\frac{MK}{\eta T}}\mathbb{E}[\mathbb{1}_{\{A_{m,i}\}}] = \Theta\Big(D^3\sqrt{\frac{MK}{\eta T}}\Big).$$

Replacing the $T$ with $T_\epsilon = \Theta\big(\frac{MKD^6}{\eta \epsilon^2}\big)$, we can finally conclude that

$$\mathcal{L}_{\mathbf{OOD}}(\mathbf{W}_V^{(T_\epsilon)}; \mathbf{W}_{KQ}^{(T_\epsilon)}) - \frac{1}{2}\mathbb{E}\big[\|\widetilde{\mathbf{Y}} - f^*(\widetilde{\mathbf{X}})\|_F^2\big]$$

$$\geq \mathbb{E}\big[\langle \widetilde{\mathbf{Y}} - f^*(\widetilde{\mathbf{X}}), f^*(\widetilde{\mathbf{X}}) - \text{TF}(\widetilde{\mathbf{Z}}; \mathbf{W}_V^{(T_\epsilon)}; \mathbf{W}_{KQ}^{(T_\epsilon)})\rangle\big] \geq \Theta\Big(D^3\sqrt{\frac{MK}{\eta}}\frac{\epsilon}{D^3}\sqrt{\frac{\eta}{MK}}\Big) = \Theta(\epsilon).$$

This validates that the upper bound is indeed attained under our construction.

# F  TECHNICAL LEMMAS

In this section, we present and prove the technical lemmas we used in the proof of the previous sections.

## F.1  CALCULATION DETAILS OF EXPECTATIONS

We introduce the details regarding

**Lemma F.1** (Calculation of $F_1(a)$ defined in (D.2)). Let $x \sim \mathcal{N}(0, a)$, then it holds that

- If $\sigma(\cdot)$ is the identity map, then

$$\mathbb{E}[x\sigma(x)\sigma'(x)] = a.$$

- If $\sigma(\cdot)$ is ReLU activation function, then

$$\mathbb{E}[x\sigma(x)\sigma'(x)] = \frac{a}{2}.$$

- If $\sigma(\cdot)$ is Leaky ReLU activation function, then

$$\mathbb{E}[x\sigma(x)\sigma'(x)] = \frac{(1 + \kappa^2)a}{2}.$$

Here, $\kappa$ is the coefficient of the Leaky ReLU activation function when the input is smaller than 0.

*Proof of Lemma F.1.* The first conclusion for the identity map is straightforward. When $\sigma(\cdot)$ is the ReLU activation function, we can rewrite that $x\sigma(x)\sigma'(x) = x \cdot x\mathbb{1}_{\{x \geq 0\}} \cdot \mathbb{1}_{\{x \geq 0\}} = x^2\mathbb{1}_{\{x \geq 0\}}$. Therefore, we have,

$$\mathbb{E}[x\sigma(x)\sigma'(x)] = \mathbb{E}[x^2\mathbb{1}_{\{x \geq 0\}}] = \frac{\mathbb{E}[x^2]}{2} = \frac{a}{2}.$$

Besides, when $\sigma(\cdot)$ is the Leaky ReLU activation function, we can rewrite that $x\sigma(x)\sigma'(x) = x^2 \mathbb{1}_{\{x \geq 0\}} + \kappa^2 x^2 \mathbb{1}_{\{x < 0\}}$. Therefore, we have,

$$\mathbb{E}[x\sigma(x)\sigma'(x)] = \mathbb{E}[x^2 \mathbb{1}_{\{x \geq 0\}}] + \kappa^2 \mathbb{E}[x^2 \mathbb{1}_{\{x < 0\}}] = \frac{(1+\kappa)^2 \mathbb{E}[x^2]}{2} = \frac{(1+\kappa)^2 a}{2},$$

which finishes the proof. $\qquad\square$

**Lemma F.2** (Calculation of $F_2(a, b)$ defined in (D.3))**.** Let $x_1 \sim \mathcal{N}(0, a)$, $x_2 \sim \mathcal{N}(0, b)$ be two independent Gaussian random variables, then it holds that

- If $\sigma(\cdot)$ is the identity map, then

$$\mathbb{E}[x_1\sigma(x_1 + x_2)\sigma'(x_1 + x_2)] = a.$$

- If $\sigma(\cdot)$ is ReLU activation function, then

$$\mathbb{E}[x_1\sigma(x_1 + x_2)\sigma'(x_1 + x_2)] = \frac{a}{2}.$$

- If $\sigma(\cdot)$ is Leaky ReLU activation function, then

$$\mathbb{E}[x_1\sigma(x_1 + x_2)\sigma'(x_1 + x_2)] = \frac{(1 + \kappa^2)a}{2}.$$

Here, $\kappa$ is the coefficient of the Leaky ReLU activation function when the input is smaller than 0.

*Proof of Lemma F.2.* The first conclusion for the identity map is straightforward. For the next two cases, we first introduce some definitions. Let $x_3 = x_1 + x_2 \sim \mathcal{N}(0, a + b)$. Then we have $\mathrm{Cov}(x_1, x_3) = \mathbb{E}[(x_1 + x_2)x_1] = a$, and $\mathbb{E}[x_1|x_3] = \frac{a}{a+b}x_3$. Consequently, when $\sigma(\cdot)$ is the ReLU activation function,

$$\mathbb{E}[x_1\sigma(x_1 + x_2)\sigma'(x_1 + x_2)] = \mathbb{E}[x_1 x_3 \mathbb{1}_{\{x_3 \geq 0\}}] = \mathbb{E}\big[\mathbb{E}[x_1 x_3 \mathbb{1}_{\{x_3 \geq 0\}}|x_3]\big]$$
$$= \frac{a}{a+b}\mathbb{E}[x_3^2 \mathbb{1}_{\{x_3 \geq 0\}}] = \frac{a}{2(a+b)}\mathbb{E}[x_3^2] = \frac{a}{2}.$$

In addition, when $\sigma(\cdot)$ is the Leaky ReLU activation function,

$$\mathbb{E}[x_1\sigma(x_1 + x_2)\sigma'(x_1 + x_2)] = \mathbb{E}[x_1 x_3 \mathbb{1}_{\{x_3 \geq 0\}}] + \kappa^2 \mathbb{E}[x_1 x_3 \mathbb{1}_{\{x_3 < 0\}}]$$
$$= \mathbb{E}\big[\mathbb{E}[x_1 x_3 \mathbb{1}_{\{x_3 \geq 0\}}|x_3]\big] + \kappa^2 \mathbb{E}\big[\mathbb{E}[x_1 x_3 \mathbb{1}_{\{x_3 < 0\}}|x_3]\big]$$
$$= \frac{a}{a+b}\mathbb{E}[x_3^2 \mathbb{1}_{\{x_3 \geq 0\}}] + \frac{\kappa^2 a}{a+b}\mathbb{E}[x_3^2 \mathbb{1}_{\{x_3 < 0\}}] = \frac{(1 + \kappa^2)a}{2}.$$

This completes the proof. $\qquad\square$

**Lemma F.3.** Let $x_1 \sim \mathcal{N}(0, a)$, $x_2 \sim \mathcal{N}(0, b)$ be two independent Gaussian random variables, then it holds that

- If $\sigma(\cdot)$ is the identity map, then

$$\mathbb{E}[x_1\sigma(x_1)\sigma'(x_1 + x_2)] = a.$$

- If $\sigma(\cdot)$ is ReLU activation function, then

$$\mathbb{E}[x_1\sigma(x_1)\sigma'(x_1 + x_2)] = \frac{a}{4} + \frac{a}{2\pi}\left(\arctan\left(\sqrt{\frac{a}{b}}\right) + \frac{\sqrt{ab}}{a+b}\right). \tag{F.1}$$

And there exist the following matching lower and upper bounds:

$$\left(\frac{a}{4} + \frac{a\sqrt{ab}}{2\pi(a+b)}\right) \vee \left(\frac{a}{2} - \frac{b\sqrt{ab}}{2\pi(a+b)}\right) \leq \mathbb{E}[x_1\sigma(x_1)\sigma'(x_1 + x_2)] \leq \frac{a}{2}. \tag{F.2}$$

• If $\sigma(\cdot)$ is Leaky ReLU activation function, then

$$\mathbb{E}[x_1\sigma(x_1)\sigma'(x_1 + x_2)] = \frac{(1+\kappa)^2 a}{4} + \frac{(1-\kappa)^2 a}{2\pi}\left(\arctan\left(\sqrt{\frac{a}{b}}\right) + \frac{\sqrt{ab}}{a+b}\right). \tag{F.3}$$

And there exist the following matching lower and upper bounds:

$$\mathbb{E}[x_1\sigma(x_1)\sigma'(x_1 + x_2)] \leq \frac{(1+\kappa^2)a}{2};$$

$$\mathbb{E}[x_1\sigma(x_1)\sigma'(x_1 + x_2)] \geq \left(\frac{(1+\kappa)^2 a}{4} + \frac{(1-\kappa)^2 a\sqrt{ab}}{2\pi(a+b)}\right) \vee \left(\frac{(1+\kappa^2)a}{2} - \frac{(1-\kappa)^2 b\sqrt{ab}}{2\pi(a+b)}\right).$$
$$\tag{F.4}$$

Here, $\kappa$ is the coefficient of the Leaky ReLU activation function when the input is smaller than 0.

*Proof of Lemma F.3.* The first conclusion for the identity map is straightforward. When $\sigma(\cdot)$ is ReLU activation function, we can rewrite that $x_1\sigma(x_1)\sigma'(x_1 + x_2) = x_1^2 \mathbb{1}_{\{x_1 \geq 0\}} \mathbb{1}_{\{x_1 + x_2 \geq 0\}}$. Let $z_1 = \frac{x_1}{\sqrt{a}}$ and $z_2 = \frac{x_2}{\sqrt{b}}$, then we have,

$$\mathbb{E}[x_1\sigma(x_1)\sigma'(x_1 + x_2)] = a \underbrace{\mathbb{E}[z_1^2 \mathbb{1}_{\{z_1 \geq 0\}} \mathbb{1}_{\{\sqrt{a}z_1 + \sqrt{b}z_2 \geq 0\}}]}_{I}. \tag{F.5}$$

For $I$, by denoting $\lambda = \sqrt{\frac{a}{b}}$, we can obtain that

$$I = \int_0^\infty \int_{-\lambda z_1}^\infty z_1^2 \phi(z_1)\phi(z_2)\mathrm{d}z_1\mathrm{d}z_2 = \int_0^\infty z_1^2 \Phi(\lambda z_1)\phi(z_1)\mathrm{d}z_1,$$

where $\phi(\cdot)$ and $\Phi(\cdot)$ are the cumulative distribution function (c.d.f.) and probability density function (p.d.f.) for the standard Gaussian distribution respectively. We can denote that $I(\lambda) = \int_0^\infty z_1^2 \Phi(\lambda z_1)\phi(z_1)\mathrm{d}z_1$. Then, by the Leibniz integral rule, we have

$$\frac{\mathrm{d}I(\lambda)}{\mathrm{d}\lambda} = \int_0^\infty z_1^3 \phi(\lambda z_1)\phi(z_1)\mathrm{d}z_1 = \frac{1}{2\pi(1+\lambda^2)^2}\int_0^\infty z^3 e^{-\frac{z^2}{2}}\mathrm{d}z = \frac{1}{\pi(1+\lambda^2)^2}.$$

Additionally, since $I(0) = \frac{1}{4}$, we can derive that

$$I = \frac{1}{4} + \frac{1}{2\pi}\left(\arctan\lambda + \frac{\lambda}{1+\lambda^2}\right) = \frac{1}{4} + \frac{1}{2\pi}\left(\arctan\left(\sqrt{\frac{a}{b}}\right) + \frac{\sqrt{ab}}{a+b}\right) \tag{F.6}$$

Applying the result of (F.6) into (F.5), we finishes the proof of (F.1). In the next, we derive the upper and lower bound for $I_1$. By the property of c.d.f., we know that $\Phi(z) \leq 1$ for all $z \in \mathbb{R}$, which implies that

$$I \leq \int_0^\infty z_1^2 \phi(z_1)\mathrm{d}z_1 = \frac{1}{2}\mathbb{E}[z_1^2] = \frac{1}{2}.$$

Additionally, by Mills ratio, we further obtain $1 - \Phi(z) \leq \phi(z)/z$ for all $z > 0$. Based on this result, we can obtain that

$$I \geq \int_0^\infty z_1^2 \phi(z_1)\left(1 - \frac{\phi(\lambda z_1)}{\lambda z_1}\right)\mathrm{d}z_1 = \frac{1}{2} - \frac{1}{\lambda}\int_0^\infty z_1 \phi(z_1)\phi(\lambda z_1)\mathrm{d}z_1,$$

where the second term can be calculated by

$$\int_0^\infty z_1 \phi(z_1)\phi(\lambda z_1)\mathrm{d}z_1 = \frac{1}{2\pi}\int_0^\infty z_1 e^{-\frac{z_1^2(1+\lambda^2)}{2}}\mathrm{d}z_1 = \frac{1}{2\pi(1+\lambda^2)}\int_0^\infty z_1 e^{-\frac{z_1^2}{2}}\mathrm{d}z_1 = \frac{1}{2\pi(1+\lambda^2)}.$$

Plugging this result into the preceding inequality, we can derive that

$$I \geq \frac{1}{2} - \frac{b^{\frac{3}{2}}}{2\pi\sqrt{a}(a+b)}.$$

Combining all these results and (F.6), we finally conclude that

$$\left( \frac{1}{4} + \frac{\sqrt{ab}}{2\pi(a+b)} \right) \vee \left( \frac{1}{2} - \frac{b^{\frac{3}{2}}}{2\pi\sqrt{a}(a+b)} \right) \leq I \leq \frac{1}{2}. \tag{F.7}$$

Applying the result of (F.7) into (F.5), we finishes the proof of (F.2). In addition, when $\sigma(\cdot)$ is the Leaky ReLU activation function, we can similarly derive that

$$\begin{aligned}
\mathbb{E}[x_1\sigma(x_1)\sigma'(x_1+x_2)] &= a\mathbb{E}[z_1^2\mathbb{1}_{\{z_1\geq 0\}}\mathbb{1}_{\{\sqrt{a}z_1+\sqrt{b}z_2\geq 0\}}] + a\kappa\mathbb{E}[z_1^2\mathbb{1}_{\{z_1<0\}}\mathbb{1}_{\{\sqrt{a}z_1+\sqrt{b}z_2\geq 0\}}] \\
&\quad + a\kappa\mathbb{E}[z_1^2\mathbb{1}_{\{z_1\geq 0\}}\mathbb{1}_{\{\sqrt{a}z_1+\sqrt{b}z_2<0\}}] + a\kappa^2\mathbb{E}[z_1^2\mathbb{1}_{\{z_1<0\}}\mathbb{1}_{\{\sqrt{a}z_1+\sqrt{b}z_2<0\}}] \\
&= (1+\kappa^2)a\mathbb{E}[z_1^2\mathbb{1}_{\{z_1\geq 0\}}\mathbb{1}_{\{\sqrt{a}z_1+\sqrt{b}z_2\geq 0\}}] + 2\kappa a\mathbb{E}[z_1^2\mathbb{1}_{\{z_1<0\}}\mathbb{1}_{\{\sqrt{a}z_1+\sqrt{b}z_2\geq 0\}}],
\end{aligned}$$

where the last equality holds by the symmetry of $z_1$ and $z_2$. By applying a very similar calculation process, we can obtain that

$$\mathbb{E}[z_1^2\mathbb{1}_{\{z_1<0\}}\mathbb{1}_{\{\sqrt{a}z_1+\sqrt{b}z_2\geq 0\}}] = \int_{-\infty}^{0} z_1^2\Phi(\lambda z_1)\phi(z_1)\mathrm{d}z_1 = \frac{1}{4} - \frac{1}{2\pi}\left( \arctan\left(\sqrt{\frac{a}{b}}\right) + \frac{\sqrt{ab}}{a+b} \right).$$

By replacing this result into the previous calculation, we can immediately prove (F.3). And (F.4) can be directly derived from (F.2). □

**Lemma F.4.** Let $x_1 \sim \mathcal{N}(0,a)$, $x_2 \sim \mathcal{N}(0,b)$ be two independent Gaussian random variables, then it holds that

- If $\sigma(\cdot)$ is the identity map, then

$$\mathbb{E}[x_2\sigma(x_1)\sigma'(x_1+x_2)] = 0.$$

- If $\sigma(\cdot)$ is ReLU activation function, then

$$\mathbb{E}[x_2\sigma(x_1)\sigma'(x_1+x_2)] = \frac{b\sqrt{ab}}{2\pi(a+b)}.$$

- If $\sigma(\cdot)$ is Leaky ReLU activation function, then

$$\mathbb{E}[x_2\sigma(x_1)\sigma'(x_1+x_2)] = \frac{(1-\kappa)^2 b\sqrt{ab}}{2\pi(a+b)}.$$

Here, $\kappa$ is the coefficient of the Leaky ReLU activation function when the input is smaller than 0.

*Proof of Lemma F.4.* The first conclusion for the identity map is straightforward. When $\sigma(\cdot)$ is ReLU activation function, we can rewrite that $x_2\sigma(x_1)\sigma'(x_1+x_2) = x_1x_2\mathbb{1}_{\{x_1\geq 0\}}\mathbb{1}_{\{x_1+x_2\geq 0\}}$. Let $z_1 = \frac{x_1}{\sqrt{a}}$ and $z_2 = \frac{x_2}{\sqrt{b}}$, then we have,

$$\mathbb{E}[x_2\sigma(x_1)\sigma'(x_1+x_2)] = \sqrt{ab}\underbrace{\mathbb{E}[z_1z_2\mathbb{1}_{\{z_1\geq 0\}}\mathbb{1}_{\{\sqrt{a}z_1+\sqrt{b}z_2\geq 0\}}]}_{I}. \tag{F.8}$$

For $I$, by denoting $\lambda = \sqrt{\frac{a}{b}}$, it can be calculated by

$$\begin{aligned}
I &= \int_{0}^{\infty}\int_{-\lambda z_1}^{\infty} z_1z_2\phi(z_1)\phi(z_2)\mathrm{d}z_1\mathrm{d}z_2 = \int_{0}^{\infty} z_1\phi(z_1)\left( \int_{-\lambda z_1}^{\infty} z_2\phi(z_2)\mathrm{d}z_2 \right)\mathrm{d}z_1 \\
&= \int_{0}^{\infty} z_1\phi(z_1)\left( \frac{1}{\sqrt{2\pi}}\int_{-\lambda z_1}^{\infty} z_2 e^{-\frac{z_2^2}{2}}\mathrm{d}z_2 \right)\mathrm{d}z_1 = \int_{0}^{\infty} z_1\phi(z_1)\left( \frac{1}{\sqrt{2\pi}}\int_{\frac{\lambda^2 z_1^2}{2}}^{\infty} e^{-z_2}\mathrm{d}z_2 \right)\mathrm{d}z_1 \\
&= \frac{1}{2\pi}\int_{0}^{\infty} z_1 e^{-\frac{z_1^2(1+\lambda^2)}{2}}\mathrm{d}z_1 = \frac{1}{2\pi(1+\lambda^2)} = \frac{b}{2\pi(a+b)}. \tag{F.9}
\end{aligned}$$

Now applying the results of (F.9) into (F.8), we finish the proof when $\sigma(\cdot)$ is the ReLU activation function. In addition, when $\sigma(\cdot)$ is the Leaky ReLU activation function, we can derive that

$$\mathbb{E}[x_2\sigma(x_1)\sigma'(x_1+x_2)]$$

$$
\begin{aligned}
=&\sqrt{ab}\mathbb{E}[z_1 z_2 \mathbb{1}_{\{z_1 \geq 0\}} \mathbb{1}_{\{\sqrt{a}z_1 + \sqrt{b}z_2 \geq 0\}}] + \sqrt{ab}\kappa\mathbb{E}[z_1 z_2 \mathbb{1}_{\{z_1 < 0\}} \mathbb{1}_{\{\sqrt{a}z_1 + \sqrt{b}z_2 \geq 0\}}] \\
&+ \kappa\sqrt{ab}\mathbb{E}[z_1 z_2 \mathbb{1}_{\{z_1 \geq 0\}} \mathbb{1}_{\{\sqrt{a}z_1 + \sqrt{b}z_2 < 0\}}] + \kappa^2\sqrt{ab}\mathbb{E}[z_1 z_2 \mathbb{1}_{\{z_1 < 0\}} \mathbb{1}_{\{\sqrt{a}z_1 + \sqrt{b}z_2 < 0\}}] \\
=&(1 + \kappa^2)\sqrt{ab}\mathbb{E}[z_1 z_2 \mathbb{1}_{\{z_1 \geq 0\}} \mathbb{1}_{\{\sqrt{a}z_1 + \sqrt{b}z_2 \geq 0\}}] + 2\kappa\sqrt{ab}\mathbb{E}[z_1 z_2 \mathbb{1}_{\{z_1 < 0\}} \mathbb{1}_{\{\sqrt{a}z_1 + \sqrt{b}z_2 \geq 0\}}],
\end{aligned}
$$

where the last equality holds by the symmetry of $z_1$ and $z_2$. In addition, by a similar calculation process, we can obtain that

$$
\mathbb{E}[z_1 z_2 \mathbb{1}_{\{z_1 < 0\}} \mathbb{1}_{\{\sqrt{a}z_1 + \sqrt{b}z_2 \geq 0\}}] = \frac{1}{2\pi} \int_{-\infty}^{0} z_1 e^{-\frac{z_1^2(1+\lambda^2)}{2}} \, dz_1 = -\frac{1}{2\pi(1+\lambda^2)} = -\frac{b}{2\pi(a+b)}.
$$

Consequently, we can finally obtain that

$$
\mathbb{E}[x_2 \sigma(x_1)\sigma'(x_1 + x_2)] = \frac{(1 - 2\kappa + \kappa^2)b\sqrt{ab}}{2\pi(a+b)} = \frac{(1-\kappa)^2 b\sqrt{ab}}{2\pi(a+b)},
$$

which finishes the proof. $\qquad\square$

Then, based on the conclusions of Lemma F.3 and Lemma F.4, we can immediately obtain the following lemma as a corollary.

**Lemma F.5** (Calculation of $F_3(a,b)$ defined in (D.4)). Let $x_1 \sim \mathcal{N}(0,a)$, $x_2 \sim \mathcal{N}(0,b)$ be two independent Gaussian random variables, then it holds that

- If $\sigma(\cdot)$ is the identity map, then

$$
\mathbb{E}[(x_1 + x_2)\sigma(x_1)\sigma'(x_1 + x_2)] = a.
$$

- If $\sigma(\cdot)$ is ReLU activation function, then

$$
\mathbb{E}[(x_1 + x_2)\sigma(x_1)\sigma'(x_1 + x_2)] = \frac{a}{4} + \frac{a}{2\pi}\arctan\left(\sqrt{\frac{a}{b}}\right) + \frac{\sqrt{ab}}{2\pi}.
$$

And there exist the following matching lower and upper bounds:

$$
\frac{a}{2} \vee \left(\frac{a}{4} + \frac{\sqrt{ab}}{2\pi}\right) \leq \mathbb{E}[(x_1 + x_2)\sigma(x_1)\sigma'(x_1 + x_2)] \leq \frac{a}{2} + \frac{b\sqrt{ab}}{2\pi(a+b)} \leq \frac{a}{2} + \frac{b}{4\pi}.
$$

- If $\sigma(\cdot)$ is Leaky ReLU activation function, then

$$
\mathbb{E}[(x_1 + x_2)\sigma(x_1)\sigma'(x_1 + x_2)] = \frac{(1+\kappa)^2 a}{4} + \frac{(1-\kappa)^2 a}{2\pi}\arctan\left(\sqrt{\frac{a}{b}}\right) + \frac{(1-\kappa)^2\sqrt{ab}}{2\pi}.
$$

And there exist the following matching lower and upper bounds:

$$
\mathbb{E}[x_1 \sigma(x_1)\sigma'(x_1 + x_2)] \geq \frac{(1+\kappa)^2 a}{2} \vee \left(\frac{(1+\kappa)^2 a}{4} + \frac{(1-\kappa)^2\sqrt{ab}}{2\pi}\right);
$$

$$
\mathbb{E}[x_1 \sigma(x_1)\sigma'(x_1 + x_2)] \leq \frac{(1+\kappa^2)a}{2} + \frac{(1-\kappa)^2 b\sqrt{ab}}{2\pi(a+b)} \leq \frac{(1+\kappa^2)a}{2} + \frac{(1-\kappa)^2 b}{4\pi}.
$$

Here, $\kappa$ is the coefficient of the Leaky ReLU activation function when the input is smaller than 0.

**Lemma F.6** (Calculation of $F_4(a,b,c)$ defined in (D.5)). Let $x_1 \sim \mathcal{N}(0,a)$, $x_2 \sim \mathcal{N}(0,b)$, $x_3 \sim \mathcal{N}(0,c)$ be three independent Gaussian random variables, then it holds that

- If $\sigma(\cdot)$ is the identity map, then

$$
\mathbb{E}[x_1 \sigma(x_1 + x_2)\sigma'(x_1 + x_2 + x_3)] = a.
$$

- If $\sigma(\cdot)$ is ReLU activation function, then

$$
\mathbb{E}[x_1 \sigma(x_1 + x_2)\sigma'(x_1 + x_2 + x_3)] = \frac{a}{4} + \frac{a}{2\pi}\left(\arctan\left(\sqrt{\frac{a+b}{c}}\right) + \frac{\sqrt{(a+b)c}}{a+b+c}\right).
$$

And there exist the following matching lower and upper bounds:

$$
\left(\frac{a}{4} + \frac{a\sqrt{(a+b)c}}{2\pi(a+b+c)}\right) \vee \left(\frac{a}{2} - \frac{ac^{\frac{3}{2}}}{2\pi\sqrt{a+b}(a+b+c)}\right) \leq \mathbb{E}[x_1 \sigma(x_1 + x_2)\sigma'(x_1 + x_2 + x_3)] \leq \frac{a}{2}.
$$

- If $\sigma(\cdot)$ is Leaky ReLU activation function, then

$$\mathbb{E}[x_1\sigma(x_1+x_2)\sigma'(x_1+x_2+x_3)] = \frac{(1+\kappa)^2 a}{4} + \frac{(1-\kappa)^2 a}{2\pi}\left(\arctan\left(\sqrt{\frac{a+b}{c}}\right) + \frac{\sqrt{(a+b)c}}{a+b+c}\right).$$

And there exist the following matching lower and upper bounds:

$$\mathbb{E}[x_1\sigma(x_1+x_2)\sigma'(x_1+x_2+x_3)]$$
$$\geq \left(\frac{(1+\kappa)^2 a}{4} + \frac{(1-\kappa)^2 a\sqrt{(a+b)c}}{2\pi(a+b+c)}\right) \vee \left(\frac{(1+\kappa^2)a}{2} - \frac{(1-\kappa)^2 ac^{\frac{3}{2}}}{2\pi\sqrt{a+b}(a+b+c)}\right);$$
$$\mathbb{E}[x_1\sigma(x_1+x_2)\sigma'(x_1+x_2+x_3)] \leq \frac{(1+\kappa^2)a}{2}.$$

Here, $\kappa$ is the coefficient of the Leaky ReLU activation function when the input is smaller than 0.

*Proof of Lemma F.6.* The first conclusion for the identity map is straightforward. When $\sigma(\cdot)$ is ReLU activation function, we can rewrite that $x_1\sigma(x_1+x_2)\sigma'(x_1+x_2+x_3) = x_1(x_1+x_2)\mathbb{1}_{\{x_1+x_2\geq 0\}}\mathbb{1}_{\{x_1+x_2+x_3\geq 0\}}$. Additionally, let $x_4 = x_1+x_2 \sim \mathcal{N}(0, a+b)$ and $z = \frac{x_4}{\sqrt{a+b}} \sim \mathcal{N}(0,1)$. Then we have $\text{Cov}(x_1, x_4) = \mathbb{E}[(x_1+x_2)x_1] = a$, and $\mathbb{E}[x_1|x_4] = \frac{a}{a+b}x_4$. Therefore, we have

$$\mathbb{E}[x_1\sigma(x_1+x_2)\sigma'(x_1+x_2+x_3)]$$
$$=\mathbb{E}[x_1(x_1+x_2)\mathbb{1}_{\{x_1+x_2\geq 0\}}\mathbb{1}_{\{x_1+x_2+x_3\geq 0\}}] = \mathbb{E}\big[\mathbb{E}[x_1(x_1+x_2)\mathbb{1}_{\{x_1+x_2\geq 0\}}\mathbb{1}_{\{x_1+x_2+x_3\geq 0\}}|x_1, x_2]\big]$$
$$=\mathbb{E}\left[x_1(x_1+x_2)\mathbb{1}_{\{x_1+x_2\geq 0\}}\Phi\left(\frac{x_1+x_2}{\sqrt{c}}\right)\right] = \mathbb{E}\left[\mathbb{E}\left[x_1 x_4\mathbb{1}_{\{x_4\geq 0\}}\Phi\left(\frac{x_4}{\sqrt{c}}\right)\Big|x_4\right]\right]$$
$$=\frac{a}{a+b}\mathbb{E}\left[x_4^2\mathbb{1}_{\{x_4\geq 0\}}\Phi\left(\frac{x_4}{\sqrt{c}}\right)\right] = a\mathbb{E}[z^2\mathbb{1}_{\{z\geq 0\}}\Phi(\lambda z)] = a\underbrace{\int_0^\infty z^2\Phi(\lambda z)\phi(z)\mathrm{d}z}_{I},$$

where $\lambda = \sqrt{\frac{a+b}{c}}$. By the similar process in the proof of Lemma F.3, we can obtain that

$$I = \frac{1}{4} + \frac{1}{2\pi}\left(\arctan\lambda + \frac{\lambda}{1+\lambda^2}\right) = \frac{1}{4} + \frac{1}{2\pi}\left(\arctan\left(\sqrt{\frac{a+b}{c}}\right) + \frac{\sqrt{(a+b)c}}{a+b+c}\right)$$

and

$$\left(\frac{1}{4} + \frac{\sqrt{(a+b)c}}{2\pi(a+b+c)}\right) \vee \left(\frac{1}{2} - \frac{c^{\frac{3}{2}}}{2\pi\sqrt{a+b}(a+b+c)}\right) \leq I \leq \frac{1}{2}.$$

Plugging these results into the previous equation of expectation, we finish the proof when $\sigma(\cdot)$ is the ReLU activation function. In addition, when $\sigma(\cdot)$ is the Leaky ReLU activation function, we have

$$\mathbb{E}[x_1\sigma(x_1+x_2)\sigma'(x_1+x_2+x_3)]$$
$$=\mathbb{E}[x_1(x_1+x_2)\mathbb{1}_{\{x_1+x_2\geq 0\}}\mathbb{1}_{\{x_1+x_2+x_3\geq 0\}}] + \kappa\mathbb{E}[x_1(x_1+x_2)\mathbb{1}_{\{x_1+x_2<0\}}\mathbb{1}_{\{x_1+x_2+x_3\geq 0\}}]$$
$$\quad + \kappa\mathbb{E}[x_1(x_1+x_2)\mathbb{1}_{\{x_1+x_2\geq 0\}}\mathbb{1}_{\{x_1+x_2+x_3<0\}}] + \kappa^2\mathbb{E}[x_1(x_1+x_2)\mathbb{1}_{\{x_1+x_2<0\}}\mathbb{1}_{\{x_1+x_2+x_3<0\}}]$$
$$=(1+\kappa^2)\mathbb{E}[x_1(x_1+x_2)\mathbb{1}_{\{x_1+x_2\geq 0\}}\mathbb{1}_{\{x_1+x_2+x_3\geq 0\}}] + 2\kappa\mathbb{E}[x_1(x_1+x_2)\mathbb{1}_{\{x_1+x_2<0\}}\mathbb{1}_{\{x_1+x_2+x_3\geq 0\}}].$$

By utilizing a similar calculation process, we have

$$\mathbb{E}[x_1(x_1+x_2)\mathbb{1}_{\{x_1+x_2<0\}}\mathbb{1}_{\{x_1+x_2+x_3\geq 0\}}] = a\int_{-\infty}^0 z^2\Phi(\lambda z)\phi(z)\mathrm{d}z$$
$$= \frac{a}{4} - \frac{a}{2\pi}\left(\arctan\left(\sqrt{\frac{a+b}{c}}\right) + \frac{\sqrt{(a+b)c}}{a+b+c}\right).$$

Plugging this result into the previous calculations, we finish the proof. And the upper and lower bounds for Leaky ReLU activation function can be directly derived by comparing the formulas. $\square$

**Lemma F.7** (Calculation of $F_5(a, b, c)$ defined in (D.6)). Let $x_1 \sim \mathcal{N}(0, a)$, $x_2 \sim \mathcal{N}(0, b)$, $x_3 \sim \mathcal{N}(0, c)$ be three independent Gaussian random variables, then it holds that

- If $\sigma(\cdot)$ is the identity map, then

$$\mathbb{E}[x_2 \sigma(x_1) \sigma'(x_1 + x_2 + x_3)] = 0.$$

- If $\sigma(\cdot)$ is ReLU activation function, then

$$\mathbb{E}[x_2 \sigma(x_1) \sigma'(x_1 + x_2 + x_3)] = \frac{b\sqrt{a(b+c)}}{2\pi(a+b+c)}.$$

- If $\sigma(\cdot)$ is Leaky ReLU activation function, then

$$\mathbb{E}[x_2 \sigma(x_1) \sigma'(x_1 + x_2 + x_3)] = \frac{(1-\kappa)^2 b\sqrt{a(b+c)}}{2\pi(a+b+c)}.$$

Here, $\kappa$ is the coefficient of the Leaky ReLU activation function when the input is smaller than 0.

*Proof of Lemma F.7.* The first conclusion for the identity map is straightforward. When $\sigma(\cdot)$ is ReLU activation function, we can rewrite that $x_2 \sigma(x_1) \sigma'(x_1 + x_2 + x_3) = x_1 x_2 \mathbb{1}_{\{x_1 \geq 0\}} \mathbb{1}_{\{x_1 + x_2 + x_3 \geq 0\}}$. Then we have

$$
\begin{aligned}
\mathbb{E}[x_2 \sigma(x_1) \sigma'(x_1 + x_2 + x_3)] &= \mathbb{E}[x_1 x_2 \mathbb{1}_{\{x_1 \geq 0\}} \mathbb{1}_{\{x_1 + x_2 + x_3 \geq 0\}}] \\
&= \mathbb{E}\big[x_1 x_2 \mathbb{1}_{\{x_1 \geq 0\}} \mathbb{1}_{\{x_1 + x_2 + x_3 \geq 0\}} | x_1, x_2\big] \\
&= \mathbb{E}\left[x_1 x_2 \mathbb{1}_{\{x_1 \geq 0\}} \Phi\left(\frac{x_1 + x_2}{\sqrt{c}}\right)\right] \\
&= \int_0^\infty \int_{-\infty}^\infty x_1 x_2 \Phi\left(\frac{x_1 + x_2}{\sqrt{c}}\right) \phi(x_1) \phi(x_2) \mathrm{d}x_1 \mathrm{d}x_2 \\
&= \int_0^\infty x_1 \frac{1}{\sqrt{2\pi a}} e^{-\frac{x_1^2}{2a}} \underbrace{\left(\int_{-\infty}^\infty x_2 \Phi\left(\frac{x_1 + x_2}{\sqrt{c}}\right) \frac{1}{\sqrt{2\pi b}} e^{-\frac{x_2^2}{2b}} \mathrm{d}x_2\right)}_{I} \mathrm{d}x_1
\end{aligned}
$$

We can utilize the integral by parts to derive that

$$
\begin{aligned}
I &= -\sqrt{\frac{b}{2\pi}} \int_{-\infty}^\infty \Phi\left(\frac{x_1 + x_2}{\sqrt{c}}\right) \mathrm{d}e^{-\frac{x_2^2}{2b}} - \sqrt{\frac{b}{2\pi}} \Phi\left(\frac{x_1 + x_2}{\sqrt{c}}\right) e^{-\frac{x_2^2}{2b}} \Big|_{-\infty}^\infty + \sqrt{\frac{b}{2\pi}} \int_{-\infty}^\infty e^{-\frac{x_2^2}{2b}} \mathrm{d}\Phi\left(\frac{x_1 + x_2}{\sqrt{c}}\right) \\
&= \frac{1}{2\pi} \sqrt{\frac{b}{c}} \int_{-\infty}^\infty e^{-\frac{x_2^2}{2b} - \frac{(x_1 + x_2)^2}{2c}} \mathrm{d}x_2 = \frac{1}{2\pi} \sqrt{\frac{b}{c}} \int_{-\infty}^\infty e^{-\frac{(x_2 + \frac{b}{b+c} x_1)^2}{2\frac{bc}{b+c}} - \frac{x_1^2}{2(b+c)}} \mathrm{d}x_2 = \frac{b}{\sqrt{2\pi(b+c)}} e^{-\frac{x_1^2}{2(b+c)}}
\end{aligned}
$$

Now substitute this result of $I$ back into the outer integral for the calculation for expectation, then we have

$$
\begin{aligned}
\mathbb{E}[x_2 \sigma(x_1) \sigma'(x_1 + x_2 + x_3)] &= \frac{b}{2\pi\sqrt{a(b+c)}} \int_0^\infty x_1 e^{-\frac{x_1^2}{2a} - \frac{x_1^2}{2(b+c)}} \mathrm{d}x_1 \\
&= \frac{b}{2\pi\sqrt{a(b+c)}} \frac{a(b+c)}{a+b+c} \int_0^\infty e^{-\frac{(a+b+c)x_1^2}{2a(b+c)}} \mathrm{d}\frac{(a+b+c)x_1^2}{2a(b+c)} \\
&= \frac{b\sqrt{a(b+c)}}{2\pi(a+b+c)}.
\end{aligned}
$$

This finish the proof when $\sigma(\cdot)$ is ReLU activation function. In addition, when $\sigma(\cdot)$ is Leaky ReLU activation function, we can derive that

$$
\begin{aligned}
&\mathbb{E}[x_2 \sigma(x_1) \sigma'(x_1 + x_2 + x_3)] \\
=&\mathbb{E}[x_1 x_2 \mathbb{1}_{\{x_1 \geq 0\}} \mathbb{1}_{\{x_1 + x_2 + x_3 \geq 0\}}] + \kappa \mathbb{E}[x_1 x_2 \mathbb{1}_{\{x_1 < 0\}} \mathbb{1}_{\{x_1 + x_2 + x_3 \geq 0\}}] \\
&+ \kappa \mathbb{E}[x_1 x_2 \mathbb{1}_{\{x_1 \geq 0\}} \mathbb{1}_{\{x_1 + x_2 + x_3 < 0\}}] + \kappa^2 \mathbb{E}[x_1 x_2 \mathbb{1}_{\{x_1 < 0\}} \mathbb{1}_{\{x_1 + x_2 + x_3 < 0\}}]
\end{aligned}
$$

$$=(1 + \kappa^2)\mathbb{E}[x_1 x_2 \mathbb{1}_{\{x_1 \geq 0\}} \mathbb{1}_{\{x_1 + x_2 + x_3 \geq 0\}}] + 2\kappa\mathbb{E}[x_1 x_2 \mathbb{1}_{\{x_1 < 0\}} \mathbb{1}_{\{x_1 + x_2 + x_3 \geq 0\}}].$$

By applying a similar calculation process, we can derive that

$$\mathbb{E}[x_1 x_2 \mathbb{1}_{\{x_1 < 0\}} \mathbb{1}_{\{x_1 + x_2 + x_3 \geq 0\}}] = \frac{b}{2\pi\sqrt{a(b+c)}} \int_{-\infty}^{0} x_1 e^{-\frac{x_1^2}{2a} - \frac{x_1^2}{2(b+c)}} \, dx_1 = -\frac{b\sqrt{a(b+c)}}{2\pi(a+b+c)}.$$

Applying this result, we finish the proof. □

**Lemma F.8.** Let $x_1 \sim \mathcal{N}(0, a)$, $x_2 \sim \mathcal{N}(0, b)$ be two independent Gaussian random variables, then it holds that

- If $\sigma(\cdot)$ is the identity map, then

$$\mathbb{E}[\sigma(x_1)\sigma(x_1 + x_2)] = a.$$

- If $\sigma(\cdot)$ is ReLU activation function, then

$$\mathbb{E}[\sigma(x_1)\sigma(x_1 + x_2)] = \frac{a}{4} + \frac{a}{2\pi} \arctan\left(\sqrt{\frac{a}{b}}\right) + \frac{\sqrt{ab}}{2\pi}$$

- If $\sigma(\cdot)$ is Leaky ReLU activation function, then

$$\mathbb{E}[\sigma(x_1)\sigma(x_1 + x_2)] = \frac{(1+\kappa)^2 a}{4} + \frac{(1-\kappa)^2 a}{2\pi} \arctan\left(\sqrt{\frac{a}{b}}\right) + \frac{(1-\kappa)^2 \sqrt{ab}}{2\pi}. \tag{F.10}$$

Here, $\kappa$ is the coefficient of the Leaky ReLU activation function when the input is smaller than 0.

*Proof of Lemma F.8.* The first conclusion for the identity map is straightforward. When $\sigma(\cdot)$ is ReLU activation function or leaky ReLU activation function, we can utilize the fact that $x\sigma'(x) = \sigma(x)$ to re-write that

$$\mathbb{E}[\sigma(x_1)\sigma(x_1 + x_2)] = \mathbb{E}[(x_1 + x_2)\sigma(x_1)\sigma'(x_1 + x_2)] = \mathbb{E}[(x_1 + x_2)\sigma(x_1)\sigma'(x_1 + x_2)].$$

And this term has already been calculated in Lemma F.5. Hence we finish the proof. □

### F.2 ARITHMETIC INEQUALITIES

**Lemma F.9.** Let $a, b, c$ be three positive scalars, it holds that

$$\frac{c}{a+b} \geq \frac{c}{a} - \frac{bc}{a^2}$$

*Proof of Lemma F.9.*

$$\frac{c}{a+b} - \frac{c}{a} = -\frac{bc}{(a+b)a} \geq -\frac{bc}{a^2}.$$

This completes the proof. □

**Lemma F.10.** Let $a, b, c$ be three positive scalars, it holds that

$$\frac{c}{a-b} \leq \frac{c}{a} + \frac{bc}{(a-b)^2}$$

*Proof of Lemma F.10.*

$$\frac{c}{a-b} - \frac{c}{a} = \frac{bc}{(a-b)a} \leq \frac{bc}{(a-b)^2}.$$

This completes the proof. □

### F.3 SEQUENCE ITERATION BOUND

The following lemmas characterize the increase of a positive sequence with matching lower and upper bounds. Similar conclusions and proofs can be found in Jelassi et al. (2022); Cao et al. (2023); Meng et al. (2024); Zhang et al. (2024a; 2025c). We include the proof here for completeness.

**Lemma F.11.** Consider a positive sequence $\{x_t\}_{t=0}^{\infty}$ satisfying the following iterative rules:

$$x_{t+1} \geq x_t + \eta \cdot c_1 \cdot x_t^q;$$
$$x_{t+1} \leq x_t + \eta \cdot c_2 \cdot x_t^q,$$

where $c_2 \geq c_1 > 0$ are positive constants. For any $v > x_0$, let $T_v$ denote the first index $t$ such that $x_t \geq v$. Then, for any constant $\zeta > 0$, the following bounds on $T_v$ hold:

$$T_v \leq \frac{1+\zeta}{\eta c_1 x_0^{q-1}} + \frac{(1+\zeta)^q c_2 \log(\frac{v}{x_0})}{c_1}, \tag{F.11}$$

and

$$T_v \geq \frac{1}{(1+\zeta)^q \eta c_2 x_0^{q-1}} - \frac{\log(\frac{v}{x_0})}{(1+\zeta)^{q-1}}. \tag{F.12}$$

*Proof of Lemma F.11.* To prove the bounds, let $\mathcal{T}_g$ be the first iteration such that $x_t \geq (1+\zeta)^g x_0$. Furthermore, define $g^*$ as the smallest integer satisfying $(1+\zeta)^{g^*} x_0 \geq v$. This implies

$$\frac{\log(\frac{v}{x_0})}{\log(1+\zeta)} \leq g^* < \frac{\log(\frac{v}{x_0})}{\log(1+\zeta)} + 1.$$

For $t = \mathcal{T}_1$, we use the lower bound iteration:

$$x_{\mathcal{T}_1} \geq x_0 + \sum_{t=0}^{\mathcal{T}_1 - 1} \eta c_1 x_t^q \geq x_0 + \mathcal{T}_1 \eta c_1 x_0^q,$$

from which we can deduce that

$$\mathcal{T}_1 \leq \frac{x_{\mathcal{T}_1} - x_0}{\eta c_1 x_0^q}. \tag{F.13}$$

Utilizing the upper-bound iteration for $x_{\mathcal{T}_1}$ and the condition $x_{\mathcal{T}_1 - 1} \leq x_0(1+\zeta)$, we get

$$x_{\mathcal{T}_1} \leq x_{\mathcal{T}_1 - 1} + \eta c_2 x_{\mathcal{T}_1 - 1}^q \leq x_0(1+\zeta) + \eta c_2 x_0^q (1+\zeta)^q. \tag{F.14}$$

Combining the results from (F.13) and (F.14) leads to

$$\mathcal{T}_1 \leq \frac{\zeta}{\eta c_1 x_0^{q-1}} + \frac{(1+\zeta)^{q-1} c_2}{c_1}.$$

The case for $g > 1$ is handled similarly. Using the lower bound iteration from $\mathcal{T}_{g-1}$ to $\mathcal{T}_g - 1$:

$$x_{\mathcal{T}_g} \geq x_{\mathcal{T}_{g-1}} + \sum_{t=\mathcal{T}_{g-1}}^{\mathcal{T}_g - 1} \eta c_1 x_t^q \geq x_{\mathcal{T}_{g-1}} + \eta c_1 (\mathcal{T}_g - \mathcal{T}_{g-1}) x_0^q (1+\zeta)^{q(g-1)}, \tag{F.15}$$

and the difference $x_{\mathcal{T}_g} - x_{\mathcal{T}_{g-1}}$ can be upper bounded using the upper bound iteration and $x_{\mathcal{T}_{g-1}} \leq x_0(1+\zeta)^g$ and $x_{\mathcal{T}_{g-1}} \geq x_0(1+\zeta)^{g-1}$:

$$x_{\mathcal{T}_g} - x_{\mathcal{T}_{g-1}} \leq x_{\mathcal{T}_{g-1}} + \eta c_2 x_{\mathcal{T}_g}^q - x_{\mathcal{T}_{g-1}} \leq \zeta(1+\zeta)^{g-1} x_0 + \eta c_2 x_0^q (1+\zeta)^{gq}. \tag{F.16}$$

Combining (F.15) and (F.16), we derive that

$$\mathcal{T}_g \leq \mathcal{T}_{g-1} + \frac{\zeta}{\eta c_1 x_0^{q-1} (1+\zeta)^{(g-1)(q-1)}} + \frac{(1+\zeta)^q c_2}{c_1}. \tag{F.17}$$

Taking a telescoping sum of the results of (F.17) from $g = 1$ to $g = g^*$ and by the fact that $T_v \le \mathcal{T}_{g^*}$, we finally get (F.11). For the lower bound, we proceed similarly starting with $t = \mathcal{T}_1$. We use the upper bound iteration:

$$x_{\mathcal{T}_1} \le x_0 + \sum_{t=0}^{\mathcal{T}_1 - 1} \eta c_2 x_t^q \le x_0 + \mathcal{T}_1 \eta c_2 x_0^q (1 + \zeta)^q.$$

Substitute that $x_{\mathcal{T}_1} - x_0 \ge \zeta x_0$, we get

$$\mathcal{T}_1 \ge \frac{\zeta}{\eta c_2 x_0^{q-1} (1 + \zeta)^q}. \tag{F.18}$$

A similar derivation for $g > 1$ using the upper bound iteration gives:

$$x_{\mathcal{T}_g} \le x_{\mathcal{T}_{g-1}} + \sum_{t=\mathcal{T}_{g-1}}^{\mathcal{T}_g - 1} \eta c_2 x_t^q \le x_{\mathcal{T}_{g-1}} + \eta c_2 (\mathcal{T}_g - \mathcal{T}_{g-1}) x_0^q (1 + \zeta)^{gq}. \tag{F.19}$$

The difference $x_{\mathcal{T}_g} - x_{\mathcal{T}_{g-1}}$ can also be lower bounded by utilizing the fact that $x_{\mathcal{T}_{g-1}-1} \le x_0(1 + \zeta)^{g-1}$:

$$x_{\mathcal{T}_g} - x_{\mathcal{T}_{g-1}} \ge x_{\mathcal{T}_g} - x_{\mathcal{T}_{g-1}-1} - \eta c_2 x_{\mathcal{T}_{g-1}-1}^{q-1} \ge \zeta (1 + \zeta)^{g-1} x_0 - \eta c_2 x_0^q (1 + \zeta)^{(g-1)q}. \tag{F.20}$$

Combining the results from (F.19) and (F.20), we obtain that,

$$\mathcal{T}_g \ge \mathcal{T}_{g-1} + \frac{\zeta}{\eta c_2 x_0^{q-1} (1 + \zeta)^{g(q-1)+1}} - \frac{1}{(1 + \zeta)^q}. \tag{F.21}$$

Taking a telescoping sum of the results of (F.21) from $g = 1$ to $g = g^* - 1$ and by the fact that $T_v \ge \mathcal{T}_{g^*-1}$, we finally get (F.12). $\qquad\square$

**Lemma F.12.** Let $x_t$ be a positive sequence for $t \ge 0$. Assume $x_t$ satisfies the iterative formula

$$x_{t+1} = x_t + c_1 e^{-c_2 x_t}$$

for given constants $c_1, c_2 > 0$. Then, for all $t \ge 0$, the sequence $x_t$ is bounded as follows:

$$\frac{1}{c_2} \log(c_1 c_2 t + e^{c_2 x_0}) \le x_t \le c_1 e^{-c_2 x_0} + \frac{1}{c_2} \log(c_1 c_2 t + e^{c_2 x_0}).$$

*Proof of Lemma F.12.* First, we establish the lower bound for $x_t$. We introduce a continuous-time sequence $\underline{x}_t$, $t \ge 0$ defined by the integral equation with the same initial value.

$$\underline{x}_t = \underline{x}_0 + c_1 \cdot \int_0^t e^{-c_2 \underline{x}_\tau} \mathrm{d}\tau, \quad \underline{x}_0 = x_0. \tag{F.22}$$

Observe that $\underline{x}_t$ is clearly an increasing function of $t$. Hence, we obtain

$$\underline{x}_{t+1} = \underline{x}_t + c_1 \cdot \int_t^{t+1} e^{-c_2 \underline{x}_\tau} \mathrm{d}\tau$$

$$\le \underline{x}_t + c_1 \cdot \int_t^{t+1} e^{-c_2 \underline{x}_t} \mathrm{d}\tau$$

$$= \underline{x}_t + c_1 \exp(-c_2 \underline{x}_t)$$

for all $t \in \mathbb{N}$. By comparing the preceding inequality with the iterative formula for $\{x_t\}$, the comparison theorem implies that $x_t \ge \underline{x}_t$ for all $t \in \mathbb{N}$. Equation (F.22) possesses an exact solution given by

$$\underline{x}_t = \frac{1}{c_2} \log(c_1 c_2 t + e^{c_2 x_0}).$$

Thus, we have

$$x_t \ge \frac{1}{c_2} \log(c_1 c_2 t + e^{c_2 x_0})$$

for all $t \in \mathbb{N}$. This concludes the derivation of the lower bound.

Next, we derive the upper bound for $x_t$. We have

$$
\begin{aligned}
x_t &= x_0 + c_1 \cdot \sum_{\tau=0}^{t-1} e^{-c_2 x_\tau} \\
&\leq x_0 + c_1 \cdot \sum_{\tau=0}^{t} e^{-\log(c_1 c_2 \tau + e^{c_2 x_0})} \\
&= x_0 + c_1 \cdot \sum_{\tau=0}^{t} \frac{1}{c_1 c_2 \tau + e^{c_2 x_0}} \\
&= x_0 + \frac{c_1}{e^{c_2 x_0}} + c_1 \cdot \sum_{\tau=1}^{t} \frac{1}{c_1 c_2 \tau + e^{c_2 x_0}} \\
&\leq x_0 + \frac{c_1}{e^{c_2 x_0}} + c_1 \cdot \int_0^t \frac{1}{c_1 c_2 \tau + e^{c_2 x_0}} \mathrm{d}\tau,
\end{aligned}
$$

where the second inequality utilizes the lower bound for $x_t$ derived in the first part of the lemma's result. Consequently, we obtain

$$
\begin{aligned}
x_t &\leq x_0 + \frac{c_1}{e^{c_2 x_0}} + \frac{1}{c_2} \log(c_1 c_2 t + e^{c_2 x_0}) - \frac{1}{c_2} \log(e^{c_2 x_0}) \\
&= c_1 e^{-c_2 x_0} + \frac{1}{c_2} \log(c_1 c_2 t + e^{c_2 x_0}).
\end{aligned}
$$

This completes the proof. $\qquad \square$

## G  PROOF OF THE CASE WHEN $D = K$

In this section, we provide the theoretical results for the special case $D = K$. Under this setting, the ground-truth softmax scores reduce to a trivial rank-one structure that $\mathbf{S}^* = \frac{1}{D} \mathbf{1}_D \mathbf{1}_D^\top$. Consequently, the initialization $\mathbf{W}_{KQ}^{(0)} = \mathbf{0}_{D \times D}$ already yields $\mathbf{S}^{(0)} = \frac{1}{D} \mathbf{1}_D \mathbf{1}_D^\top$, achieving an exact recovery of $\mathbf{S}^*$ at the start of training. As a result, the gradient with respect to $\mathbf{W}_{KQ}$ remains zero throughout the optimization, and the problem effectively reduces to optimizing the single parameter matrix $\mathbf{W}_V$. Under this reduced setting, the loss becomes strongly convex in $\mathbf{W}_V$, and gradient descent enjoys a linear convergence rate, which is much faster than the $\Theta(1/T)$ rate established in Theorem 3.1. Since Theorem 3.1 provides matching upper and lower bounds and is therefore tight and can not be improved, this linear convergence phenomenon is exclusive to the degenerate case $D = K$. This explains why the proof strategy for Theorem 3.1 does not extend to the $D = K$ setting.

Now, we present the following Theorem G.1 to characterize the loss convergence when a one-layer transformer is supervised by a teacher model $f^*(\cdot)$ with $\mathbf{S}^* = \frac{1}{D} \mathbf{1}_D \mathbf{1}_D^\top$.

**Theorem G.1.** Suppose that $\eta \leq \frac{1}{2}$, then for any $t > 0$, the excess loss is minimized as

$$
\mathcal{L}(\mathbf{W}_V^{(t)}, \mathbf{W}_{KQ}^{(t)}) - \mathcal{L}_{\mathrm{opt}} \leq \frac{\sum_{m=1}^{M} \|\mathbf{v}_m\|_2^2}{2} e^{-\eta(t-1)}.
$$

Before we provide the proof for Theorem G.1, we first provide and prove the following lemma.

**Lemma G.2.** Under the same conditions of Theorem 3.1, there exist a time dependent non-negative scalar $C(t)$, such that

$$
\mathbf{w}_{V,m}^{(t)} = C(t) \cdot \mathbf{v}_m^*, \text{ for all } m \in [M]; \tag{G.1}
$$

and $C(t)$ has the following closed formulation:

$$
C(t) = \left(1 - \eta F_1(1)\right)^{t-1},
$$

where the function $F_1(\cdot)$ is defined in (D.2). In addition, $\mathbf{W}_{KQ}^{(t)}$ remains zero throughout the training.

*Proof of Lemma G.2.* W.L.O.G., we assume that $\mathbf{v}_m^*$ is already normalized, and $\boldsymbol{\Gamma}_m = [\mathbf{v}_m^*, \boldsymbol{\xi}_{m,2}, \ldots, \boldsymbol{\xi}_{m,d}] \in \mathbb{R}^{d \times d}$ be an orthogonal matrix with $\mathbf{v}_m$ being its first column. We prove this lemma by induction. Since these two conclusions holds at initialization with $C(0) = 0$. It is sufficient to prove that $\nabla_{\mathbf{w}_{V,m}} \mathcal{L}(\mathbf{W}_V^{(t)}; \mathbf{W}_{KQ}^{(t)}) = c(t) \cdot \mathbf{v}_m^*$ and $\nabla_{\mathbf{W}_{KQ}} \mathcal{L}(\mathbf{W}_V^{(t)}; \mathbf{W}_{KQ}^{(t)}) = \mathbf{0}$, when assuming $\mathbf{w}_{V,m}^{(t)} = C(t) \cdot \mathbf{v}_m^*$ and $\mathbf{W}_{KQ}^{(t)} = \mathbf{0}$. Notice that $\mathbf{W}_{KQ}^{(t)} = \mathbf{0}$ implies that $\mathbf{S}_{i',i}^{(t)} = 1/D$ for all $i', i \in [D]$. By the gradient calculations demonstrated in Lemma D.1, we have

$$
\begin{aligned}
\nabla_{\mathbf{w}_{V,m}} \mathcal{L}(\mathbf{W}_V^{(t)}; \mathbf{W}_{KQ}^{(t)}) = & -\sum_{i=1}^{D} \sum_{i_1=1}^{D} \mathbb{E}\left[\left[\mathbf{Y}_{m,i} - \sigma\left(\sum_{i_1=1}^{D} \langle \mathbf{w}_{V,m}^{(t)}, \mathbf{x}_{i_1} \rangle \mathbf{S}_{i_1,i}^{(t)}\right)\right] \right. \\
& \left. \cdot \sigma'\left(\sum_{i_1=1}^{D} \langle \mathbf{w}_{V,m}^{(t)}, \mathbf{x}_{i_1} \rangle \mathbf{S}_{i_1,i}^{(t)}\right) \mathbf{x}_{i_1} \mathbf{S}_{i_1,i}^{(t)}\right] \\
= & -\underbrace{\sum_{i=1}^{D} \sum_{i_1=1}^{D} \mathbb{E}\left[\mathbf{Y}_{m,i} \sigma'\left(\sum_{i_1=1}^{D} \frac{\langle \mathbf{w}_{V,m}^{(t)}, \mathbf{x}_{i_1} \rangle}{D}\right) \frac{\mathbf{x}_{i_1}}{D}\right]}_{I_1} \\
& +\underbrace{\sum_{i_1=1}^{D} \mathbb{E}\left[\sigma\left(\sum_{i_1=1}^{D} \frac{\langle \mathbf{w}_{V,m}^{(t)}, \mathbf{x}_{i_1} \rangle}{D}\right) \sigma'\left(\sum_{i_1=1}^{D} \frac{\langle \mathbf{w}_{V,m}^{(t)}, \mathbf{x}_{i_1} \rangle}{D}\right) \mathbf{x}_{i_1}\right]}_{I_2} \quad \text{(G.2)}
\end{aligned}
$$

For $I_1$, we have

$$
\begin{aligned}
I_1 = & \sum_{i=1}^{D} \sum_{i_1=1}^{D} \mathbb{E}\left[\mathbf{Y}_{m,i} \sigma'\left(\sum_{i_1=1}^{D} \frac{\langle \mathbf{w}_{V,m}^{(t)}, \mathbf{x}_{i_1} \rangle}{D}\right) \frac{\boldsymbol{\Gamma}_m \boldsymbol{\Gamma}_m^\top \mathbf{x}_{i_1}}{D}\right] \\
= & \sum_{i=1}^{D} \sum_{i_1=1}^{D} \mathbb{E}\left[\left[f^*(\mathbf{X})\right]_{m,i} \sigma'\left(\sum_{i_1=1}^{D} \frac{\langle \mathbf{w}_{V,m}^{(t)}, \mathbf{x}_{i_1} \rangle}{D}\right) \frac{\langle \mathbf{v}_m^*, \mathbf{x}_{i_1} \rangle}{D}\right] \cdot \mathbf{v}_m^* \\
& +\sum_{i=1}^{D} \sum_{i_1=1}^{D} \sum_{k=2}^{d} \mathbb{E}\left[\left[f^*(\mathbf{X})\right]_{m,i} \sigma'\left(\sum_{i_1=1}^{D} \frac{\langle \mathbf{w}_{V,m}^{(t)}, \mathbf{x}_{i_1} \rangle}{D}\right) \frac{\langle \boldsymbol{\xi}_{m,k}, \mathbf{x}_{i_1} \rangle}{D}\right] \cdot \boldsymbol{\xi}_{m,k} \\
= & \sum_{i=1}^{D} \sum_{i_1=1}^{D} \mathbb{E}\left[\left[f^*(\mathbf{X})\right]_{m,i} \sigma'\left(\sum_{i_1=1}^{D} \frac{\langle \mathbf{w}_{V,m}^{(t)}, \mathbf{x}_{i_1} \rangle}{D}\right) \frac{\langle \mathbf{v}_m^*, \mathbf{x}_{i_1} \rangle}{D}\right] \cdot \mathbf{v}_m^*.
\end{aligned}
$$

The first quality holds as $\mathcal{E}$ is mean-zero and independent with $\mathbf{X}$, and the last equality holds as the orthogonality between $\mathbf{v}_m^*$ and $\boldsymbol{\xi}_{m,k}$ implies that $\langle \mathbf{v}_m^*, \mathbf{x}_{i_2} \rangle$ is independent with $\langle \boldsymbol{\xi}_{m,k}, \mathbf{x}_{i_1} \rangle$ for all $i_1, i_2 \in [D]$. Notice that $\left[f^*(\mathbf{X})\right]_{m,i} = \frac{1}{D} \sigma\left(\sum_{i_1=1}^{D} \langle \mathbf{v}_m^*, \mathbf{x}_{i_1} \rangle\right)$ and $\sigma'\left(\sum_{i_1=1}^{D} \frac{\langle \mathbf{w}_{V,m}^{(t)}, \mathbf{x}_{i_1} \rangle}{D}\right) = \sigma'\left(\frac{C(t)}{D} \sum_{i_1=1}^{D} \langle \mathbf{v}_m^*, \mathbf{x}_{i_1} \rangle\right) = \sigma'\left(\sum_{i_1=1}^{D} \langle \mathbf{v}_m^*, \mathbf{x}_{i_1} \rangle\right)$. Consequently, $\langle \boldsymbol{\xi}_{m,k}, \mathbf{x}_{i_1} \rangle$ is a mean-zero Gaussian random variable, and independent with both $\left[f^*(\mathbf{X})\right]_{m,i}$ and $\sigma'\left(\sum_{i_1=1}^{D} \frac{\langle \mathbf{w}_{V,m}^{(t)}, \mathbf{x}_{i_1} \rangle}{D}\right)$ simultaneously, implying that

$$
\begin{aligned}
& \mathbb{E}\left[\left[f^*(\mathbf{X})\right]_{m,i} \sigma'\left(\sum_{i_1=1}^{D} \frac{\langle \mathbf{w}_{V,m}^{(t)}, \mathbf{x}_{i_1} \rangle}{D}\right) \langle \boldsymbol{\xi}_{m,k}, \mathbf{x}_{i_1} \rangle\right] \\
= & \mathbb{E}\left[\left[f^*(\mathbf{X})\right]_{m,i} \sigma'\left(\sum_{i_1=1}^{D} \frac{\langle \mathbf{w}_{V,m}^{(t)}, \mathbf{x}_{i_1} \rangle}{D}\right)\right] \mathbb{E}[\langle \boldsymbol{\xi}_{m,k}, \mathbf{x}_{i_1} \rangle] = 0.
\end{aligned}
$$

Based on previous results, by plugging $\left[f^*(\mathbf{X})\right]_{m,i} = \frac{1}{D} \sigma\left(\sum_{i_1=1}^{D} \langle \mathbf{v}_m^*, \mathbf{x}_{i_1} \rangle\right)$ and utilizing the definition of $F_1(a)$ in (D.2), we can further derive that

$$
I_1 = \sum_{i=1}^{D} \sum_{i_1=1}^{D} \mathbb{E}\left[\frac{1}{D} \sigma\left(\sum_{i_1=1}^{D} \langle \mathbf{v}_m^*, \mathbf{x}_{i_1} \rangle\right) \sigma'\left(\sum_{i_1=1}^{D} \frac{\langle \mathbf{w}_{V,m}^{(t)}, \mathbf{x}_{i_1} \rangle}{D}\right) \frac{\langle \mathbf{v}_m^*, \mathbf{x}_{i_1} \rangle}{D}\right] \cdot \mathbf{v}_m^*
$$

$$= \frac{1}{D}\mathbb{E}\left[\sigma\left(\sum_{i_1=1}^{D}\langle \mathbf{v}_m^*, \mathbf{x}_{i_1}\rangle\right)\sigma'\left(\sum_{i_1=1}^{D}\langle \mathbf{v}_m^*, \mathbf{x}_{i_1}\rangle\right)\sum_{i_1=1}^{D}\langle \mathbf{v}_m^*, \mathbf{x}_{i_1}\rangle\right]\cdot \mathbf{v}_m^* = F_1(1)\cdot \mathbf{v}_m^*.$$

The second equality is derived by fact that $\sigma(ax) = a\sigma(x)$ and $\sigma'(ax) = \sigma'(x)$ if $a \geq 0$. Then we can conclude the final result by the definition of $F_1(a)$ in (D.2). Similar to the process of handling $I_1$, we have the following for $I_2$:

$$I_2 = \sum_{i_1=1}^{D}\mathbb{E}\left[\sigma\left(\sum_{i_1=1}^{D}\frac{\langle \mathbf{w}_{V,m}^{(t)}, \mathbf{x}_{i_1}\rangle}{D}\right)\sigma'\left(\sum_{i_1=1}^{D}\frac{\langle \mathbf{w}_{V,m}^{(t)}, \mathbf{x}_{i_1}\rangle}{D}\right)\mathbf{\Gamma}_m\mathbf{\Gamma}_m^\top \mathbf{x}_{i_1}\right]$$

$$= \frac{C(t)}{D}\mathbb{E}\left[\sigma\left(\sum_{i_1=1}^{D}\langle \mathbf{v}_m^*, \mathbf{x}_{i_1}\rangle\right)\sigma'\left(\sum_{i_1=1}^{D}\langle \mathbf{v}_m^*, \mathbf{x}_{i_1}\rangle\right)\sum_{i_1=1}^{D}\langle \mathbf{v}_m^*, \mathbf{x}_{i_1}\rangle\right]\cdot \mathbf{v}_m^* = C(t)F_1(1)\cdot \mathbf{v}_m^*.$$

Plugging the calculation results for $I_1$ and $I_2$ into (G.2), we can immediately derive that $\nabla_{\mathbf{w}_{V,m}}\mathcal{L}(\mathbf{W}_V^{(t)}; \mathbf{W}_{KQ}^{(t)}) = c(t)\cdot \mathbf{v}_m^*$, which, as we stated previously, directly conclude (G.1). In addition, we can further calculate that

$$\mathbf{w}_{V,m}^{(t+1)} = C(t+1)\cdot \mathbf{v}_m^* = \left(C(t) + \eta F_1(1)\big(1 - C(t)\big)\right)\cdot \mathbf{v}_m^*,$$

which implies $C(t)$ possesses the updating rules as:

$$C(t+1) = C(t) + \eta F_1(1)\big(1 - C(t)\big).$$

Subtracting 1 on both sides of the equation above and rearranging the terms, we can obtain that

$$1 - C(t+1) = \big(1 - \eta F_1(1)\big)\big(1 - \eta C(t)\big) = \ldots = \big(1 - \eta F_1(1)\big)^t\big(1 - \eta C(0)\big) = \big(1 - \eta F_1(1)\big)^t.$$

This proves the closed formulation of $C_1(t)$. In the next, we prove that $\nabla_{\mathbf{W}_{KQ}}\mathcal{L}(\mathbf{W}_V^{(t)}; \mathbf{W}_{KQ}^{(t)}) = \mathbf{0}$. By Lemma D.1, we have

$$\sqrt{D}\nabla_{\mathbf{W}_{KQ}}\mathcal{L}(\mathbf{W}_V^{(t)}; \mathbf{W}_{KQ}^{(t)})$$

$$= -\frac{1}{D^2}\sum_{m=1}^{M}\sum_{i=1}^{D}\mathbb{E}\left[\left[\big[f^*(\mathbf{X})\big]_{m,i} - \sigma\left(\sum_{i_1=1}^{D}\frac{\langle \mathbf{w}_{V,m}^{(t)}, \mathbf{x}_{i_1}\rangle}{D}\right)\right]\sigma'\left(\sum_{i_1=1}^{D}\frac{\langle \mathbf{w}_{V,m}^{(t)}, \mathbf{x}_{i_1}\rangle}{D}\right)\right.$$
$$\left.\cdot \sum_{i_1=1}^{D}\sum_{i_2=1}^{D}\langle \mathbf{w}_{V,m}^{(t)}, \mathbf{x}_{i_1}\rangle(\mathbf{p}_{i_1} - \mathbf{p}_{i_2})\mathbf{p}_i^\top\right]$$

$$= -\frac{1}{D^2}\underbrace{\sum_{m=1}^{M}\sum_{i=1}^{D}\mathbb{E}\left[\big[f^*(\mathbf{X})\big]_{m,i}\sigma'\left(\sum_{i_1=1}^{D}\frac{\langle \mathbf{w}_{V,m}^{(t)}, \mathbf{x}_{i_1}\rangle}{D}\right)\sum_{i_1=1}^{D}\sum_{i_2=1}^{D}\langle \mathbf{w}_{V,m}^{(t)}, \mathbf{x}_{i_1}\rangle\mathbf{p}_{i_1}\mathbf{p}_i^\top\right]}_{I_3}$$

$$+\frac{1}{D^2}\underbrace{\sum_{m=1}^{M}\sum_{i=1}^{D}\mathbb{E}\left[\big[f^*(\mathbf{X})\big]_{m,i}\sigma'\left(\sum_{i_1=1}^{D}\frac{\langle \mathbf{w}_{V,m}^{(t)}, \mathbf{x}_{i_1}\rangle}{D}\right)\sum_{i_1=1}^{D}\sum_{i_2=1}^{D}\langle \mathbf{w}_{V,m}^{(t)}, \mathbf{x}_{i_1}\rangle\mathbf{p}_{i_2}\mathbf{p}_i^\top\right]}_{I_4}$$

$$+\frac{1}{D^2}\underbrace{\sum_{m=1}^{M}\sum_{i=1}^{D}\mathbb{E}\left[\sigma\left(\sum_{i_1=1}^{D}\frac{\langle \mathbf{w}_{V,m}^{(t)}, \mathbf{x}_{i_1}\rangle}{D}\right)\sigma'\left(\sum_{i_1=1}^{D}\frac{\langle \mathbf{w}_{V,m}^{(t)}, \mathbf{x}_{i_1}\rangle}{D}\right)\sum_{i_1=1}^{D}\sum_{i_2=1}^{D}\langle \mathbf{w}_{V,m}^{(t)}, \mathbf{x}_{i_1}\rangle\mathbf{p}_{i_1}\mathbf{p}_i^\top\right]}_{I_5}$$

$$-\frac{1}{D^2}\underbrace{\sum_{m=1}^{M}\sum_{i=1}^{D}\mathbb{E}\left[\sigma\left(\sum_{i_1=1}^{D}\frac{\langle \mathbf{w}_{V,m}^{(t)}, \mathbf{x}_{i_1}\rangle}{D}\right)\sigma'\left(\sum_{i_1=1}^{D}\frac{\langle \mathbf{w}_{V,m}^{(t)}, \mathbf{x}_{i_1}\rangle}{D}\right)\sum_{i_1=1}^{D}\sum_{i_2=1}^{D}\langle \mathbf{w}_{V,m}^{(t)}, \mathbf{x}_{i_1}\rangle\mathbf{p}_{i_2}\mathbf{p}_i^\top\right]}_{I_6}.$$

(G.3)

In the next, we discuss the value of $I_3$, $I_4$, $I_5$, and $I_6$ respectively. For $I_3$, it can be calculated as

$$I_3 = C(t) \sum_{m=1}^{M} \sum_{i_1=1}^{D} \mathbb{E}\left[ \sigma\left( \sum_{i_1=1}^{D} \langle \mathbf{v}_m^*, \mathbf{x}_{i_1} \rangle \right) \sigma'\left( \sum_{i_1=1}^{D} \langle \mathbf{v}_m^*, \mathbf{x}_{i_1} \rangle \right) \langle \mathbf{v}_m^*, \mathbf{x}_{i_1} \rangle \right] \mathbf{p}_{i_1} \sum_{i=1}^{D} \mathbf{p}_i^\top$$

$$= \frac{C(t)}{D} \sum_{m=1}^{M} \mathbb{E}\left[ \sigma\left( \sum_{i_1=1}^{D} \langle \mathbf{v}_m^*, \mathbf{x}_{i_1} \rangle \right) \sigma'\left( \sum_{i_1=1}^{D} \langle \mathbf{v}_m^*, \mathbf{x}_{i_1} \rangle \right) \sum_{i_1=1}^{D} \langle \mathbf{v}_m^*, \mathbf{x}_{i_1} \rangle \right] \sum_{i_1=1}^{D} \mathbf{p}_{i_1} \sum_{i=1}^{D} \mathbf{p}_i^\top$$

$$= MC(t)F_1(1) \sum_{i_1=1}^{D} \mathbf{p}_{i_1} \sum_{i=1}^{D} \mathbf{p}_i^\top.$$

The second equation holds as $\mathbb{E}\left[ \sigma\left( \sum_{i_1=1}^{D} \langle \mathbf{v}_m^*, \mathbf{x}_{i_1} \rangle \right) \sigma'\left( \sum_{i_1=1}^{D} \langle \mathbf{v}_m^*, \mathbf{x}_{i_1} \rangle \right) \sum_{i_1=1}^{D} \langle \mathbf{v}_m^*, \mathbf{x}_{i_1} \rangle \right]$ takes identical value for all $i_1 \in [D]$ as they follows the same distribution. Similarly, for $I_4$, we can calculate it as

$$I_4 = \frac{C(t)}{D} \sum_{m=1}^{M} \mathbb{E}\left[ \sigma\left( \sum_{i_1=1}^{D} \langle \mathbf{v}_m^*, \mathbf{x}_{i_1} \rangle \right) \sigma'\left( \sum_{i_1=1}^{D} \langle \mathbf{v}_m^*, \mathbf{x}_{i_1} \rangle \right) \sum_{i_1=1}^{D} \langle \mathbf{v}_m^*, \mathbf{x}_{i_1} \rangle \right] \sum_{i_2=1}^{D} \mathbf{p}_{i_2} \sum_{i=1}^{D} \mathbf{p}_i^\top$$

$$= MC(t)F_1(1) \sum_{i_2=1}^{D} \mathbf{p}_{i_2} \sum_{i=1}^{D} \mathbf{p}_i^\top.$$

This implies that $I_3 = I_4$. Through a similar calculation, we can also get

$$I_5 = I_6 = MC^2(t)F_1(1) \sum_{i_2=1}^{D} \mathbf{p}_{i_2} \sum_{i=1}^{D} \mathbf{p}_i^\top.$$

Plugging the results that $I_3 = I_4$, and $I_5 = I_6$ into (G.3), we immediately concludes that $\nabla_{\mathbf{W}_{KQ}} \mathcal{L}(\mathbf{W}_V^{(t)}; \mathbf{W}_{KQ}^{(t)}) = \mathbf{0}$. This completes the proof. $\qquad \square$

With the conclusions of Lemma G.2, we are ready to prove Theorem G.1.

*Proof of Theorem G.1.* Since we have demonstrated in Lemma G.2 that $\mathbf{W}_{KQ}^{(t)} = \mathbf{0}$, implying $\mathbf{S}^{(t)} = \frac{1}{D} \mathbf{1}_D \mathbf{1}_D^\top$. We can decompose and simplify the loss as

$$\mathcal{L}(\mathbf{W}_V^{(t)}; \mathbf{W}_{KQ}^{(t)}) = \frac{1}{2} \sum_{m=1}^{M} \sum_{i=1}^{D} \mathbb{E}\left[ \left( \mathbf{Y}_{m,i} - \sigma\left( \sum_{i_1=1}^{D} \frac{\langle \mathbf{w}_{V,m}^{(t)}, \mathbf{x}_{i_1} \rangle}{D} \right) \right)^2 \right]$$

$$= \frac{1}{2} \sum_{m=1}^{M} \sum_{i=1}^{D} \mathbb{E}\left[ \left( [f^*(\mathbf{X})]_{m,i} - \sigma\left( \sum_{i_1=1}^{D} \frac{\langle \mathbf{w}_{V,m}^{(t)}, \mathbf{x}_{i_1} \rangle}{D} \right) \right)^2 \right] + \frac{1}{2} \mathbb{E}\left[ \|\mathcal{E}\|_F^2 \right],$$

where the last term is essential $\mathcal{L}_{\mathbf{opt}}$, and the last inequality holds by the independence between $\mathbf{X}$ and $\mathcal{E}$ and the fact that $\mathcal{E}$ is zero-mean. Since this equation holds, in the next, we directly deal with $\mathcal{L}(\mathbf{W}_V^{(t)}; \mathbf{W}_{KQ}^{(t)}) - \mathcal{L}_{\mathbf{opt}}$. By utilizing the fact that $|\sigma(x) - \sigma(y)| \le |x - y|$ for all $x, y \in \mathbb{R}$, we can derive that

$$\mathcal{L}(\mathbf{W}_V^{(t)}; \mathbf{W}_{KQ}^{(t)}) - \mathcal{L}_{\mathbf{opt}} = \frac{1}{2} \sum_{m=1}^{M} \sum_{i=1}^{D} \mathbb{E}\left[ \left( [f^*(\mathbf{X})]_{m,i} - \sigma\left( \sum_{i_1=1}^{D} \frac{\langle \mathbf{w}_{V,m}^{(t)}, \mathbf{x}_{i_1} \rangle}{D} \right) \right)^2 \right]$$

$$\le \frac{1}{2D} \sum_{m=1}^{M} \mathbb{E}\left[ \left( (1 - C(t)) \sum_{i_1=1}^{D} \langle \mathbf{v}_m^*, \mathbf{x}_{i_1} \rangle \right)^2 \right] = \frac{(1 - C(t))^2 \sum_{m=1}^{M} \|\mathbf{v}_m\|_2^2}{2}.$$

Notice that we have derived $1 - C(t) = (1 - \eta F_1(1))^{t-1} \le e^{-\eta F_1(1)(t-1)} \le e^{-\eta(t-1)/2}$ in Lemma G.2, where the last inequality holds as $F_1(1) \ge \frac{1}{2}$ demonstrated by Lemma F.1. Plugging this result into the upper bound above, then we complete the proof.

$\qquad \square$

## H  ADDITIONAL EXPERIMENTS

In this section, we present additional experimental results on transformer learning of bilinear teacher models under more general training data distributions.

Each batch of training data $(\mathbf{X}_n, \mathbf{Y}_n)_{n=1}^N$ is generated with $\mathbf{X}_n$ drawn from either (i). a Student-t distribution with $df = 5$; or (ii). a mean-centered Gumbel distribution with $loc = 0$ and $scale = 1$. We then repeat the learning experiments for the six types of teacher models described in Section 4. Except for the change in the input data distribution, all other configurations remain identical to those in the Gaussian-data experiments.

The results are demonstrated in the following Figures 6, 7, 8, and 9. Figure 6 and 7 report the results when training data are generated from Student-T distribution, while Figure 8 and 9 report the results when training data are generated from mean-centered Gumbel distribution. We could observe that all these results seems almost identical to those demonstrated in main body. Specifically, for both different distributed training data, we can still observe that the curves of training loss have slopes approximately $-1$ on their tails, and the curves of O.O.D. loss have slopes approximates $-0.5$. These results empirically shows that the $\Theta(1/T)$ convergence rate for training loss and $\mathcal{O}(1/\sqrt{T})$ convergence rate for O.O.D. loss still hold, even the model are trained on Gaussian data. In addition, we can also observe that the trained softmax attention scores $\mathbf{S}^{(T)}$ perfectly replicate the patterns of $\mathbf{S}^*$, almost identical to the results obtained on Gaussian data.

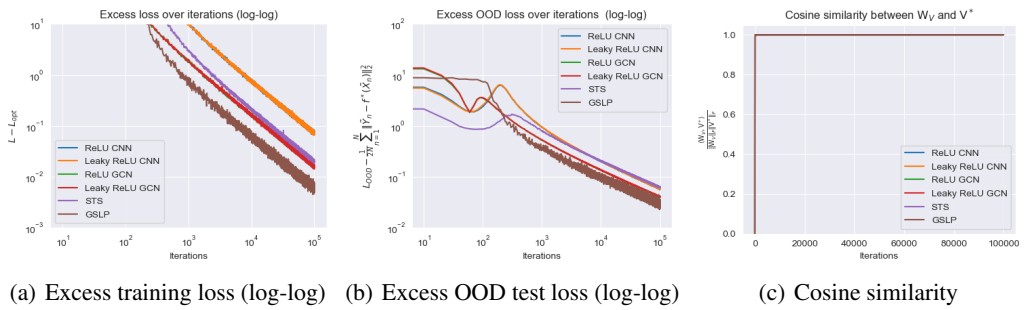

(a) Excess training loss (log-log)  (b) Excess OOD test loss (log-log)  (c) Cosine similarity

Figure 6: Excess training loss, excess OOD test loss (both in log-log scales), and cosine similarity between the value matrix $\mathbf{W}_V$ of one layer transformer (2.4), and ground truth value matrix $\mathbf{V}^*$. These results are presented for experiments where training data is generated from Student-T distribution.

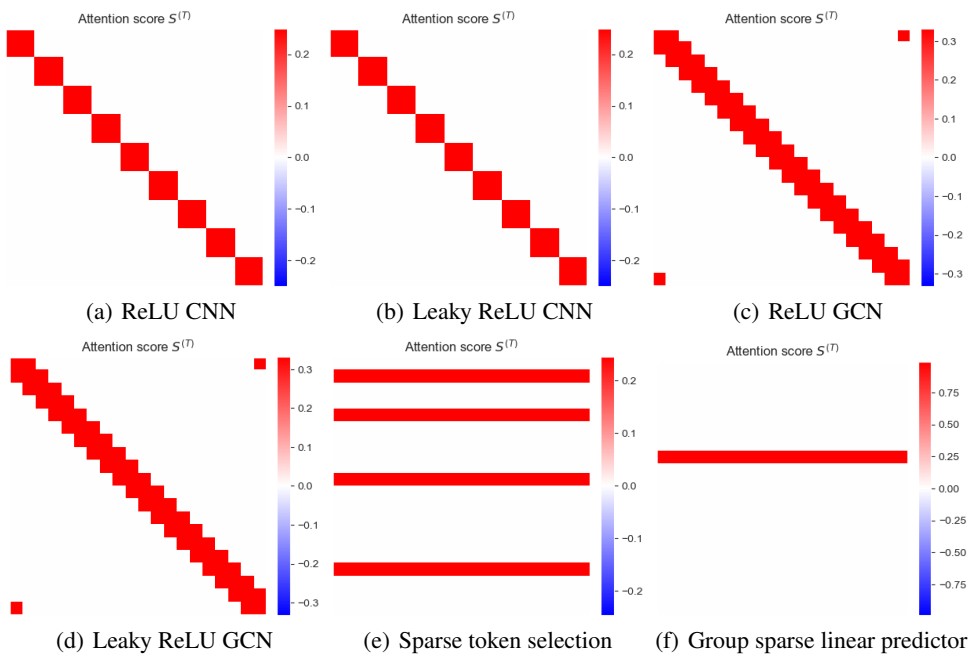

Figure 7: Heatmap of attention score matrix $\mathbf{S}^{(T)}$ when the training loss converges. These results are presented for experiments where training data is generated from Student-T distribution.

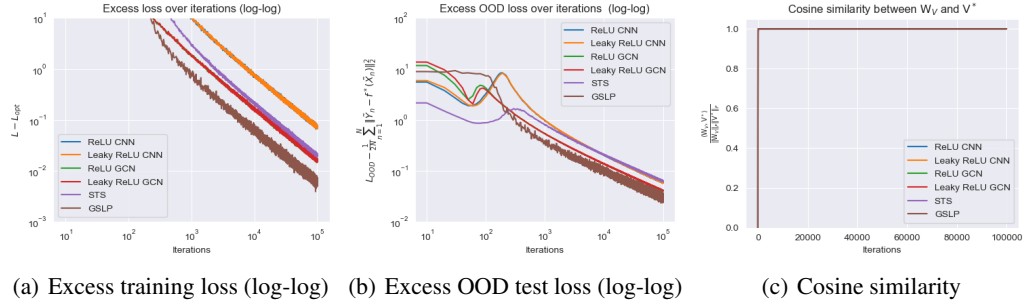

Figure 8: Excess training loss, excess OOD test loss (both in log-log scales), and cosine similarity between the value matrix $\mathbf{W}_V$ of one layer transformer (2.4), and ground truth value matrix $\mathbf{V}^*$. These results are presented for experiments where training data is generated from mean-centered Gumbel distribution.

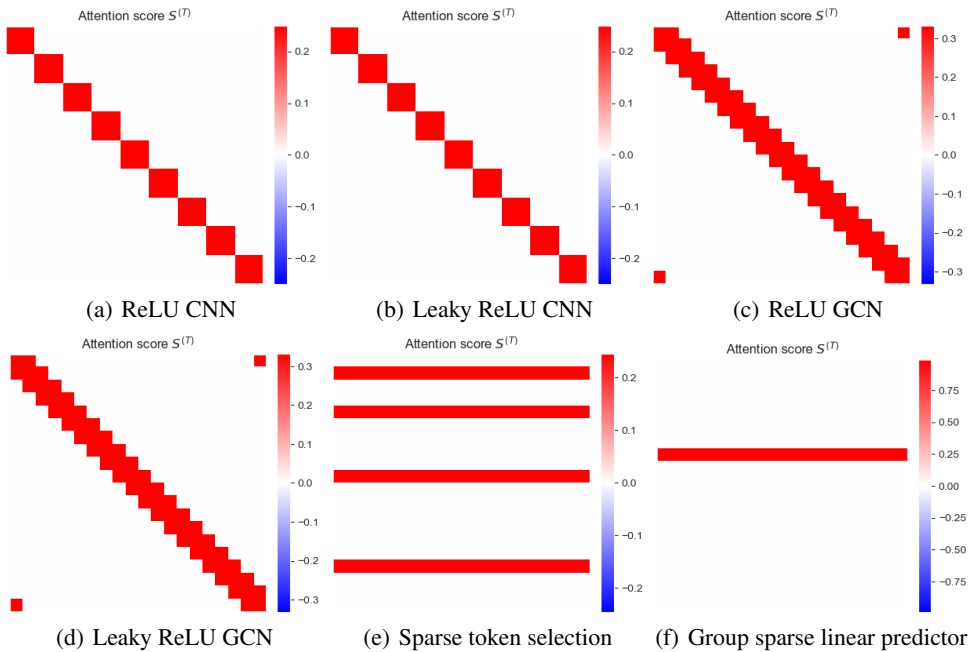

Figure 9: Heatmap of attention score matrix $\mathbf{S}^{(T)}$ when the training loss converges. These results are presented for experiments where training data is generated from mean-centered Gumbel distribution.

