# OpenReview forum: "Transformers Trained via Gradient Descent Can Provably Learn a Class of Teacher Models"
_ICLR.cc/2026/Conference — ICLR 2026 Poster_

### Official Review · Reviewer_hPEb · 2025-10-17

**Soundness:** 2
**Presentation:** 4
**Contribution:** 4
**Rating:** 4
**Confidence:** 4

**Summary:**

The work demonstrates both theoretically and empirically that a modified version of Transformers can mimic a collection of well-known models (GCN, CNN, etc.). Briefly, this modified version replaces the separation between query and key matrices in traditional Transformers with a single parameter and introduces position-only attention weights. Importantly, the authors provide proven polynomial-time convergence rates for the gradient descent-based minimization of the population loss for Gaussian data. On top of this, theoretical claims are well-positioned within the broader literature and clearly supported by a comprehensive empirical analysis.

All in all, the main document is nicely written and provides apparently significant contributions for the theoretical understanding of Transformers. Nonetheless, the supplementary material is very dense and it is difficult to assess the correctness of the theoretical results. Lemma C.2, for instance, states that the rows of the learned value matrix are always a positive multiple of the rows of the true value matrix, and the proof is long and convoluted. Also, it is unclear why the data’s dimensionality should be larger than a polynomial function of $M$ and $K$, and why many of the presented inequalities break when $D = K$ (See weaknesses below).

**Strengths:**

1. The main document is very well written; the notation is clean and easy-to-follow.

2. The proposed convergence bounds are, AFAIK, the tightest known for this specific type of problem (which, as remarked by the authors, has been extensively studied lately).

3. The presented empirical results strongly agree with the introduced theoretical analysis.

4. Assumptions for both the Theorems and experiments are clearly stated (e.g., Gaussian data for in-distribution learning, exponential data for out-of-distribution learning, hypothesis class for optimization, Gaussian noise for the target, etc.).

**Weaknesses:**

My main concern is regarding the lack of a clear explanation for the results obtained through extensive and dense calculation, as I will detail below. I will be more than happy to raise my score in case the following points are addressed.

1. Convergence rates of $\mathcal{O}(1 / T)$ (non-tight) are well-known for strongly convex problems (or, more broadly, for Polyak-Lojasiewicz-type problems). However, this is clearly not satisfied by the squared objective function in this work. Which other regularity conditions do the authors think have allowed these tight bounds to exist?

2. Lemma C.2, as I referred earlier, seems to be a strong result, and its demonstration is broken up into several pages with many hand-waved computations. Could the authors provide a short and intuitive explanation for this result? What is (if there is) a meaning for $C_1$, $C_2$, and $C_3$?

3. Also, demonstrations look invalid when $D = K$, that is, when $\mathbf{S}$ is a simple pooling matrix - this might be the case, e.g., when $g = [D]$ in Example 2.3. What is the fundamental reason for this constraint (if there is such a reason)?

4. On the same note, the bounds also seem to fail when $K = 0$, i.e., when all entries in $\mathbf{S}$ are identically null. As I understand it, this should be the easiest-to-learn scenario. Could the authors briefly discuss this restriction?

5. In line 1088, it is stated that an inequality follows due to Lemma E.1, which only derives a few equalities. Could the authors please elaborate on this regard?

6. Theorem 3.1 bound for $T$ depends linearly on the inverse of the quadratic norm of $\mathbf{V}^{\star}$, while Theorem 3.2 depends linearly on the quadratic norm of $\mathbf{V}^{\star}$. That is, the dependence is inverted in both results. Why is there such a difference between in-distribution and out-of-distribution results?

7. Could the authors provide the slope for the linear curves fitting the tails of Figures 2(a) and 2(b)? Also, do the authors plan to release the code?

**Questions:**

Please refer to the questions above.

On top of that, calculations heavily rely on the Gaussianity of the data. When the data deviates slightly from a Gaussian - a t distribution or a Gumbel distribution - how do the authors believe that the results would behave? Would we keep the polynomial convergence rate, for instance?

---

> ### Author Response · Authors · 2025-11-22
>
> Thanks for your insightful suggestions and recognition of our work. We carefully address your concerns in the following. Due to the space limitations, we have to reply your questions in multiple comments.
> >**Q1:** Explanations of $\Theta(1/T)$ convergence rate as strongly convexity does not hold. Which other regularity conditions do the authors think have allowed these tight bounds to exist?
>
> **A1:** Thank you for the insightful comment. You are absolutely right that the mean-squared loss alone does not ensure convexity of the training objective. Therefore, the classical $\mathcal{O}(1/T)$ convergence guarantee of gradient descent does not directly apply in our setting. We believe that our proof cannot be simply summarized as utilizing any sort of standard regularity conditions of the loss either. Obtaining the tight convergence rate despite the challenging non-convex setting is exactly a strength of our results.
>
> The key reason we can obtain a rate without relying on standard regularity conditions is that, unlike generic optimization analyses, we are studying a class of very concrete and specific learning tasks. We also have the strong intuition that one-layer transformer can solve these learning tasks by having its value and softmax matrices to match the corresponding targets $\mathbf V^\*$ and $\mathbf S^\*$ respectively.
>
> In our proof, we utilize the symmetry properties of the learning tasks to give an accurate characterization of the optimization trajectories (Lemma D.2/Lemma C.2 in original manuscript), and we precisely study the optimization process along these trajectories to obtain tight convergence rates (Lemma D.5, D.15, D.18/Lemma C.5, C.15, C.18 in original manuscript). Please also refer to our reply in **A2** below for more details.
>
>
> >**Q2:** Explanations of Lemma D.2 (Lemma C.2 in original manuscript), specifically the meaning of $C_1(t)$, $C_2(t)$, $C_3(t)$.
>
> **A2:** Lemma D.2 (Lemma C.2 in original manuscript) is indeed a key lemma. It shows that during training, the weight matrices $\mathbf W_V$ and $\mathbf W_{KQ}$ always have the decompositions
> $$\mathbf W_V^{(t)}=C_1(t) \mathbf V^*;$$
> $$\mathbf W_{KQ}^{(t)}=C_{2}(t)\sum_{i=1}^D\sum_{i’\in G^i} \mathbf p_{i’} \mathbf p_{i}^\top - C_{3}(t)\sum_{i=1}^D\sum_{i’\notin G^i} \mathbf p_{i’} \mathbf p_{i}^\top,$$
> $C_1(t)$, $C_2(t)$, $C_3(t)$ are the coefficients in these decompositions. In this way, Lemma D.2 reduces the optimization analysis regarding full matrices $\mathbf W_V$ and $\mathbf W_{KQ}$ into studying the updates of three scalars $C_1(t)$, $C_2(t)$, $C_3(t)$.
> With these decompositions given by Lemma D.2, we can also observe that
> - $\mathbf W_V^{(t)}\to  \mathbf V^\*\Leftrightarrow C_1(t)\to 1;$
> - $\mathbf S^{(t)}\to  \mathbf S^\*\Leftrightarrow C_2(t) + C_3(t)\to \infty.$
>
> **In summary, recovery of the teacher model is entirely determined by the evolution of coefficients $C_1(t), C_2(t), C_3(t)$, and is achieved exactly when $C_1(t)\to 1$ and $C_2(t) + C_3(t)\to \infty$.**
>
> In addition, the decompositions obtained in Lemma D.2 implies that the coefficients $C_1(t),C_2(t),C_3(t)$ essentially follow gradient descent starting from zero initialization minimizing the loss (Here we only demonstrate the results for $\sigma(\cdot)$ being the identity map for simplicity. The general formulation for $\sigma(\cdot)$ is activation function is demonstrated in Lemma D.2):
> $$ \tilde{\mathcal{L}}(C_1,C_2,C_3) \propto \frac{D-K}{K}\Bigg[1-  \frac{KC_1}{K+(D-K)e^{-\frac{C_2 + C_3}{\sqrt{D}}}}\Bigg]^2 + C_1^2\Bigg[1-\frac{K}{K+(D-K)e^{-\frac{C_2 + C_3}{\sqrt{D}}}}\Bigg]^2.
> $$
> Then by carefully analyzing the training dynamics, we can show that for sufficiently large $T$,
> $$C_1(T) -1  = \Theta\bigg(\frac{D^2\sqrt{K}}{\\|\mathbf V^\*\\|\_F\sqrt{\eta T}}\bigg);$$
> $$C_2(T) + C_3(T) = \frac{\sqrt D}{2}\log\bigg(\Theta\Big(\frac{\eta\\|\mathbf V^\*\\|\_F^2}{K^3D^2}\Big)T + e^{\frac{2}{K\sqrt{D}}}\bigg). $$
> The details are provided in  Lemmas D.2, D.5, D.15, D.18, and F.12. Based on these results, we can obtain $\\|\mathbf S^{(T)} - \mathbf S^\*\\|_F, \\|\mathbf W_V^{(T)} - \mathbf V^\*\\|_F = \Theta(1/\sqrt T)$. Under mean-squared loss, the $\Theta(1/\sqrt T)$ convergence of parameters directly suggests that loss will decay at the rate of $\Theta(1/T)$.
>
> We have now added a new proof sketch section in our manuscript to illustrate this procedure.

---

> ### Author Response · Authors · 2025-11-22
>
> >**Q3:** Demonstration looks invalid when $D=K$, or $K=0$ (when all entries of $\mathbf{S}^*$ are identically null). In addition, why is the assumption $D>\mathrm{poly}(M, K)$ required?
>
> **A3:** First of all, we would like to point out that $K=0$ may not be a very well-defined setting. We believe a more formal learning setup should be defined as $V^* = 0$. Please let us know if this is not the setting you meant to ask. In the case when $V^* = 0$, please note that zero initialization is already a global minimizer, and no training is needed.
>
> Regarding the setting of $K=D$, we would like to clarify that this is a significantly simpler setting that can also be accurately analyzed. We have included an additional section Appendix G (Theorem G.1) in the revised manuscript showing linear convergence under this setting.
>
> The reason that we do not include the setting of $K=D$ in our main theorem is exactly because this is a much easier setting yielding faster linear convergence rate, and therefore cannot be unified into Theorem 3.1 which establishes upper and lower bounds on the convergence rate of order $\Theta(1/T)$.
>
> Here we briefly explain why the setting of $K=D$ is significantly easier. Under this case, the ground truth $\mathbf S^\*=\frac{1}{D}\mathbf{1}\_D\mathbf{1}\_D^\top$. Notice that the initialization $\mathbf W\_{KQ}^{(0)}$ can already yield $\mathbf S^{(0)} = \frac{1}{D}\mathbf{1}\_D\mathbf{1}\_D^\top$, achieving an exact recovery of $\mathbf S^\*$ at the start of training. Consequently, the gradient with respect to $\mathbf W\_{KQ}$ remains zero throughout the optimization, and the problem effectively reduces to optimizing the single parameter matrix $\mathbf W_V$. For this reduced optimization task, the loss becomes strongly convex in $\mathbf W_V$, and therefore gradient descent enjoys a linear convergence rate.
>
> Lastly, the assumption $D > \mathrm{poly}(M, K)$ is merely a technical condition introduced to overcome certain analytical challenges and to establish a tight convergence rate. In fact, we have set $D = M = 20$ in several sets of experiments, and this choice does not affect the $\Theta(1/T)$ decay rate of the loss.
>
> >**Q4:** Explanations of derivation in line 1212 (line 1088 in original manuscript).
>
> **A4:** The derivation in line 1212 (line 1088 in original manuscript) follows directly from Lemma F.1 (Lemma E.1 in original manuscript). As defined in equation (D.2) (line 894 in revised manuscript),
> $F_1(a)=\mathbb{E}[x_1 \sigma(x_1)\sigma'(x_1)], \  x_1 \sim \mathcal{N}(0,a).$
> Lemma F.1 provides the explicit form of $F_1(a)$ for three different choices of $\sigma(\cdot)$ considered in our work:
> - Identity map: $F_1(a)=a$;
> - ReLU: $F_1(a)=\frac{a}{2} \le a$;
> - Leaky ReLU: $F_1(a)=\frac{(1+\kappa^2)a}{2} \le a$, where $\kappa$ is the negative-slope parameter of Leaky ReLU.
>
> Therefore, Lemma F.1 implies the general upper bound that $F_1(a) \le a$. In Lemma D.2 (line 997 in revised manuscript), we define that
> $F_1^{(t)} = F_1\left(Kp(t)^2 + \frac{(1-Kp(t))^2}{D-K}\right)$. Thus, $F_1^{(t)} \le Kp(t)^2 + \frac{(1-Kp(t))^2}{D-K}= \frac{D K p(t)^2 - 2Kp(t) + 1}{D-K}$, which is exactly the derivation appearing in Line 1212.

---

> ### Author Response · Authors · 2025-11-22
>
> >**Q5:** The convergence of Theorem 3.1 bound for $T$ depends linearly on the inverse of the quadratic norm of $\mathbf{V}^\*$, while Theorem 3.2 depends linearly on the quadratic norm of $\mathbf{V}^\*$.
>
> **A5:** We first clarify the quantity $T^*$ in Theorem 3.1 is not a convergence rate: it only serves as a burn-in threshold beyond which our asymptotic convergence rate becomes valid. The actual convergence rate in Theorem 3.1 is
> $$\mathcal L^{(T)}-\mathcal L_{\mathrm{opt}} = \Theta\bigg(\frac{KD^4}{\eta T}\bigg),$$ which is irrelevant with the norm of $\mathbf V^\*$. This result is equivalent to a $\epsilon$-loss iteration complexity as:
> - For any small $\epsilon$, there exists $T\_\epsilon = \Theta\Big(\frac{KD^4}{\eta \epsilon}\Big)$, such that $\mathcal L^{(T)}-\mathcal L_{\mathrm{opt}} \leq \epsilon$ holds for any $T\geq T\_\epsilon$.
>
> Obviously, $T\_\epsilon$ here does not depend on $\\|\mathbf V^\*\\|\_F$. In fact, we have some typos in the original manuscript regarding the convergence of $\mathbf S^{(T)}$ and $\mathbf W_{V}^{(T)}$. The factor in the denominator should be $\\|\mathbf V^\*\\|\_F = \sqrt{\sum_{m=1}^M \\|\mathbf v\_m^\*\\|\_2^2}$ instead of $\sum_{m=1}^M \\|\mathbf v\_m^\*\\|\_2$. Consequently, this factor $\sum_{m=1}^M \\|\mathbf v\_m^\*\\|\_2^2$ only appears in the convergence of $\mathbf S^{(T)}$, and it will be cancelled out with $\mathbf V^\*$ in the $f^\*(\cdot)$, rendering the final convergence of loss irrelevant with the norm of $\mathbf V^\*$.
> Next, we explain why a factor $\sum_{m=1}^M \\|\mathbf v\_m^\*\\|\_2^2$​ appears in Theorem 3.2 but not 3.1. The difference arises because Theorem 3.1 assumes the noise $\mathcal{E}$ is mean-zero and independent with $\mathbf{X}$. Consequently, we can decompose the loss as
> $$\mathcal L^{(T)} = \frac{1}{2}\mathbb E\big[\\|\mathbf Y -\mathrm{TF}(\mathbf Z)\\|\_F^2\big] = \frac{1}{2}\mathbb E\big[\\| f^\*(\mathbf X) -\mathrm{TF}(\mathbf Z)\\|\_F^2\big] +  \frac{1}{2}\mathbb E\big[\\|\mathcal E\\|\_F^2\big] + \mathbb E\big[\langle f^\*(\mathbf X) -\mathrm{TF}(\mathbf Z), \mathcal E \rangle\big],$$
> where the last cross term $\mathbb E\big[\langle f^\*(\mathbf X) -\mathrm{TF}(\mathbf Z), \mathcal E \rangle\big]$ is zero as $\mathcal{E}$ is mean-zero and independent with $\mathbf{X}$, and the second term $\frac{1}{2}\mathbb E\big[\\|\mathcal E\\|\_F^2\big]$ is exactly $\mathcal L_{\mathrm{opt}}$. Thus the rate $\mathbb E\\|f^\*(\mathbf X)-\mathrm{TF}(\mathbf Z)\\|\_F^2 = \Theta(1/T)$ directly yields $\mathcal L^{(T)}-\mathcal L_{\mathrm{opt}}=\Theta(1/T)$.
> However, in the O.O.D. setting we do not assume that $\tilde{\mathbf{Y}} -f^\*(\tilde{\mathbf{X}})$ is mean-zero or independent of $\tilde{\mathbf{X}}$. Hence the interaction term does not vanish, and we provide an upper bound using Cauchy–Schwarz as
>  $$\mathbb E\big[\langle f^\*(\tilde{\mathbf{X}})-\mathrm{TF}(\tilde{\mathbf{Z}}), \tilde{\mathbf{Y}} -f^\*(\tilde{\mathbf{X}}) \rangle\big] \leq \sqrt{2\mathbb E\big[\\|\tilde{\mathbf{Y}}\\|\_F^2] +2\mathbb E\big[\\|f^\*(\tilde{\mathbf{X}})\\|\_F^2]} \sqrt{\mathbb E\big[\\|f^\*(\tilde{\mathbf{X}})-\mathrm{TF}(\tilde{\mathbf{Z}})\\|\_F^2\big]}. \ (\*)$$
> The factor $\sum\_{m=1}^M \\|\mathbf v\_m^\*\\|\_2^2$ is exactly introduced as a component of upper bound for $\mathbb E\big[\\|f^\*(\tilde{\mathbf{X}})\\|\_F^2]$ in $(\*)$, by the definition of $f^\*(\cdot)$. Moreover, $\mathbb E\big[\\|f^\*(\tilde{\mathbf{X}})-\mathrm{TF}(\tilde{\mathbf{Z}})\\|_F^2\big]$​ in $(\*)$ exhibits a similar $\mathcal O(1/T)$ decay as in Theorem 3.1, which explains why the overall convergence rate in the O.O.D. case becomes  $\mathcal O(\sqrt{\sum\_{m=1}^M \\|\mathbf v\_m^\*\\|\_2^2/T})$.

---

> ### Author Response · Authors · 2025-11-22
>
> >**Q6:** Could the authors provide the slope for the linear curves fitting the tails of Figures 2(a) and 2(b)? Also, do the authors plan to release the code?
>
> **A6:**
> We report the following two tables of training loss and O.O.D. loss with respect to iteration number in log-scales, where the last column displays the slope of the linear model fitted with $\log$-loss and $\log t$:
>
> Training loss:
> | Model|$\log_{10} t$|4|4.25|4.5|4.75|5|slope|
> |-|-|-|-|-|-|-|-|
> | ReLU CNN||-0.09|-0.37|-0.62|-0.88|-1.13|-1.03|
> | Leaky CNN||-0.12|-0.38|-0.65|-0.89|-1.14|-1.01|
> | ReLU GCN||-0.79|-1.05|-1.29|-1.53|-1.77|-0.97|
> | Leaky GCN||-0.79|-1.05|-1.29|-1.54|-1.81|-1.00|
> | STS||-0.68|-0.95|-1.19|-1.44|-1.70|-1.01|
> | GSLP||-1.23|-1.46|-1.72|-1.95|-2.20|-0.98|
>
> O.O.D loss:
> | Model|$\log_{10} t$|4|4.25|4.5|4.75|5|slope|
> |-|-|-|-|-|-|-|-|
> | ReLU CNN||-0.57|-0.71|-0.85|-0.99|-1.12|-0.55|
> | Leaky CNN||-0.56|-0.71|-0.85|-0.98|-1.11|-0.55|
> | ReLU GCN||-0.74|-0.88|-1.01|-1.14|-1.27|-0.53|
> | Leaky GCN||-0.74|-0.88|-1.01|-1.14|-1.27|-0.53|
> | STS||-0.57|-0.70|-0.83|-0.96|-1.09| -0.52|
> | GSLP||-0.91|-1.07|-1.16|-1.32|-1.44|  -0.52|
>
> We have released our codes on https://anonymous.4open.science/r/trmcnn-856A.
>
> >**Q7:** How do the authors expect the results would behave when data deviates slightly from Gaussian distribution, like T-distirbution or Gumbel distribution.
>
> **A7:** The primary aspect of Gaussian distribution used in our analysis is its symmetry and rotation invariance. Utilizing these properties, we can simplify the optimization concedrning the full parameter to study the evolutions of several key scalars, as detailed in our **A2** to you. Therefore the current proof may be difficult to directly apply to other distributions. While the theoretical analysis is intractable, we found that in experiments the
> loss still exhibits approximately $\Theta(1/T)$ once the data is properly normalized. The following two tables report the training loss when data is generated from Student-t distribution, as well as to a mean-centered Gumbel distribution (i.e., the standard Gumbel distribution after subtracting its mean). The complete experimental results are provided in Appendix H for reference.
>
> Training loss on student-T($df=5$) distribution:
> | Model|$\log_{10} t$|4|4.25|4.5|4.75|5|slope|
> |-|-|-|-|-|-|-|-|
> | ReLU CNN||-0.11| -0.37|-0.62|-0.86|-1.14| -1.01|
> | Leaky CNN||-0.13| -0.37| -0.67| -0.90| -1.13| -1.01|
> | ReLU GCN||-0.80|-1.06|-1.30|-1.53|-1.80| -0.99|
> | Leaky GCN||-0.78|-1.04|-1.27|-1.55| -1.79| -1.01|
> | STS||-0.68|-0.94|-1.18|-1.46|-1.70| -1.02|
> | GSLP||-1.23|-1.46|-1.62|-2.04|-2.20| -1.01 |
>
>
> Training loss on mean-centered Gumbel($loc=0, scale =1$) distribution:
> | Model|$\log_{10} t$|4|4.25|4.5|4.75|5|slope|
> |-|-|-|-|-|-|-|-|
> | ReLU CNN||-0.12| -0.39|-0.62|-0.90|-1.15| -1.02|
> | Leaky CNN||-0.12| -0.38| -0.64| -0.92| -1.12| -1.02|
> | ReLU GCN||-0.80|-1.04|-1.27|-1.53|-1.79| -0.99|
> | Leaky GCN||-0.79|-1.03|-1.30|-1.55| -1.81| -1.03|
> | STS||-0.65|-0.94|-1.19|-1.47|-1.70| -1.06|
> | GSLP||-1.16|-1.48|-1.72|-1.98|-2.18| -1.01 |

---

### Official Review · Reviewer_xeJq · 2025-10-28

**Soundness:** 3
**Presentation:** 3
**Contribution:** 3
**Rating:** 6
**Confidence:** 3

**Summary:**

This paper provides a theoretical analysis of one-layer transformers trained via gradient descent to learn from a class of teacher models with bilinear structure. The teacher models covered include convolutional layers with average pooling, graph convolutional layers on regular graphs, sparse token selection models, and group-sparse linear predictors. The authors prove that transformers with simplified "position-only" attention can recover all parameter blocks of these teacher models, achieving optimal population loss with a convergence rate of Θ(1/T). They also establish out-of-distribution generalization bounds and validate their theory through experiments on both synthetic and real-world (MNIST) data.

**Strengths:**

1. Unified theoretical framework: The paper identifies a fundamental bilinear structure shared across diverse learning tasks, enabling unified learning guarantees. This is a significant conceptual contribution that connects previously studied disparate settings.

2. Tight convergence guarantees: The paper establishes matching upper and lower bounds for the convergence rate of Θ(1/T), improving upon prior work

3. Empirical alignment with theory. Synthetic experiments match predicted slopes and show early directional alignment between learned value matrix and teacher weights.

**Weaknesses:**

1. **Limited model complexity** The analysis is restricted to one-layer transformers with simplified "position-only" attention. While the authors justify this simplification empirically, the gap between this architecture and practical multi-layer transformers with full attention is substantial.

2. **Notation density**: The paper is extremely dense with notation making it difficult to follow. A notation table and more intuitive explanations would improve accessibility.

3. **Scalability concerns**: The convergence time $T^*$ grows quadratically with sequence length D, which may be prohibitive for longer sequences. This scalability issue is not discussed.

4. **Gap between theory and practice:** Theory–practice gap in positional encodings. Assuming a fixed  D×D orthogonal positional matrix departs from learned or sinusoidal encodings used in practice; further clarification is needed to justify this assumption.

**Questions:**

See weaknesses.

---

> ### Author Response · Authors · 2025-11-22
>
> We appreciate your detailed and supportive review. We have carefully addressed your comments and questions as follows.
>
> **Q1:** Simplified one-layer transformers with “position-only” attention and a fixed orthogonal positional matrix.
>
> We acknowledge that our theoretical conclusions rely on several simplifications, including one-layer transformers with position-only attention and orthogonal positional encodings. However, we would like to emphasize that such simplifications are commonly adopted in recent theoretical studies on transformer optimization, and we provide empirical evidence supporting the rationale behind such assumptions. We detail it as follows.
>
>    - The “position-only” attention architecture aligns with our observations in experiments when considering the learning tasks covered in this paper. As demonstrated in Figure 1, even when using a typical TF (2.5) to study our teacher model, only the bottom-right block of $W_{KQ}$ is significantly trained, while the other three blocks remain close to zero, implying that when learning from the teachers considered in the paper, the softmax attention is indeed almost determined by its positional encodings.
>
>    - We would also like to point out that similar theoretical simplifications have been considered in many existing works, with or without empirical justifications. For example,
> [3] adopts a more simplified architecture which treats the softmax attention as $\mathrm{softmax}(\mathbf A)$, where $\mathbf A$ is a trainable matrix. While not directly assuming a ‘position-only’ attention, [1] assumes that the model output corresponds to an input column with $\mathbf x_{\mathrm{query}}=\mathbf 0$, implying that attention is also only determined by position encodings. Similarly, [2] considered positional encodings whose scales dominate the data and hence can show that attention is almost determined by position encodings. [4] considers a much stronger setting where the softmax attention is identical to ours, but fixing the value matrix $\mathbf{W}_V$ during training. In general, our ‘position-only’ attention is equivalent to keeping some parameters in TFs as fixed, which is common in many theoretical works [5, 6, 7, 8, 9, 10] as we have discussed in the paper around line 204.
>
>   - The fixed orthogonal positional encodings are also adopted in most of the theoretical studies of transformers’ optimization involving the positional encodings [1, 3, 4, 5, 11]. We agree that involving the popular positional encodings like RoPE or Sinusoidal encodings is an interesting and promising future work direction for theoretical studies.
>
> >**Q2:** The paper is extremely dense with notation, making it difficult to follow. A notation table and more intuitive explanations would improve accessibility.
>
> **A2:** Thank you for the suggestion. Due to page limitations in the main paper, we placed the comprehensive notation list in Appendix A. In addition, we have now added another notation table in Appendix A, which summarizes the key variables used in the theoretical section to further improve readability and accessibility.
>
> [1]. Wang et al. Transformers provably learn sparse token selection while fully-connected nets cannot.
>
> [2]. Zhang et al. Transformer learns optimal variable selection in group-sparse classification.
>
> [3]. Jelassi et al. Vision transformers provably learn spatial structure.
>
> [4]. Submission to iclr 2026. Data shifts hurt CoT: a theoretical study.
>
> [5]. Wu et al. How many pretraining tasks are needed for in-context learning of linear regression?
>
> [6]. Tarzanagh et al. Transformers as support vector machines.
>
> [7]. Huang et al.  In-context convergence of transformers.
>
> [8]. Sakamoto and Sato. Benign or not-benign overfitting in token selection of attention mechanism.
>
> [9]. Frei and Vardi. Trained transformer classifiers generalize and exhibit benign overfitting in-context.
>
> [10]. He et al. In-context linear regression demystified: Training dynamics and mechanistic interpretability of multi-head softmax attention.
>
> [11]. Nichani et al. How Transformers Learn Causal Structure with Gradient Descent.

---

> ### Author Response · Authors · 2025-11-22
>
> >**Q3:** The convergence time grows quadratically with sequence length $D$, which may be prohibitive for longer sequences.
>
> **A3:** We note that the factor $D^4$ indicates that the convergence takes a large number of iterations when the sequence length $D$ is large. However, the matching lower bound in Theorem 3.1 confirms that this rate is already optimal and cannot be improved under our current setting.
>
> In fact, this polynomial dependence on $D$ originates from two intrinsic aspects of the learning task:
> - Since the loss is the squared Frobenius distance between two $M\times D$ matrices, it necessarily aggregates errors over all $D$ columns, and thus scales proportionally with the sequence length;
> - The $1/\sqrt{D}$ factor appears in the gradients of $\mathbf W_{KQ}$ and requires $\mathbf W_{KQ}$ to scale larger to achieve sufficient convergence, thereby introducing additional factors of $D$ into the convergence rate.
>
> Following your suggestion, we have added discussions regarding these two reasons after Theorem 3.1 in our revision.

---

> > ### Comment · Reviewer_xeJq · 2025-11-26
> >
> > I thank the authors for their detailed rebuttal, including the additional theoretical explanation and experiments in other replies, which has addressed my main concerns. I acknowledge their effort in developing the theoretical framework, which might requires simplified assumptions. I will therefore maintain the positive score of Borderline Accept.

---

> > > ### Author Response · Authors · 2025-11-30
> > >
> > > Dear Reviewer xeJq,
> > >
> > > Thank you for your positive assessment of our work and for the time and effort you devoted to reviewing our manuscript. We truly appreciate your recognition and thoughtful feedback.
> > >
> > > Best regards,
> > >
> > > Authors

---

### Official Review · Reviewer_MrQP · 2025-10-29

**Soundness:** 4
**Presentation:** 4
**Contribution:** 2
**Rating:** 6
**Confidence:** 4

**Summary:**

This work presents a theoretical analysis of the optimization dynamics for learning a class of teacher models via a single layer transformer. This class of models satisfies a bilinear structure, and is general enough to recover the results of previous theoretical analyses. In particular, when optimizing over the population loss of the least squares error, the authors show a tight convergence rate, improving upon previous analyses. An additional result on mild OOD generalization as well as experimental results are provided.

**Strengths:**

- The results provide a tight convergence rate of $\Theta(1/T)$ when training on the population loss, improving previous analyses while generalizing the setting.
- Experimental results for different examples of teacher models are provided, which supports the theoretical analysis.

**Weaknesses:**

- The setting is restricted to the population loss, as well as Gaussian data.
- The generalization of teacher models is nice, but ultimately it is still the idea of sparse selection from the input.

**Questions:**

- Is there any example of a teacher model in this class that does not do sparse selection as a subtask? If not, what are the challenges in characterizing such a class that does something other than sparse selection?
- I know this does not change the takeaway much since experimental results are provided, but what are the challenges in getting sample complexity guarantees? I suspect it will be some online SGD type analysis; would the authors elaborate on this?

I am mostly curious about the first question, and I am happy to raise my score if the authors address it sufficiently.

---

> ### Author Response · Authors · 2025-11-22
>
> Thank you for your insightful and supportive comments. Please find our detailed responses to your questions below​​. Due to the space limitations, we have to answer your questions in multiple comments.
>
> >**Q1:** Simple settings on population loss and Gaussian data. Possibilities to analyze the sample complexities.
>
> **A1:** The setting of population loss minimization under Gaussian data helps simplify our theoretical analysis, as the symmetry in the learning task ensures a highly structured form for the gradients of $\mathbf{W}\_V$ and $\mathbf{W}\_{KQ}$​. This symmetry effectively reduces the optimization over full matrices to a low-dimensional set of scalar factors (Lemma D.2). In principle, one may extend our results to the empirical-loss setting by conducting a uniform convergence analysis that bounds the deviation between the empirical gradient and the population gradient. However, such an extension would introduce substantial additional technical challenges to the current proof, particularly due to the highly coupled attention weights yielded by the softmax normalization. More importantly, such an approach would require a sufficiently large sample size to ensure that the empirical gradient concentrates uniformly. For these reasons, we focus on the setting with population loss to deliver relatively clearer and simpler analyses.
>
> We would like to point out that most of the existing related theoretical studies considered similar settings, i.e. population loss minimization under Gaussian data or Gaussian mixture data [1, 2, 3, 4, 5]. Our work follows this common setting and, compared with these papers, exhibits advantages in the following aspects:
> 1. [1, 2] study one-layer linear transformers, whereas we analyze a more complicated softmax transformer;
> 2. Our model incorporates non-linear activations, which are not considered in [1, 2, 3, 4];
> 3. Our analysis does not require any form of particular warm-start initialization, while [5] critically relies on such assumptions.
>
> [1]. Zhang et al. Trained Transformers Learn Linear Model In-Context.
>
> [2]. Frei and Vardi. Trained transformer classifiers generalize and exhibit benign overfitting in-context.
>
> [3]. Wang et al. Transformers provably learn sparse token selection while fully-connected nets cannot.
>
> [4]. Zhang et al. Transformer learns optimal variable selection in group-sparse classification.
>
> [5]. Jelassi et al. Vision transformers provably learn spatial structure.

---

> ### Author Response · Authors · 2025-11-22
>
> >**Q2:** Is there any example of the teacher model that is not a sparse selection? Any challenges?
>
> **A2:** It is true that our bilinear teacher models all perform a certain kind of "sparse-selection". We adopt this formulation because it already covers several widely used architectures, such as CNN layers with average pooling and GCN layers.
>
> By considering such teacher models performing "sparse-selection", we can also establish unified theoretical guarantees. A critical step in our proof is to establish the separation between the pre-softmax attention scores of the target tokens (intended for selection) and those of the non-target tokens. Specifically, with $G^i$ denoting the set of the target indices for the $i$-th query, the term $\mathbf p_{i'}\mathbf W_{KQ}\mathbf p_{i}$ represents the pre-softmax attention scores of a target token if $i' \in G^i$, and that of a non-target token otherwise. Accordingly, our proof focuses on the gap $\mathbf p_{i_1}\mathbf W_{KQ}\mathbf p_{i} - \mathbf p_{i_2}\mathbf W_{KQ}\mathbf p_{i}$, where $i_1 \in G^i$ and $i_2 \notin G^i$. As this gap diverges to infinity, the softmax function assigns vanishing weight to non-target tokens, effectively causing transformers to perform “sparse selections” solely on the target tokens.
>
> Regarding the setting where the teacher model goes beyond "sparse-selection", we would like to point out there is a trivial case when $K=D$. Under this case, the ground truth $\mathbf S^\*=\frac{1}{D}\mathbf{1}\_D\mathbf{1}\_D^\top$, meaning that the teacher model takes an average among all tokens. The initialization $\mathbf W\_{KQ}^{(0)}$ can already yield $\mathbf S^{(0)} = \frac{1}{D}\mathbf{1}\_D\mathbf{1}\_D^\top$, achieving an exact recovery of $\mathbf S^\*$ at the start of training. Consequently, the gradient with respect to $\mathbf W_{KQ}$ remains zero throughout training, and the problem effectively reduces to optimizing the single parameter matrix $\mathbf W_V$. For this reduced optimization task, the loss can be equivalently treated as a strongly convex function of $\mathbf W_V$, and gradient descent enjoys a linear convergence rate, which is much faster than the convergence rate of $\Theta(1/T)$ in Theorem 3.1. Since the convergence rate of $\Theta(1/T)$ in Theorem 3.1 is established with matching lower bounds; hence this trivial case can not be unified in our Theorem 3.1. We have added section Appendix G (Theorem G.1) in our revised manuscript to provide a rigorous proof for the loss convergence under this case.
>
> As for more general bilinear teacher models in which $\mathbf S^\*$ may contain arbitrary positive entries as long as each column sums to one, we believe that this represents a substantially more challenging setting. In this case, one must ensure that the pre-softmax attention scores preserve precise pairwise differences at convergence so that the softmax function can accurately recover every entry of $\mathbf S^\*$. Consequently, our current proof techniques are not sufficient to characterize the training dynamics in this setting. To the best of our knowledge, most of the existing theoretical analyses of transformer optimization can not cover such general settings.

---

### Official Review · Reviewer_sWSz · 2025-11-03

**Soundness:** 2
**Presentation:** 3
**Contribution:** 2
**Rating:** 2
**Confidence:** 3

**Summary:**

In this paper, the authors study how simplified one-layer transformers with position-only attention trained via gradient descent can learn a class of teacher models defined through a bilinear structure combined with a non-linearity. Included in this class of models are, among others, convolution layers with average pooling, graph convolution layers, sparse token selection, and group sparse linear predictors. Several experiments on synthetic and real-world data are carried out to confirm the theoretical results.

**Strengths:**

The paper is well written and well organized. The proofs are complex and they seem sensible to me, even though I could only superficially go over them.

**Weaknesses:**

My main concerns with the paper are the strong theoretical assumptions and what I perceive is a somewhat unfair comparison with previous works.

Regarding the first point, the assumption that the attention matrix only depends on the position of the tokens is extremely strong. In fact, this simply means that the attention matrix does not depend on the input at all, since the positions are the same for all inputs. (In Eq. (2.4), P is fixed and independent of X). With this assumption, the paper neglects what is arguably the main feature of attention, that is, to attend to other tokens depending on their value. With this assumption, the architecture precisely mimics the teacher class considered (compare Eq. 2.1 with Eq. 2.4), therefore it is not surprising that the simplified architecture can correctly learn this class (albeit, I concede that proving this fact is still challenging).

Regarding the second point, I find the way the authors compared to some previous work quite misleading. Specifically, the authors claim that their work generalizes (Wang et al. 2024), which also consider how a simplified transformer architecture learns the sparse token selection task. However, the two works differ on two important points. Firstly, in (Wang et al., 2024), the fact that the attention only focuses on the positional information of X is shown to be learned during gradient descent, and not assumed from the start. Secondly, and more importantly, in (Wang et al., 2024) the attention matrix depends on the input. In fact, in their work, the sparse subset of indices over which X is averaged is sampled randomly and given as an input to the transformer. The transformer then selects the correct indices of X through the attention layer based on the input. In this paper, instead, the subset of indices over which the average is taken is presumably fixed before training for all inputs. This is a much simpler task, which explains why the attention is chosen to be input independent. Comparing the two works, and in particular their convergence rates, seems unfair.

**Questions:**

I would like the authors to address my main concerns expounded above. In particular, I'd like the authors to discuss how their work is still relevant despite the strong theoretical assumptions (in particular, the input-independent attention), and to discuss more thoroughly how their assumptions compare with those of the previous works they compare to (Wang et al., 2024; Zhang et al., 2025).

As a minor side question, I see that S in (2.1) is assumed to have all non-zero entries to be equal to 1/K. Can your approach be generalized to non-uniform S. What are the main difficulties for this case?

---

> ### Author Response · Authors · 2025-11-22
>
> Thank you for your constructive review and thoughtful questions. We appreciate your feedback, and have addressed your questions as follows. Due to the space limitation, the full response of **A2** is separated into two comments.
>
> >**Q1:** The "position-only" attention assumption is strong.
>
> **A1:** We would like to clarify that, particularly for the class of learning tasks we consider, the ‘position-only’ attention assumption is reasonable from the following aspects:
>
>    - This "position-only" attention architecture aligns with our observations in experiments when considering the learning tasks covered in this paper. As demonstrated in Figure 1, even when using a typical TF (2.5) to study our teacher model, only the bottom-right block of $W_{KQ}$ is significantly trained, while the other three blocks remain close to zero, implying that the softmax attention is almost determined by its positional encodings.
>
>    - We would also like to point out that similar theoretical simplifications have been considered in many existing works, with or without empirical justifications. For example,
> [3] adopts a more simplified architecture which treats the softmax attention as $\mathrm{softmax}(\mathbf A)$, where $\mathbf A$ is a trainable matrix. [2] considered positional encodings whose scales dominate the data and hence can show that attention is almost determined by position encodings. [4] considers a much stronger setting where the softmax attention is identical to ours, but fixing the value matrix $\mathbf{W}\_V$ during training. Particularly, it is true that [1] does not explicitly assume a "position-only" attention.  However, we would like to emphasize that this is because [1] assumes that the model output corresponds to the last input column with its token value $\mathbf x\_{\mathrm{query}}$ always equal to zero. This directly results in the inner product between token values always being zero, and the attention values only depend on positional encodings.
>
>  - In general, our "position-only" attention is equivalent to keeping some parameters in TFs as fixed, which is common in many theoretical works [5, 6, 7, 8, 9] as we have discussed in the paper around line 214 (line 204 in our original manuscript).
>
> For more general learning tasks, we agree that a full version of softmax attention may be necessary. But this is beyond the scope of our work.
>
> >**Q2:** Unfair comparison with [1] as this work considers a much simpler task. More comparison with [1, 2].
>
> **A2:** Thanks for pointing out this issue. We acknowledge that the "sparse token selection" task described in Example 2.3 does not exactly match the setting considered in [1]. Specifically, [1] studies the setting where positions of target tokens are given as a part of the input. Their analysis is conducted under a setting where the model’s output corresponds to the last input query column. Then, [1] assumes that the average of the positional encodings among target positions is provided in the query column. In comparison, we consider a different variant of "sparse token selection" task, for which the target positions are specified as a part of the learning objectives, hence remain fixed for all inputs, and are never fed to the model. We have added a remark after Example 2.3 to clarify this distinction, and revised the discussions regarding our contribution and interpretations of Theorem 3.1.
>
> However, we respectfully disagree with your comment that the task considered in our work is much simpler than that of [1]. Notice that in our case, while the target positions remain fixed, they have never been explicitly provided to the model; hence, the model can only learn them by utilizing the correlations between the output and the input from these target positions. In comparison, [1] adopts a setting where positional information of target indices has been embedded into the query column, which means this positional information can be directly utilized by the model, instead of learning from the correlations between the output and the input from these target positions.
>
> More importantly, we would like to emphasize that the improved convergence rate over [1] stems from our sharper optimization analysis, rather than from any discrepancy in the problem settings. Please note that [1] and our work both rely on the symmetry of Gaussian data and the uniform distribution among the target tokens expected to be selected.
> A critical technical step shared by both analyses is to simplify the optimization regarding the full parameter matrices to investigate the evolutions of several specific scalars, as demonstrated in Lemma 3.2 in [1] and in our Lemma D.2. Specifically, the analysis in [1] tracks the evolution of two scalars, $\alpha(t)$ and $C(t)$, for which the training loss takes the form
> $$\tilde{\mathcal L}(\alpha, C) = \frac{d}{2(D-K)}\Bigg[K(D-K) \bigg(\frac{\alpha}{K+(D-K)e^{-C}}-\frac{1}{K}\bigg)^2 + \alpha^2\bigg(1-\frac{K}{K+(D-K)e^{-C}}\bigg)^2\Bigg] \ (I)$$

---

> ### Author Response · Authors · 2025-11-22
>
> **A2 (continued part):** In our work, we focus on three scalars $C_1(t), C_2(t), C_3(t)$. When the teacher model is reduced to the "sparse token selection" task with $\mathbf V^\* = \mathbf{I}_d$ and without activation function, the excess training loss (see Lemma D.2) can be equivalently expressed as
>
> $$\tilde{\mathcal L}(C_1, C_2, C_3) = \frac{dD}{2(D-K)}\Bigg[K(D-K) \bigg(\frac{C_1}{K+(D-K)e^{-(C_2 + C_3) / \sqrt{D}}}-\frac{1}{K}\bigg)^2 + C_1^2\bigg(1-\frac{K}{K+(D-K)e^{-(C_2+ C_3) / \sqrt{D}}}\bigg)^2\Bigg] \ (II)$$
>
> Comparing two loss functions in $(I)$ and $(II)$, we can observe that they essentially share the same function structure. Specifically, if we regard $(C_2 + C_3) / \sqrt{D}$ in (II) as one term, playing the role as $C$ in $(I)$, then these two functions only differ by a factor $D$. Therefore, while the setting of the "sparse token selection" task in our work is different from that considered in [1], they can be formulated into an essentially identical optimization problem. Notably, the loss $(II)$ in our setting is only the special case with $\mathbf V^\* = \mathbf{I}_d$ and without activation function. Therefore, we believe the setting considered  in our work is more general compared with that in [1], from a technical perspective. This also highlights that establishing a tight convergence rate with a matching lower bound indeed constitutes a technical advantage of our work. We have now included Appendix C to present these discussions in our revised manuscript.
>
> In addition, we compare the settings and assumptions considered in [2] and our work.
>
> 1. [2] considers the group-sparse linear classification task, where the output is given as the sign of the inner product between the target group features and the ground truth linear vector. As study the classification task, they consider binary cross-entropy loss for training the transformer models. In comparison, we consider the group-sparse linear regression, and choose the mean-squared loss for training the transformer models.
> 2. While not explicitly adopting the “position-only” attention setting, [2] consider the positional encodings with large scales. Consequently, softmax attention scores converge within constant steps, and their values are essentially determined by the positional encodings. In comparison, we consider a more continuous optimization process with a small learning rate, matching the common practice.
>
> >**Q3:** Generalizations to non-uniform $\mathbf S^\*$? What are the main difficulties?
>
> **A3:** The current proof highly relies on the symmetry among the target positions and hence might be difficult to directly extend to the setting where $\mathbf S^*$ is non-uniform. Specifically, the critical step in our proof is to establish the separation between the pre-softmax attention scores of the target tokens and those of the non-target tokens. Specifically, with $G^i$ denoting the set of the target indices for the $i$-th query, the term $\mathbf p_{i'}\mathbf W_{KQ}\mathbf p_{i}$ represents the pre-softmax attention scores of a target token if $i' \in G^i$, and that of a non-target token otherwise. Accordingly, our proof focuses on the gap $\mathbf p_{i_1}\mathbf W_{KQ}\mathbf p_{i} - \mathbf p_{i_2}\mathbf W_{KQ}\mathbf p_{i}$, where $i_1 \in G^i$ and $i_2 \notin G^i$. As this gap diverges to infinity, the softmax normalization assigns vanishing weight to non-target tokens, effectively causing transformers to attend to the target tokens.
> However, in the setting where $\mathbf S^\*$ may contain arbitrary positive entries, one must ensure that the pre-softmax attention scores preserve precise pairwise differences at convergence so that the softmax normalization can accurately recover every entry of $\mathbf S^\*$. Consequently, our current proof techniques focusing on establishing a separation between pre-softmax attention scores corresponding to target and non-target tokens, are not sufficient to characterize the training dynamics in this setting. To the best of our knowledge, most of the existing theoretical analyses of transformer optimization can not cover this complicated setting.
>
> [1]. Wang et al. Transformers provably learn sparse token selection while fully-connected nets cannot.
>
> [2]. Zhang et al. Transformer learns optimal variable selection in group-sparse classification.
>
> [3]. Jelassi et al. Vision transformers provably learn spatial structure.
>
> [4]. Submission to iclr 2026. Data shifts hurt CoT: a theoretical study.
>
> [5]. Wu et al. How many pretraining tasks are needed for in-context learning of linear regression?
>
> [6]. Tarzanagh et al. Transformers as support vector machines.
>
> [7]. Huang et al.  In-context convergence of transformers.
>
> [8]. Frei and Vardi. Trained transformer classifiers generalize and exhibit benign overfitting in-context.
>
> [9].  He et al. In-context linear regression demystified: Training dynamics and mechanistic interpretability of multi-head softmax attention.

---

> > ### Comment · Reviewer_sWSz · 2025-11-28
> >
> > I would like to thank the authors for the very detailed response. While I still believe that an input-independent attention is a strong assumption, the thorough comparison between this work and (Wang et al., 2024), which has also been included in the revised manuscript, clarified the authors' contribution, which is fundamentally a generalization of the analysis carried out in (Wang et al., 2024). Since I believe that this generalization is indeed non-trivial and worthy of publication, I will increase my score to 6 at the end of the rebuttal phase.

---

> > > ### Author Response · Authors · 2025-11-30
> > >
> > > Dear Reviewer sWSz,
> > >
> > > We are glad that our clarifications have addressed your concerns regarding the contributions of our manuscript. Your constructive feedback has greatly helped us improve the quality of our work. Thank you for raising your score from 2 to 6, and for the time and effort you invested in reviewing our submission!
> > >
> > > Best regards,
> > >
> > > Authors

---

### Author Response · Authors · 2025-11-22
**Summary of the revisions**

Dear reviewers,

We sincerely thank the reviewers for their thoughtful evaluations and valuable suggestions. We have addressed all comments in detail in the individual responses and revised the manuscript accordingly. We hope that these revisions adequately resolve the concerns raised, and we would be grateful for any additional feedback. The main updates are summarized below.

1. We add a remark after Example 2.3 to clarify the distinctions between the settings of “sparse token selection” tasks covered in our work and those studied in [1]. We also revised the discussion of our contributions and the interpretation of Theorem 3.1 to avoid any potentially misleading comparisons or overstatements. In addition, we added Appendix C, where we carefully compare the optimization dynamics in [1] with those arising in our settings.

2. We add a discussion explaining the polynomial dependence on the sequence length $D$ that appears in the excess loss convergence rate in Theorem 3.1.

3. We add a new proof sketch section in the main body of our manuscript, briefly outlining the main steps to establish the tight $\Theta(1/T)$ convergence rate in Theorem 3.1.

4. We add a notation table in Appendix A, summarizing the key variables and their meanings that appear in the main body of our manuscript.

5. We add a new Theorem G.1 and its proof in Appendix G. This theorem establishes a linear convergence rate for the excess loss when the teacher model$f^\*(\cdot)$ corresponds to the case $D=K$, i.e. $\mathbf S^\* = \frac{1}{D}\mathbf 1_{D}\mathbf 1_{D}^\top$.
Since Theorem 3.1 already provides a matching lower bound which demonstrates the $\Theta(1/T)$ convergence rate is already optimal when $D>K$, this linear rate for the case $D=K$ cannot be unified into Theorem 3.1. The discrepancy in convergence rates precisely explains why the case $D=K$ must be treated separately.

6. We added experiments on synthetic data generated from Student-t and mean-centered Gumbel distributions in Appendix H. These results empirically show that the convergence rates in Theorem 3.1 and Theorem 3.2 continue to hold beyond the Gaussian setting.

We would greatly appreciate any further feedback and are happy to clarify any remaining issues.

Best regards,

Authors

---

### Meta-Review · Area_Chair_8fQA · 2026-01-07

**Summary:**

This paper presents a theoretical and empirical analysis of a simplified one-layer Transformer trained via gradient descent to learn a broad class of teacher models. The considered teacher models encompass several well-known architectures, including CNNs with average pooling, GNNs, and group-sparse linear predictors. The proposed Transformer variant replaces the standard query–key separation with a single parameter matrix and employs position-only attention, enabling a unified analysis. The authors prove that, under Gaussian data assumptions, gradient descent on the population least-squares loss converges in polynomial time with a tight $\Theta(1/T)$ rate, improving upon prior theoretical results. They further establish out-of-distribution guarantees under mild distribution shifts. The theoretical findings are supported by experiments on synthetic data and MNIST.

Reviewers raised relevant issues and requested clarifications, most of which were satisfactorily addressed by the authors through revisions incorporated into the paper. At the end of discussion, most reviewers seem to support acceptance of this paper. In particular, reviewers appear to agree that the paper is well written and makes a meaningful contribution to the understanding of Transformers. In addition, reviewers also emphasized that, despite the simplifying assumptions, the proposed analysis is non-trivial, and that the experimental results support the theoretical findings, which offer tight convergence guarantees. Thus, I am happy to recommend acceptance of this paper.

**Reviewer Concerns:**

Reviewer ``sWSz`` raised concerns regarding the strong assumption of position-only (input-independent) attention and unfair comparison with work by (Wang et al., 2024). In their response, authors overviewed different assumptions in prior works and how their work generalize (Wang et al., 2024). Reviewer acknowledged that the authors detailed answer alleviated the concerns and increased score from 2 to 6.

Reviewer ``MrQP`` first mentioned as weaknesses the simple setting on population loss and Gaussian data. To authors clarified that this simplifies analysis and has been largely used in related theoretical studies, highlighting advantages over these. Another identified weakness was the focus on sparse-selection tasks. The authors justified their choice by arguing it covers several architectures, and entertained with a trivial example beyond sparse selection where the teacher model takes an average among all tokens.

Similarly to ``sWSz``, reviewer ``xeJq`` have concerns regarding the analysis being restricted to one-layer transformers with simplified "position-only" attention. Reviewer ``xeJq`` also found the work difficult to follow due to its heavy notation, and raised concerns regarding scalability and theory–practice gap in the positional encodings studied. Overall, authors clarified the rationale behind their assumptions and the obtained converge rates, and improved readability by including a notation table in Appendix A. The reviewer acknowledged the authors’ response and decided to maintain the initial support through a score of 6.

Finally, reviewer ``hPEb`` mainly raised concerns regarding clarity, mentioning that the supplementary material is very dense and it is difficult to assess the correctness of the theoretical results. In particular, they asked for intuitive explanations/clarifications regarding the derivations in Lemma C.2; possible failure cases for the proposed bounds; differences between in-distribution and out-of-distribution results (Theorems 3.1 and 3.2); and reliance on Gaussian data. This reviewer also asked for the code and additional information related to results in Figure 2. In response, authors have added a proof sketch (related to Lemma C.2), and clarified the results in light of the identified cases, e.g., by adding a new Appendix G that establishes learning guarantees for the non-sparse case $K=D$. The authors also provided code and the requested details regarding Figure 2. Moreover, authors provided additional empirical results for non-Gaussian cases.

**Reviewer Scores:**

Two out of four reviewers engaged in the discussion and acknowledged the rebuttal. In particular, reviewer ``sWSz`` increased score from 2 to 6 following the authors’ clarifications, while ``xeJq`` decided to maintain the score of 6. While they didn’t reply, reviewers ``MrQP`` and ``hPEb`` explicitly indicated in their original review that they would be open to increasing their scores pending satisfactory clarifications. Given the authors’ responses, I believe that their evaluations could have been revised upward had the discussion phase continued.

---

### Decision · Program_Chairs · 2026-01-26

Accept (Poster)